

# A dual-population Constrained Many-Objective Evolutionary Algorithm based on reference point and angle easing strategy

Chen Ji[1], Linjie Wu[1], Tianhao Zhao[1] and Xingjuan Cai[1,2]

[1] Shanxi Key Laboratory of Big Data Analysis and Parallel Computing, Taiyuan University of Science and Technology, Taiyuan, ShanXi, China
[2] State Key Laboratory for Novel Software Technology, Nanjing University, Nanjing, Jiangsu, China

## ABSTRACT

Constrained many-objective optimization problems (CMaOPs) have gradually emerged in various areas and are significant for this field. These problems often involve intricate Pareto frontiers (PFs) that are both refined and uneven, thereby making their resolution difficult and challenging. Traditional algorithms tend to over prioritize convergence, leading to premature convergence of the decision variables, which greatly reduces the possibility of finding the constrained Pareto frontiers (CPFs). This results in poor overall performance. To tackle this challenge, our solution involves a novel dual-population constrained many-objective evolutionary algorithm based on reference point and angle easing strategy (dCMaOEA-RAE). It relies on a relaxed selection strategy utilizing reference points and angles to facilitate cooperation between dual populations by retaining solutions that may currently perform poorly but contribute positively to the overall optimization process. We are able to guide the population to move to the optimal feasible solution region in a timely manner in order to obtain a series of superior solutions can be obtained. Our proposed algorithm's competitiveness across all three evaluation indicators was demonstrated through experimental results conducted on 77 test problems. Comparisons with ten other cutting-edge algorithms further validated its efficacy.

# INTRODUCTION

The constrained multi-objective optimization problem (CMOP) is a type of problem that widely occurs often in real-life scenarios (*Zhao et al., 2023*; *Wang et al., 2023*). One major characteristic of CMOPs is the involvement of diverse and complex constraints related to decision variables or objective functions, which makes it difficult to find ideal or

Corresponding author
Xingjuan Cai, xingjuancai@163.com

approximately ideal solutions. The theoretical model of CMOPs can be presented as:

minimize $\quad F(x) = (f_1(x), ..., f_j(x))^T$

subject to $\quad x \in S$

$$g_m(x) \geq 0, j = 1, 2, ..., M;$$
$$h_n(x) = 0, k = 1, 2, ..., N.$$

Where $x = (x_1, ..., x_j)^T$ denotes the decision vector, and j describes the decision variables count, and $x \in S$, addition $S \in Rn$ is the search area. $g_m(x)$ and $h_n(x)$ represent the m-th inequality constraint and the n-th equality constraint, respectively. m indicates the number of objectives. When m $\geq$4, the equation can be used to describe a constrained many-objective optimization problem (CMaOP). Many real-world applications involve CMaOPs, such as scheduling optimization (*Sindhu & Mukherjee, 2018*; *Zhang et al., 2017*; *Zhang et al., 2021*), routing problems (*Li et al., 2017*; *Shen et al., 2022*; *Shen et al., 2023*; *Guo et al., 2023*), portfolio optimization (*Hemici & Zouache, 2023*; *Wang, Huang & Wang, 2023*; *Setiawan & Rosadi, 2020*), and water resource allocation (*Wang, Wang & Li, 2020*).

For a CMaOP, the extent of the constraint violation (CV) of the p-th constraint for a solution x is formulated as follows:

$$\varphi_p(x) = \begin{cases} \max(0, g_p(x)), p = 1, ..., l \\ \max(0, |h_p(x)|), p = l+1, ..., q \end{cases}.$$

The total CV can be determined by:

$$\varphi(x) = \sum_{j=1}^{P} \varphi_j(x).$$

Over the past few decades, there have been continuous efforts made to solve CMOPs. Algorithms often need to find new and better feasible solutions by bypassing the infeasible regions. Therefore, transforming the original problem into a multi-population-based cooperative optimization problem is an often used approach (*Bao et al., 2023*; *Liang et al., 2023a*). Relevant evidence indicates which methods can effectively balance objectives and constraints (*Liang et al., 2023b*). Previous algorithms have been successful in solving traditional constraint multi-objective optimization problems (CMOPs). However, they have encountered limitations when it comes to constrained many-objective optimization problems (CMaOPs). These challenges predominantly stem from the following:

- They get stuck in local optima caused by overemphasizing convergence. To illustrate the drawbacks of traditional selection strategies, we present the following example. Figure 1 demonstrates the evolution of decision variables and objective values for three traditional optimization algorithms—CTAEA (*Li et al., 2019*), NSGAIII (*Jain & Deb, 2014*), and IDBEA (*Asafuddoula, Ray & Sarker, 2015*)—tackling the C1-DTLZ3 (*Jain & Deb, 2014*) problem. As iterations progress, these algorithms gradually approach the optimal CPFs but often get stuck in local optima. Simultaneously, there is a gradual reduction in the diversity of their decision variables, which is not coincidental. These behaviors are controlled by their selection strategies to move toward CPFs, it is necessary

to promptly eliminate poorly performing solutions. However, this might also prevent the population from finding the final CPF. To further support this, we conducted multiple runs and obtained a notably ideal result. Figure 2 displays the distribution of decision variables and objective values in this ideal result. It is evident that the entire population is uniformly distributed over the CPF, with a significant portion of its decision variables distributed around 0.5. In the results depicted in Fig. 1, the presence of decision variables at 0.5 gradually diminishes with further iterations because these solutions perform poorly in the current environment. However, these solutions are crucial for the population's progression towards CPFs. As a result, some solutions that might not perform well in the current context but are instrumental for the overall optimization process were discarded. It is important to note the distinction in our study compared to *Wang & Xu (2020)*, which focused on infeasible solutions that show good convergence, or *Ming et al. (2023a)*, which concentrated on suboptimal solutions tailored for the current population. Instead, our study prioritized solutions that perform poorly in the current environment but contribute to the overall optimization process. This highlights the need to ease the population's convergence pressure, which is crucial in CMaOPs, as it might prevent the population from converging to CPFs in time. Employing a multi-population collaborative approach is effective but it also brings forth the second issue.

- In the process of population collaboration, it is crucial to strike a balance between exploring feasible and infeasible regions. While exploring infeasible regions, it is also important to avoid missing feasible areas caused by excessively rapid convergence (*Zeng, Cheng & Liu, 2023*), which could result in inefficiency and waste of computational resources. This balance becomes more fragile in CMaOPs. To address this issue, *Zeng, Cheng & Liu (2023)* divided the archive into an inverse archive and a diversity archive, which enables better performance across various types of problems. However, the algorithm relies on the distance between solutions, making them susceptible to local optima influences in smaller dimensions with different value ranges. Moreover, distances lose meaning in higher dimensions (*Myszkowski & Laszczyk, 2021*). Similarly, *Li et al. (2019)* used two archives, CA and DA, to store convergence and diversity solutions. However, in CMaOPs, updating a solution in the archive often requires traversing the entire population for convergence or diversity, which is inefficient. Furthermore, mating-selection strategies lead to deficiencies in exploring new feasible regions (*Yang et al., 2023*). Some dual-population algorithms (*Bao et al., 2023*; *Ming et al., 2022*) utilize a bidirectional search strategy. This approach involves rapidly converging an infeasible solution towards Unconstrained Pareto fronts (UPFs). Subsequently, together with the population of exploring feasible solutions, it gradually moves towards CPFs from both directions. This method proves highly effective in CMOPs. However, in CMaOPs, the emergence of numerous nondominated solutions weakens the population's convergence toward the PF (*Zou et al., 2023*; *Elarbi, Bechikh & Ben Said, 2021*). This prevents the traditional method from moving the population closer toward PF in time. There is a critical need for a search strategy that explores infeasible solutions while balancing convergence and diversity.

- In MaOPs, preserving diversity among solutions, aiming for their uniform distribution along the PF, remains crucial. Two representative approaches to achieving this are reference points (*Jain & Deb, 2014*) and angles among solutions (*He & Yen, 2017*). The former ensures uniformity across the solution space by generating reference vectors, while the latter promptly eliminates poor-quality solutions based on the angles between them. However, in CMaOPs, the scenario differs: the reference points might be uniform, while the CPF might not be, indicating that traditional reference point-based methods cannot guarantee uniform solution distribution (*Wang, Huang & Pan, 2023*). Similarly, the angle-based approach has its drawbacks. To illustrate the limitations of these traditional strategies, consider the following artificial scenario depicted in Fig. 3, which demonstrates the selection process. The solutions—A, B, C, and D——are four non-dominated solutions, where B has a y-coordinate of 0 and one solution needs to be eliminated. If the angle-based method is applied, A or B would be chosen from the AB pair: Solutions dominating B would also have a y-coordinate of 0. Comparatively, the solution space dominating B would be less than that of A. In this case, eliminating A seems like a favorable choice. However, it would drive the population closer to the *x*-axis. In MaOPs, such a strategy could severely damage population diversity. If B is eliminated, there might appear non-dominated solutions with a y-coordinate of 0 but an extremely high x-coordinate, especially in problems with multimodal attributes like DC3-DTLZ3 (*Li et al., 2019*). When B has been eliminated, any new solutions of this type would be non-dominated, consequently compromising the overall quality of the population. The optimization problem of irregular PFs can be effectively addressed by employing neural networks (*Wang, Huang & Pan, 2023*; *Liu et al., 2020*), which produces favorable outcomes. However, it has been noted by *Ming et al. (2023c)* that the excessive integration of multiple techniques tends to complicate the algorithm. Additionally, while some problems have a uniform CPFs, others might not. Algorithms need to balance both scenarios to provide a series of high-performance solutions.

These challenges lead to varying degrees of reduced efficiency when using traditional methods in CMaOPs. Striking the right balance between feasibility, convergence, and diversity has been a critical challenge.

The motivation for this article is as follows:

As a type of meta-heuristic algorithm, the evolutionary algorithm does not rely on problem continuity or differentiability, making it highly suitable for solving CMOPs, particularly CMaOPs with intricate constraints. However, it should be noted that there is no universally versatile meta-heuristic algorithm capable of effectively addressing all types of optimization problems. This concept is known as the No Free Lunch (*Wang et al., 2020*; *Del Ser et al., 2019*). The verification of this theorem also underscores the necessity for continuous theoretical research on meta-heuristic algorithms.

In contrast to other evolutionary algorithms that have undergone substantial development, there has been limited research on CMaOEAs, especially with methods based on multi-population collaborative techniques. However, the performance decreased significantly when using existing CMOEAs to handle CMaOPs (*Ming et al., 2023c*). Hence,

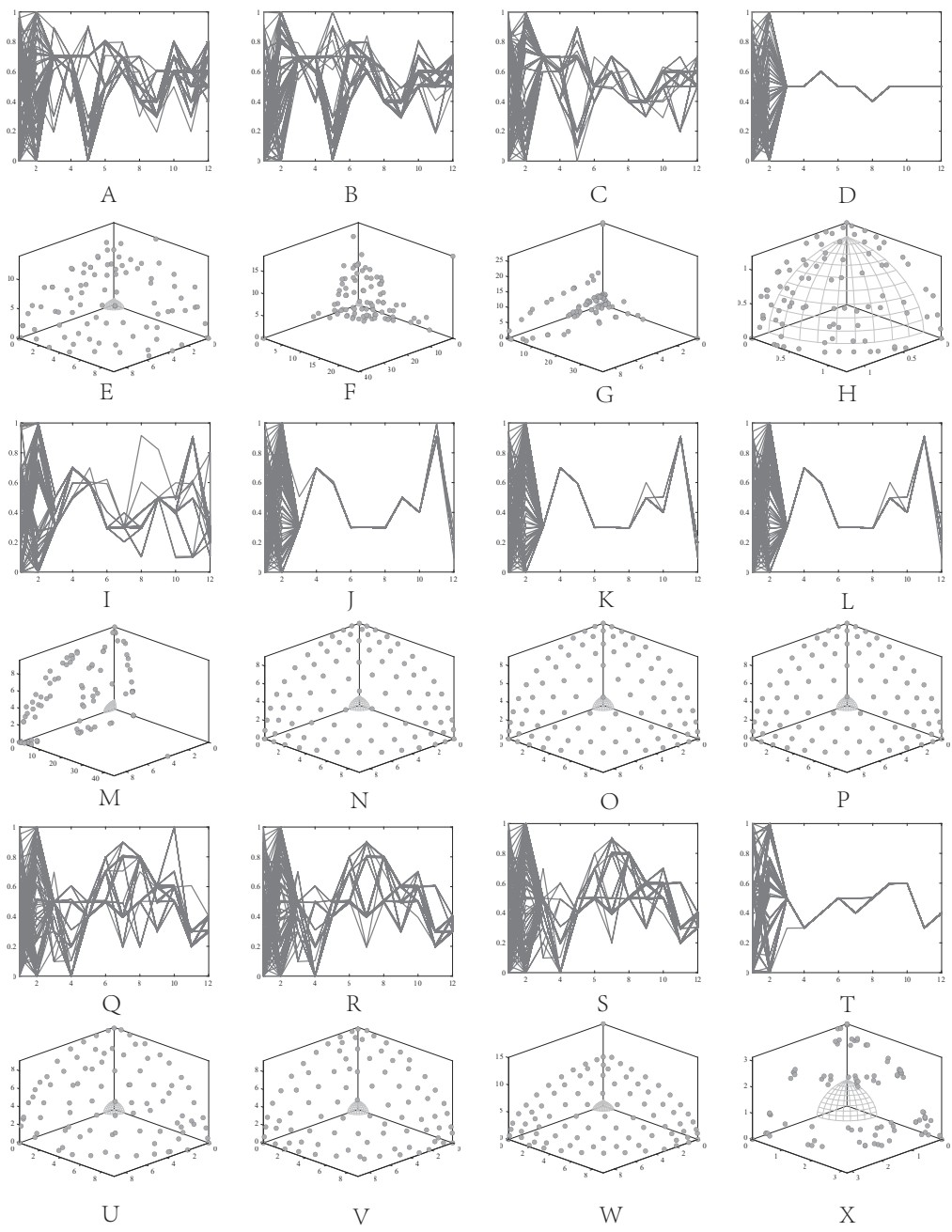

**Figure 1 Changes in decision variables and objective values of the conventional algorithm in C1-DTLZ3.**

conventional multi-objective optimization algorithms are not suitable for CMaOPs, leaving ample room for advancement and potential when using CMaOEA to handle CMaOPs.

The design of effective CMaOEAs requires a proper balance among convergence, diversity, and feasibility (*Ming et al., 2023c*). Traditional multi-population collaborative algorithms perform well in CMOPs. An improved multi-population collaborative

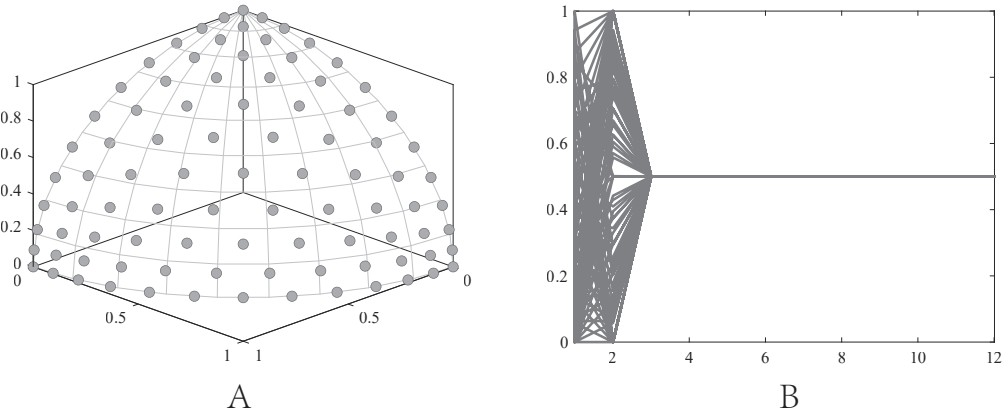

**Figure 2** The ideal result for C1-DTLZ3.

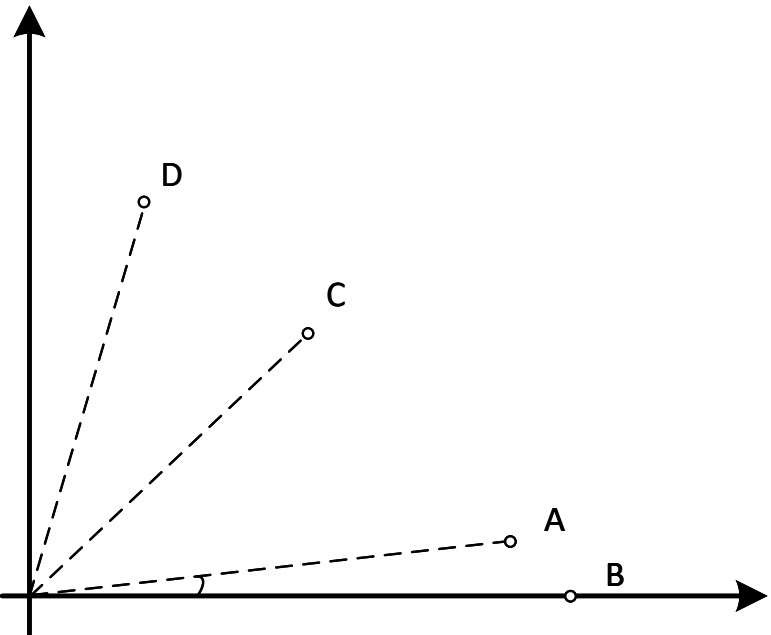

**Figure 3** An artificial scenario depicted.

algorithms could yield better performance and results in CMaOPs if appropriate consideration is given to the feasibility.

In this article, a unique dual-population constrained many-objective evolutionary algorithm based on reference point and angle easing strategy (dCMaOEA-RAE) was developed. This method effectively resolves CMaOPs by striking a balance feasibility, convergence, and diversity. Specifically, we divided the population into two parts: the main population (called PopulationMain) was responsible for searching feasible zones. A selection strategy was proposed to optimize the distribution of the population in

discrete and irregular CPFs by combining reference points and angles. The other auxiliary population (called PopulationExplore) was dedicated to exploring infeasible regions. We retained some current poor but beneficial solutions for the evolutionary process by slowing down the convergence rate of some solutions, thereby uncovering new and superior areas of feasible solutions.

The primary achievements of this study can be briefly described as follows:

- For the main population (PopulationMain), the process involves selecting non-dominated sets from parents and offspring that using the constraints. Subsequently, the solution region was divided into a series of sub-regions using reference vectors. Within each region, the closest solutions to the reference line were initially chosen to ensure the distribution of solutions in the outcome. Then, among the remaining solutions, those closer in angle within the current region and adjacent regions were eliminated. The aim of this was to ensure a more uniform distribution within the population.
- For the auxiliary population (PopulationExplore), solutions were selected in pairs using binary tournaments based on angular distance, favoring solutions with better dominance relationships and distances from the ideal point. This approach enhanced diversity while guiding the population to spread in various directions within the search space. This balance the convergence and diversity, which led the population to superior feasible solution regions.
- Extensive experiments encompassing three test suites with a total of 77 benchmark problems were conducted. The goals were to confirm the effectiveness and competitiveness of dCMaOEA-RAE against 10 advanced CMOEA/CMAOEA methods.

The following is the structure of the remainder of this article: Section 'Related work' offers a short review of the related work on reference point adaptation methods and their underlying motivations. Section 'Proposed algorithm' explains the overall structure of the dCMaOEA-RAE with a detailed description of its components. Section 'Experimental results and analysis' analyzes the experimental setup and comprehensive experiments conducted. Finally, Section 'Conclusions' concludes this article, and highlights further directions for research.

## RELATED WORK

### Existing constraint-handling techniques

With the development of CMAOEA/CMOEA, an increasing number of constraint-handling techniques (CHTs) have been invented. This article briefly reviews some representative techniques in this regard. It first provides a concise introduction to several representative CHTs, followed by an introduction to collaborative optimization. Generally, these techniques can be divided into six classes: (1) Penalty functions; (2) constrained dominance principle (CDP); (3) stochastic ranking (SR); (4) $\varepsilon$-constraints; (5) multi-objective methods (MOs); (6) hybrid methods.

### Penalty function

Penalty functions convert constrained optimization problems into unconstrained problems by incorporating the degree of CV into the objectives. In general, the fitness F'(x) of solution x can be calculated on the basis of penalty functions:

$$F'(x) = F(x) + \beta \cdot \varphi(x)$$

where F(x) represents the fitness of solution x without considering constraints, while $\beta$ is the penalty factor, which can be set in three ways: static, dynamic, or adaptive penalty coefficients. For instance, in *Yahya & Tokhi (2017)*, the authors successfully tackled constrained optimization problems by introducing penalty functions into the bat algorithm, which produced promising results. In *Nargundkar & Kulkarni (2023)*, a combination of cohort intelligence algorithms and penalty functions yielded very promising results. *Chen & Ni (2014)* employed an enhanced logistic chaotic mapping combined with penalty functions to address a resource-constrained project scheduling problem. However, adjusting the penalty factor's parameters to adapt to complex and dynamic problems posed a significant challenge (*Ming et al., 2023b*).

### CDP

CDP was presented by *Jain & Deb (2014)*. Specifically, for two given solutions, A and B, it can be stated that A constraint dominates B (defined as A ≺ B) if any of the below conditions are applicable:

- A is a feasible solution whereas B is not.
- Both A and B are feasible solutions, but A dominates B.
- Both A and B are infeasible solutions, where the CV of A is less than the one of B.

Since it was first proposed, CDP has been widely accepted and used due to its simple structure and ease of implementation. When considering violation degrees as an objective, this handling process can be seen as a one-dimensional search. CDP only moves the population toward lower violation degrees, often leading to the population getting stuck in locally optimal solutions, particularly in cases where feasible regions are discrete.

### ε-constraint

To address the limitations of CDP, *Takahama & Sakai (2006)* relaxed the definition of feasible solutions. They introduced a variable $\varepsilon$, considering solution x feasible as long as its violation degree doesn't exceed $\varepsilon$. The rest remains similar to CDP. *Noman & Iba (2011)* combined differential evolution with the $\varepsilon$-constraint technique to solve economic load dispatch problems. *Wang & Li (2022)* developed a multi-objective distribution planning system using an improved $\varepsilon$-constraint algorithm. Experimental outcomes demonstrated that its optimization results were nearly identical to the optimal path. While the $\varepsilon$-constraint allows the population to break out of local optima, it also leads to unstable outcomes. Additionally, defining the size of $\varepsilon$ remains a challenge (*Ming et al., 2023c*).

### Stochastic ranking

A probability parameter pr (where pr ∈ (0, 1)) is introduced when comparing two individuals. With a probability of pr, only the constraint violation degree is compared.

With a probability of (1 - pr), only the objective functions are compared. This allows the comparison to retain infeasible solutions.

### Multi-objective methods

In this approach, constraints are regarded as one or more independent objective functions, thereby transforming the initial problem into an unconstrained optimization problem. Unlike penalty functions, this method increases the number of objective functions, often leading to multi-objective or even many-objective problems. Different problems pose various challenges to optimization within this framework.

### Hybrid methods

Due to the individual limitations of the aforementioned methods, an increasing number of researchers are combining multiple handling mechanisms to improve the performance of CMAOP/CMOPS. *Wang, Huang & Pan (2023)* observed that in CMaOPs, feasible regions are often irregular and discrete, whereas reference point algorithms assume uniformly distributed feasible regions. Based on this observation, they proposed a constraint-based many-objective evolutionary algorithm, CMaOEA/RPA, which adapts reference points by using a learning vector quantization network to generate feasible region-adaptive reference points. They introduced an adaptive constraint-handling technique based on $\varepsilon$-truncation to incorporate infeasible solutions. Similarly, *Ming et al. (2023a)* combined machine learning with $\varepsilon$-constraint techniques, proposing CMaDPPs, which retain temporarily underperformed solutions to enhance overall performance. This provides a range of high-performance solutions. However, the combination of multiple techniques can lead to algorithmic complexity. Moreover, the limitations of combined CHTs might result in imbalances among convergence, diversity, and feasibility, thereby leading to suboptimal performance (*Ming et al., 2023c*).

## Use the information of the other solutions

In order to effectively push the population to CPFs, solutions which are non-dominated or infeasible have gradually attracted attention due to their clear advantages. In the research of *Wang & Xu (2020)*, an angle-based constrained dominance relation was proposed to use the function of the objective information carried by infeasible solutions. In the research of *Myszkowski & Laszczyk (2021)*, the proposed NTGA2 guides the evolution of the population towards the unexplored parts of the space, promoting diversity and spreading of the population. In the research of *Bao et al. (2022)*, promising solutions were archived to be used to improve search performance. In the research of *Long et al. (2023)*, the proposed EGDCMO used an efficient global diversity strategy to maintain some infeasible solutions. In the research of *Liang et al. (2023c)*, the proposed CMaOEA-AIR explored the potentially feasible regions and escaped from local optima over time by adjusting the selection criteria of infeasible solutions. In the research of *Ming et al. (2023a)*, the authors note that existing algorithms mainly focus on evaluating the quality of individual solutions, instead of evaluating the quality of the overall solutions. They pay more attention to poorly converged, distributed, and infeasible solutions by selecting the population for the next generation in its entirety.

## Multi-population collaborative techniques

To more effectively address the CMOPs/CMaOPs, many researchers have translated the problem into other problems, such as in collaborative optimization. This involves dividing the original population into two functionally different populations. This allows for better exploration of the solution space, uncovers unexplored and potential information, and ultimately obtains a more comprehensive CPFs. In the research of *Chafekar, Xuan & Rasheed (2003)*, the decomposition of a CMOP into multiple optimization problems with a single-objective and the design of different algorithms to optimize each objective have been explored. However, the effectiveness of the algorithm may be compromised when faced with an excessive amount of constraints, potentially leading to local optima if the focus is solely placed on a single objective. In a similar manner, *Wang, Liang & Zhang (2019)* established (M+1) populations. Where M subpopulations were used for the constrained optimization of the single objective, while another population was used for the constrained optimization of the M-objective. Each sub-population optimized its respective problem using differential evolution. To enhance solution diversity, *Yang, Liu & Tan (2021)* partitioned of the objective space, dividing the initial problem into multiple sub-problems and employing multiple CHTs to solve optimization problems. However, irregular feasible regions might lead to poorer performance. In the research of *Liu et al. (2007)*, the proposed COGA assigned populations to optimize objectives and constraints separately while allowing them to exchange information. *Li et al. (2019)* proposed C-TAEA, which maintains two archives concurrently during evolution: one focusing on convergence (CA) and the other emphasizing diversity (DA). The former is used for simultaneous optimization of constraints and objectives, ensuring the final result's reliability, while the latter primarily aims to explore infeasible regions. However, the algorithm suffers from the drawback of updating the archive sequentially, which results in low efficiency. Moreover, the offspring do not exhibit good feasibility and convergence (*Tian et al., 2021*). Considering that information sharing could diminish population diversity, *Tian et al. (2021)* presented a collaborative evolution framework, called CCMO, in which two populations evolve independently. This class of independent ways is known as "weak cooperation" and it enhanced diversity by not sharing information and thereby improving performance. To avoid excessive positive exploration which leads to neglecting feasible solutions, *Ming et al. (2022)* employed a new mechanism in the proposed dual-stages and dual-population constrained multi-objective evolutionary algorithm (DDCMOEA): initially they explored infeasible solutions to UPFs by rapidly converging the population and then allowing both populations to converge towards CPFs simultaneously from two directions. Similarly, *Bao et al. (2023)* used bidirectional searches to enhance search capabilities and exploit infeasible solutions. *Liang et al. (2023b)* noted that when UPFs and CPFs don't fully overlap, the population used for exploration has a diminished role in assisting the main population in later stages. Based on this observation, they proposed dual-population constrained multi-objective evolutionary algorithm with variable auxiliary population size (DPVAPS). Which strategically reduces the computational resource consumption of the exploration population and allows the primary population to devote a larger amount of available resources to the search for CPF. Additionally, in the research of *Zeng, Cheng & Liu (2023)*,

a method for updating diversity and reverse archives was proposed, which was capable of handling fraudulent constraints or narrow feasible areas. In the context of CMaOPs, new environments pose new challenges for algorithms. Some CMaOPs may contain complex constraints, which can make it challenging for the population to traverse multiple disjoint infeasible zones that are too close to the CPFs. *Geng et al. (2023)* introduced NSGA-III based on dual populations, which improved the performance.

## Dealing with complex CPFs

In order to obtain a better distribution of the discrete and discontinuous CPF of the population in many-objective problems, *Wang & Xu (2020)* used angle as an index to judge population density and to evaluate the diversity of the population. However, it did not perform well in the face of some complex CMOPs (such as DC3-DTLZ3 *Jain & Deb, 2014*). In the research of *Cheng et al. (2016)*, there exists two sets of reference vectors, one of which remains uniformly distributed, and the other one adaptively adjusts. In order to adapt to discrete CMaOPs with different sizes, clustering methods are added to the optimization algorithm. For instance, an adaptive strategy based on the k-means clustering method is proposed in the research of *Liu et al. (2022b)*, in which the method used and the PF shape could be fitted gradually over the evolutionary process. In addition, in the research of *Liu et al. (2022a)*, an improved growing neural gas (GNG) was used to adapt the reference vectors in order to solve CMaOPs. In the research of *Wang, Huang & Pan (2023)*, in order to better adjust the reference points toward the feasible regions, both the feasible and infeasible solutions were used as two classes of samples to train the learning model. However, the addition of other techniques can complicate the algorithm and thus affect its performance (*Ming et al., 2023c*).

## PROPOSED ALGORITHM

### Overview of the proposed dCMaOEA-RAE

dCMaOEA-RAE maintains two populations, both of size N. The main population, PopulationMain, is responsible for exploring feasible solutions and providing a uniformly distributed final result with strong convergence. The exploration population, PopulationExplore, explores infeasible solutions to discover new, better feasible solution regions. Additionally, different selection strategies are applied to the two populations to fulfill their distinct functions. Algorithm 1 outlines the overall framework of dCMaOEA-RAE. In the beginning, each population is initialized and reference points are generated. Subsequently, the populations enter an evolutionary loop. Parents are selected using binary tournament which is based on non-dominated sorting. This generates offspring, P'. The offspring and parents together undergo environmental selection. The environmental selection process is conducted differently for the two populations: feasible non-dominated solutions that meet the constraints undergo the environmental selection strategy of the main population. These solutions are screened, based on angles, to eliminate crowded solutions and then form the new generation of the main population. The remaining solutions undergo the environmental selection strategy of the exploration population, creating a

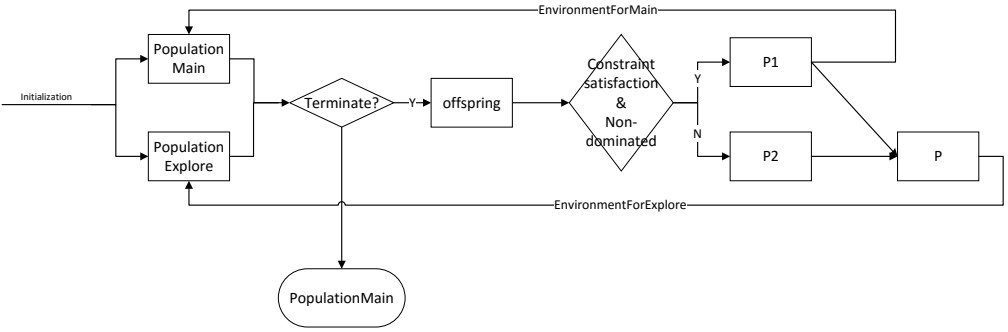

**Figure 4** The flowchart of dCMaOEA-RAE.

new generation of the exploration population. This cycle continues until completion. The proposed algorithm's process flow is illustrated in Fig. 4.

---

**Algorithm 1:** Framework of dCMaOEA-RAE

---

1  Input: N (Population size), (Maximum generation)
2  Output: (final Population set)
3  $Z \leftarrow Reference\_point\_Generation(N)$;
4  $P_{main}, P_{explore} \leftarrow Initialization(N)$;
5  $t \leftarrow 0$;
6  **repeat**
7      FrontNo $\leftarrow$ nondominated_sorting($P_{main} \cup P_{explore}$);
8      MatingPool $\leftarrow$ Select n solutions by FrontNo based binary-tournament-
        selection method as mating parents;
9      P' = OperatorGAhalf(MatingPool);
10     $P = P_{main} \cup P_{explore} \cup P$
11     Calculate the CV values of solutions in P
12     FrontNo $\leftarrow$ nondominated_sorting(P)
13     Pm$\leftarrow$ FrontNo==1 and CV==0
14     Pm = EnvironmentSelectionForMain(Pm,Z)
15     Pe$\leftarrow$ P-Pm
16     Pe = EnvironmentSelectionForExplore(Pe,Z)
17     $t \leftarrow t+1$
18 **until** $t < tmax$;

---

## Environmental selection of PopulationMain

After obtaining a range of feasible non-dominated solutions that exceeded the population size, an environmental selection process was required. As all the solutions within this set are non-dominated, this step focuses on enhancing diversity. Algorithm 2 delineated

---

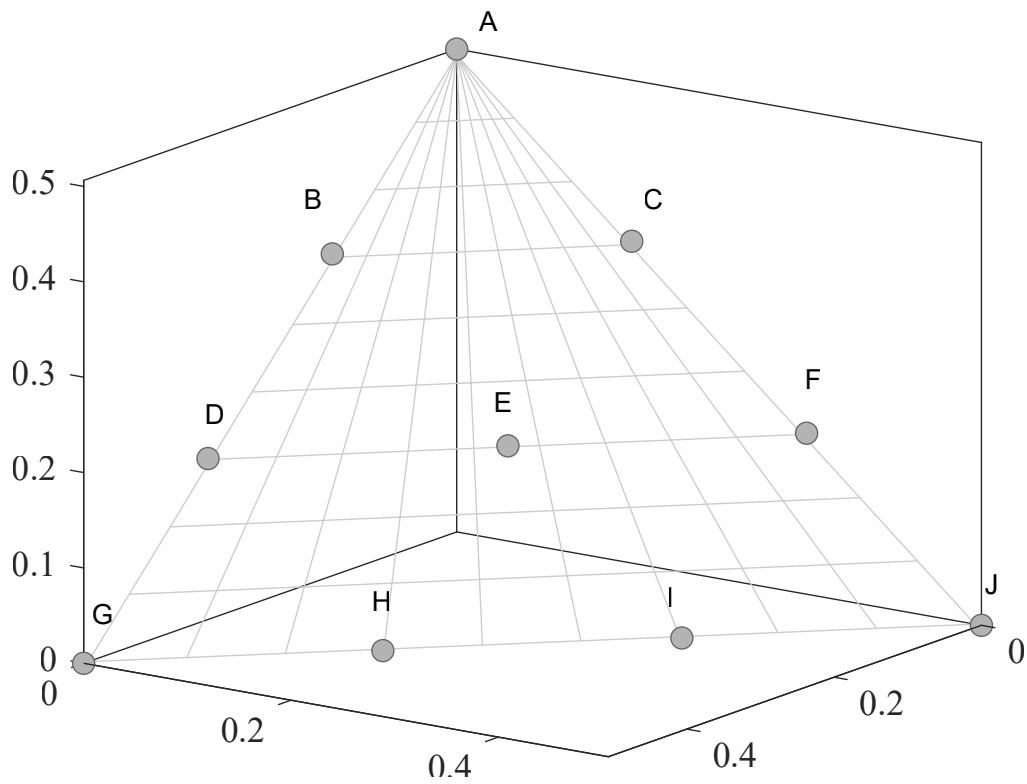

**Figure 5   An example of adjacent weight vectors.**

the population selection process. The search space was divided into subspaces using pre-generated weight vectors and each solution was associated with the vector perpendicularly closest (Line 1). This step effectively reduced the computational complexity. For each subspace, the algorithm first identified the solution closest to the weight vector and designated it as 'the key of this vector' (Line 2). Following this, we identified the weight vector associated with the highest number of connected solutions, denoted by z. (Line 4). We calculated the angles between the solution set S (excluding 'key') associated with the vector z and the set comprising themselves along with adjacent weight vectors (Line 6). The method is further illustrated by Fig. 5, which depicts a hyperplane composed of ten weight vectors (A-J) in a three-objective environment. For F, its adjacent weight vectors were C, E, I, and J. We found the pair of solutions with the smallest angle. At least one solution in this pair should be from set P. If the other solution was not part of set S, we eliminated the other solution (Lines 8-11). Otherwise, we compared these two solutions with the second smallest angle excluding each other. This was used as the criteria for diversity. And eliminated the solution with the smaller angle (Line 13-14). The process continued iteratively utill the number of solutions was same as N.

**Algorithm 2:** EnvironmentSelectionForMain

1  Input: P (The Population), Z (Reference point), N (Population Size)
2  Output: P
3  Normalize P;
4  Assign each solution of P to the nearest reference point by the perpendicular distance.
5  In the set of solutions connected to each reference point, the solution with the shortest distance is selected and named as the 'key' of each reference point.
6  **repeat**
7      Find the reference point that connects the largest number of solutions in the same reference point, denoted as z.
8      The solutions associated with the vector z are denoted as S, while the set of solutions belonging to the reference vectors adjacent to z is denoted as S'.
9      Calculate the Angle between (S - key) and (S' ∪ S), and the result is denoted as R.
10     Find the pair of solutions with the smallest Angle in R, denoted as A and B.
11     **if** $A \notin S$ **then**
12         discard B
13     **else if** $B \notin S$ **then**
14         discard A
15     **else**
16         compute the Angle of the second closest solution of A and B apart from each other, denoted a_ and b_. If a_ > b_ then eliminate B from P and vice versa.
17     **end**
18 **until** *size of P > N*;

## Environmental selection of population explore

After determining the primary population, the remaining individuals required filtration. The exploration population, which disregards the constraints, was used to explore infeasible solutions and discover new feasible regions. Algorithm 3 outlines the selection process for the exploration population.

Similar to Section 'Environmental Selection of PopulationMain' initially, predefined reference points generated weight vectors and were linked to nearby solutions. Subsequently, the algorithm calculated the angles between solution sets associated with each reference vector. It recorded the minimum angle within each reference vector's set of solutions (Line 1). This method avoids traversing the entire population each time a solution is chosen which reduces computational complexity. Then, the algorithm conducted non-dominated sorting of the solution sets and computed the distances of each solution from the origin (Lines 2–3) for assessment.

Entering the 'while' loop, it selected the reference vector corresponding to the minimum value in Angle_min. It picked the two solutions with the smallest angle among these

(Line 6). The algorithm first compared the domination levels of the two solutions; it eliminated the one with the larger domination level (Lines 7-10). If both solutions had equal domination levels, it eliminated the solution farthest from the origin (Line 12). Subsequently, the algorithm updated the Angle_min for that reference vector (Line 14). This process was repeated until the population reached the required size.

---

**Algorithm 3:** EnvironmentSelectionForExplore

---

1 Input: P (The Population), Z (Reference point)

2 Output: P(The screened population)

3 Calculate the minimum angle among the solutions of the set of solutions that connect the same reference point, denoted as Angle_min. If there is only 0 or 1 solution in the solution set, the Angle is denoted as inf.

4 FrontNo ← nondominated_sorting(P)

5 Cons ← The distance of the solution from the origin

6 Normalize P

7 **repeat**

8     Find the reference point where the minimum value in Angle_min lies and identify the pair of solutions with the smallest angle, denoted as A and B;

9     **if** *FrontNo(A) > FrontNo(B)* **then**

10         Eliminate A from P;

11     **else if** *FrontNo(A) < FrontNo(B)* **then**

12         Eliminate B from P;

13     **else**

14         The solution with larger Cons is eliminated;

15     **end**

16     Update Angle_min

17 **until** *size of P > N*;

18 return P;

---

## Remark

Certainly, it is important to note the significant differences between the use of reference points and angles in the search methods of these two populations.

The main population aims for an even distribution to ensure the outcome. We used the 'key' solution to maintain uniform distribution, particularly in cases of a uniform CPF. In situations with non-uniform CPFs, we calculated the angle with adjacent reference vectors, allowing the solutions to distribute as evenly as possible. In Fig. 6, we demonstrate the effectiveness of this strategy through an artificial selection scenario: Fig. 6A comprises four non-dominated solutions, A, B, C, and D, where ABC divides the solution space into four quarters and D is near B. If we needed to eliminate one solution it would be either B or D.

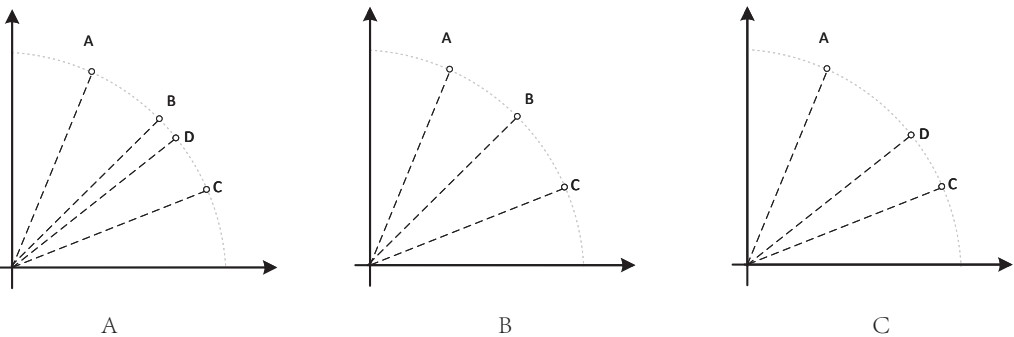

**Figure 6** An artificial selection scenario.

Our proposed search strategy accurately eliminated D by calculating the angle between BD and other solutions.

For the auxiliary population tasked with exploring infeasible solutions, achieving maximum uniformity, unlike the main population, was not crucial. Instead, it aimed to slow down convergence pressure through angle-based strategies, preserving some sub-optimal yet beneficial solutions throughout the evolution process. As the main population used non-dominated sorting to decrease convergence pressure, the overall convergence pressure was guaranteed. Additionally, while comparing non-dominant layers and their distance from the origin are somewhat similar but different, we illustrated this distinction through an example: Fig. 7 displays three solutions, A, B, and C, within a selection environment where one needs elimination. It is clear that AC is non-dominated, and B is dominated by C. When considering their distances from the origin, C < B < A, which would lead to the elimination of A. However, when considering the dominance layer, B is to be eliminated, which is the intended outcome. Therefore, by first comparing dominance layers and then distances, we achieved our intended objective.

## EXPERIMENTAL RESULTS AND ANALYSIS

A concise overview of the experimental setup and algorithm parameters is included in this section. We then evaluated our designed method against ten advanced CMOEAs/CMaOEAs on three representative benchmark suites containing a total of 77 test problems. A performance analysis was carried out on the results. All experiments were performed using PlatEMO (*Tian et al., 2017*).

### Experimental settings
#### Benchmark problems
Similar to most literature, we selected well-established and highly regarded test suites, including C-DTLZ (*Wang & Xu, 2020*), DC-DTLZ (*Li et al., 2019*), MW (*Jiao et al., 2021a*; *Ma & Wang, 2019*), and CF (*Ming et al., 2023a*; *Zhou, Xiang & He, 2021*), which encompassed a total of 16 MaOPs. Each problem was scalable and the objectives varied from 3, 5, 8, 10, and 15. In particular, the DC3-DTLZ1 with 8, 10, and 15 objectives was not included due to the unavailability of ideal points for evaluation on the PlatEMO platform.

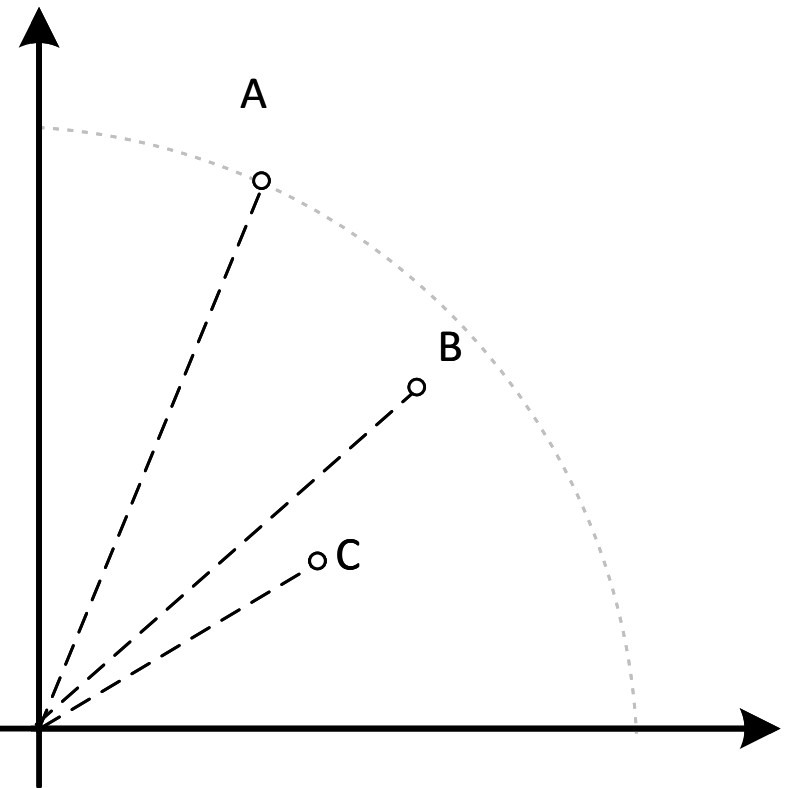

**Figure 7  An example.**

As for the decision variables, following *Ming et al. (2023b)*, for C1-DTLZ1, DC1-DTLZ1, DC2-DTLZ1, and DC3-DTLZ1, the number of decision variables was d = m + 4. For the following cases, C1-DTLZ3, C2-DTLZ2, C3-DTLZ4, DC1-DTLZ3, DC2-DTLZ3, and DC3-DTLZ3, d = m + 9. For MW4, MW8, and MW14, $d = 15$. For CF4, CF8, CF12, d = m + 10.

### The methods used for comparison and parameter settings

We ran comparative trials with four CMOEAs (Top (*Liu & Wang, 2019*), CCMO (*Tian et al., 2021*), DDCMOEA (*Ming et al., 2022*), and BiCO (*Liu, Wang & Tang, 2022*)), and six CMaOEAs (NSGA-III (*Jain & Deb, 2014*), the improved decomposition-based evolutionary algorithm (IDBEA) (*Asafuddoula, Ray & Sarker, 2015*), Two-Archive Evolutionary Algorithm for Constrained Multiobjective Optimization (C-TAEA) (*Li et al., 2019*), TiGE2 (*Zhou et al., 2020*), DCNSGAIII (*Jiao et al., 2021b*), and CMME (*Ming et al., 2023b*)) to show the effectiveness of our work. The selected algorithms are representative of their respective categories: In CMOEAs, Top utilizes DE operators and is a well-established algorithm; CCMO is a highly effective weak-cooperative population-based algorithm; DDCMOEA and BiCo are recent prominent algorithms based on bidirectional search. Among CMaOEAs, NSGA-III is a classic optimization algorithm with significant guidance for many other algorithms; IDBEA relies on decomposition for CMaOPs; C-TAEA

is an archive-based optimization algorithm often included in literature comparisons; TiGE2 converts constraints into a third criterion apart from convergence and diversity; DCNSGAIII and CMME are both notable CMaOEAs introduced in recent years, known for their competitiveness. All of these methods were implemented within PlatEMO (*Tian et al., 2017*).

Referring to the work of *Ming et al. (2023a)* and *Wang, Huang & Pan (2023)*, the population sizes (N) and the maximum fitness evaluation (maxFE) for the algorithm across various test problems are outlined in Tables 1 and 2. Among these, Top utilizes DE-based genetic operators, while other CMaOEAs/CMOEAs employ GA-based operators using simulated binary crossover (*Deb & Agrawal, 2000*) and polynomial mutation (*Edupuganti, Prasad & Ravi, 2010*), with a crossover probability of 1 and a distribution index of 20. The mutation probability $p_m$ is set to 1/n, where n represents the number of decision variables, and the distribution index is set to The parameters for all algorithms are the same as those suggested in their original references in order to maintain fairness. All parameters below are unchanged unless otherwise stated.

All of the algorithms were independently run 30 times on every test case. Both the mean and standard deviation were recorded. Statistics were calculated with MATLAB software (The MathWorks, Natick, MA, USA), using the Wilcoxon test at a significance level of 0.05 and the Friedman test with Bonferroni correction at a significance level of 0.05 to analyze the experimental results. In particular, we transformed the HV metric for the algorithms as follows: $HV' = 1 - HV$, to satisfy the 'the smaller, the better' property required by the Friedman test for the data.

### Performance Indicators

We used several metrics to comprehensively judge the effectiveness of different algorithms: Inverted Generational Distance (IGD), IGDp (*Ishibuchi et al., 2015*), and Hypervolume (HV) (*Zitzler & Thiele, 1999*). Multiple metrics offer enhanced reliability (*Ming et al., 2023c*), providing a more comprehensive assessment of algorithmic performance.

## Comparison results

The designed algorithm was compared with the ten methods mentioned previously. Values such as "NaN" and "0.0000e+0" signify that the results were too distant from the true Pareto front to compute the metric. We used symbols "+", "-", and "=" respectively to indicate results that were statistically superior to dCMaOEA-RAE, significantly inferior to dCMaOEA-RAE, or similar to dCMaOEA-RAE. The result with the best performance in each problem had been bolded. Within the Friedman ranking, in addition to the best data, we bolded the second-best data.

### Comparison results on DTLZ test problem

Table 3 presents the Friedman rankings of eleven algorithms across various objective quantities in the C-DTLZ and DC-DTLZ test problems. It is evident that dCMaOEA-RAE consistently outperformed other algorithms across all three metrics. CMME and DCNSGAIII, as recent CMaOEAs, exhibited relatively good performance. These test sets included diverse feasible regions and offered a comprehensive evaluation of algorithm

**Table 1  The upper limit of the fitness estimates for different problems.**

| Problem | $M = 3$ | $M = 5$ | $M = 8$ | $M = 10$ | $M = 15$ |
|---|---|---|---|---|---|
| C1-DTLZ1 | 46,000 | 127,200 | 124,800 | 276,000 | 204,000 |
| C1-DTLZ3 | 92,000 | 318,000 | 390,000 | 966,000 | 680,000 |
| C2-DTLZ2 | 23,000 | 74,200 | 78,000 | 207,000 | 136,000 |
| C3-DTLZ4 | 69,000 | 265,000 | 312,000 | 828,000 | 544,000 |
| DC-DTLZ and MW | 69,000 | 265,000 | 312,000 | 828,000 | 544,000 |
| CF | 40,000 | 80,000 | 140,000 | 180,000 | 240,000 |

**Table 2  The population size of different problems.**

| Problem | $M = 3$ | $M = 5$ | $M = 8$ | $M = 10$ | $M = 15$ |
|---|---|---|---|---|---|
| C-DTLZ, DC-DTLZ and MW | 92 | 212 | 156 | 276 | 136 |
| CF | 92 | 126 | 156 | 220 | 240 |

performance. Tables 4, 5 and 6 report the IGD, IGDp, and HV performance indices for the test problems, respectively. Looking at the three metrics collectively, it is apparent that dCMaOEA-RAE achieved the highest number of superior rankings in all three metrics. Specifically 34, 31, and 35, respectively, exceeded more than half of the total. Particularly notable performances were observed in C1-DTLZ3, DC1-DTLZ3, DC2-DTLZ1, and DC3-DTLZ3. Due to the complex nature of the constraints, the designed algorithm faced significant challenges in dealing with these types of problems, yet it outperformed other algorithms. For instance, C1-DTLZ3 encounter infeasible obstacles when approaching the PFs and DC2-DTLZ1 contained extensive infeasible regions, which showed the ability of dCMaOEA-RAE to reach superior CPFs through such regions. For example, DC1-DTLZ3 had a narrow feasible region, while DC3-DTLZ3's CPF consisted of a couple of segmented, narrow, tapered strips and a flexible sheet area above the PF. dCMaOEA-RAE strengthened population diversity in the auxiliary population through binary tournaments. It guided the main population toward the precise exploration of minute feasible regions. Additionally, algorithms like CCMO are competitive in solving some problems such as C2-DTLZ2, DC1-DTLZ1, and DC2-DTLZ3 in the 3-objective problems. This competitiveness stems from CCMO's utilization of a weak cooperation approach, where two independent populations evolve. This enhances certain aspects of diversity and enables the discovery of smaller feasible regions. However, the slow convergence of the CCMO as the number of objectives grows prevents it from being suitable for CMaOPs. Furthermore, it is notable that dCMaOEA-RAE exhibited relatively inferior performance in D1-DTLZ1 and DC1-DTLZ1. This could be due to the simplicity of these test problems in terms of the constraint complexity, where even using the straightforward CDP of NSGA-III yielded reasonably good performance.

### Comparison results on MW and CF test problem

Table 7 presents the Friedman ranking of the 11 algorithms in the MW and CF test problems with different numbers of objectives, respectively. Known for their discrete irregularity

**Table 3** The Friedman rankings of eleven algorithms across various objective quantities in DTLZ test problems.

|  | Igd ranking | Igdp ranking | Hv ranking |
|---|---|---|---|
| NSGAIII | 3.89 | 3.97 | 4.60 |
| IDBEA | 8.20 | 8.22 | 8.47 |
| CTAEA | 4.40 | 3.99 | **4.47** |
| TiGE2 | 7.59 | 7.12 | 7.82 |
| DCNSGAIII | **3.50** | 3.57 | **4.47** |
| CMME | 3.62 | **3.42** | 4.71 |
| ToP | 8.75 | 8.76 | 6.68 |
| CCMO | 6.20 | 6.45 | 5.62 |
| DDCMOEA | 6.75 | 7.00 | 6.00 |
| BiCo | 6.43 | 6.72 | 5.86 |
| dCMaOEA-RAE | **1.36** | **1.48** | **1.98** |

**Notes.**
The best results are in bold.

and small feasible ratios, MW and CF highlighted the robust integrated capability of dCMaOEA-RAE in navigating infeasible regions and exploring minute irregular feasible solutions. It is evident that dCMaOEA-RAE consistently secured the top position across all indicators and test problems. It was closely followed by CMME. Tables 8, 9 and 10 illustrate the detailed results of the algorithms using different metrics. CMME was effective certain effectiveness due to its enhanced mating and environmental selection. However, its performance was relatively unstable, resulting in higher standard deviations. Meanwhile, dCMaOEA-RAE demonstrated favorable execution in MW8, CF4, and CF12. These problems encompass various scenarios such as multimodal landscapes, irregular CPFs, convex shapes, and small feasible regions. This showcased dCMaOEA-RAE's adeptness in tackling these complexities, which are challenging for other algorithms.

### Effects of the proposed selection strategy

The proposed dual-population mechanism is further validated in this subsection by employing two variants of dCMaOEA-RAE. We adopted the reference point selection strategy from *Jain & Deb (2014)* to replace the selection strategy of the subject and exploration populations in the dCMaOEA-RAE algorithm, denoted as dCMAOEA-RAE-I and dCMAOEA-RAE-II, respectively. In C-DTLZ and DC-DTLZ experiments with identical parameters as in Section 'Experimental settings', we compared these aforementioned variants and are presented in Table 11. It is worth noting that even when the CPF is non-uniform, the selection strategy for the main population maintained a favorable distribution. Furthermore, the auxiliary population's selection strategy effectively guided the population towards discovering a superior feasible solution region. Figure 8 shows the effectiveness of the three methods used in the DC3-DTLZ3 problem. It is evident that dCMAOEA-RAE-I failed to achieve a uniform distribution on the CPF, while dCMAOEA-RAE-II was unable to locate the CPF within the given time frame. dCMAOEA-RAE achieved the best performance.

Ji et al. (2024), *PeerJ Comput. Sci.*, DOI 10.7717/peerj-cs.2102

**Table 4  The IGD performance values of dCMaOEA-RAE and other schemes on DTLZ benchmark problems.**

| Problem | M | NSGAIII | IDBEA | CTAEA | TiGE2 | DCNSGAIII | CMME | ToP | CCMO | DDCMOEA | BiCo | dCMaOEA-RAE |
|---|---|---|---|---|---|---|---|---|---|---|---|---|
| | 3 | 2.0452e−2 (8.62e−4) = | 4.3115e−1 (6.06e−2) − | 2.2959e−2 (3.13e−4) − | 2.7828e−1 (6.90e−2) − | 2.0328e−2 (2.04e−4) = | **2.0270e−2 (1.09e−4)** = | NaN (NaN) | 2.0796e−2 (1.43e−4) − | 2.0929e−2 (1.90e−4) − | 2.1079e−2 (2.43e−4) − | 2.0304e−2 (1.06e−4) |
| | 5 | 5.2035e−2 (3.62e−4) − | 4.6390e−1 (7.21e−2) − | 5.9488e−2 (4.48e−4) − | 3.2485e−1 (7.57e−2) − | 5.2180e−2 (2.13e−4) − | 5.2317e−2 (3.29e−4) − | NaN (NaN) | 5.2123e−2 (3.83e−4) − | 5.2182e−2 (3.52e−4) − | 5.2005e−2 (2.94e−4) = | **5.1809e−2 (3.13e−4)** |
| C1_DTLZ1 | 8 | 9.5924e−2 (6.26e−4) − | 4.9600e−1 (7.32e−2) − | 1.2051e−1 (2.62e−3) − | 4.4614e−1 (7.41e−2) − | 9.9733e−2 (1.08e−2) − | 9.8088e−2 (4.73e−3) − | 4.4803e−1 (0.00e+0) = | 1.0814e−1 (6.34e−3) − | 1.0586e−1 (1.35e−3) − | 1.0352e−1 (9.60e−4) − | **9.5362e−2 (3.70e−4)** |
| | 10 | 1.0859e−1 (3.77e−4) − | 4.7535e−1 (9.26e−2) − | 1.3984e−1 (1.92e−3) − | 4.7119e−1 (3.66e−2) − | 1.1141e−1 (9.27e−3) − | 1.1105e−1 (7.93e−3) − | 3.3122e−1 (7.98e−2) − | 1.1906e−1 (8.58e−3) − | 1.1743e−1 (1.64e−3) − | 1.1234e−1 (8.15e−4) − | **1.0777e−1 (4.01e−4)** |
| | 15 | 1.8217e−1 (3.43e−4) = | 5.7091e−1 (4.73e−2) − | 2.1316e−1 (7.48e−3) − | 4.9683e−1 (2.94e−2) − | 1.7852e−1 (9.79e−3) − | 1.7490e−1 (1.14e−2) + | 3.5315e−1 (7.93e−2) − | 2.0141e−1 (1.60e−2) − | 2.0306e−1 (1.60e−2) − | **1.6217e−1 (4.91e−3)** + | 1.8093e−1 (2.25e−3) |
| | 3 | 4.5626e+0 (4.01e+0) − | 6.7824e+0 (2.84e+0) − | 7.4347e−1 (1.90e+0) − | 6.7013e+0 (2.77e+0) − | 4.3189e+0 (4.02e+0) − | 2.7092e+0 (3.82e+0) − | 2.5549e+0 (3.74e+0) − | 5.5936e−2 (6.16e−4) − | 5.9077e−2 (1.23e−2) − | 1.1201e+0 (2.76e+0) − | **5.4471e−2 (5.79e−6)** |
| | 5 | 5.9626e+0 (5.41e+0) − | 1.0480e+1 (3.46e+0) − | 8.9830e−1 (4.88e+0) − | 8.1656e+0 (5.06e+0) − | 5.2901e+0 (5.61e+0) − | 7.0263e+0 (5.68e+0) − | 4.6736e+0 (5.19e+0) − | 1.8980e−1 (4.11e−3) − | 2.5247e−1 (5.80e−2) − | 1.9953e−1 (1.81e−2) − | **1.6511e−1 (5.45e−5)** |
| C1_DTLZ3 | 8 | 6.4110e+0 (5.71e+0) − | 1.0124e+1 (4.20e+0) − | 8.8411e+0 (4.85e+0) − | 1.1572e+1 (2.72e+0) − | 5.2733e+0 (5.71e+0) − | 8.7426e+0 (4.98e+0) − | 2.1311e+1 (2.24e+0) − | 1.1604e+2 (1.93e+1) − | 1.2365e+2 (1.70e+1) − | 1.0942e+2 (9.85e+0) − | **3.1542e−1 (4.01e−4)** |
| | 10 | 3.5311e+0 (5.49e+0) − | 1.2517e+1 (5.13e+0) − | 1.1578e+1 (5.44e+0) − | 1.1682e+1 (5.76e+0) − | 4.6522e+0 (6.39e+0) − | 1.1051e+1 (5.87e+0) − | 2.2314e+1 (1.91e+0) − | 1.5002e+2 (1.55e+1) − | 1.4579e+2 (1.48e+1) − | 1.3473e+2 (1.57e+1) − | **4.1996e−1 (4.59e−4)** |
| | 15 | 1.3161e+1 (3.96e+0) − | 1.4881e+1 (7.23e−15) − | 9.3568e+0 (6.24e+0) − | 1.4107e+1 (3.45e+0) − | 1.2142e+1 (5.18e+0) − | 1.3874e+1 (2.48e+0) − | 2.2541e+1 (1.52e+0) − | 1.8118e+2 (1.05e+1) − | 1.7849e+2 (7.35e+0) − | 1.7511e+2 (7.15e+0) − | **6.2513e−1 (1.07e−2)** |
| | 3 | 4.8388e−2 (3.82e−4) − | 8.0929e−1 (1.87e−1) − | 5.6440e−2 (1.21e−3) − | 1.4636e−1 (2.94e−2) − | 4.8131e−2 (4.14e−4) − | 5.5408e−2 (9.57e−4) − | 1.5413e−1 (2.63e−1) − | **4.4865e−2 (6.60e−4)** + | 4.5238e−2 (7.34e−4) + | 4.5092e−2 (5.79e−4) + | 4.6736e−2 (4.47e−4) |
| | 5 | 1.3880e−1 (3.21e−4) − | 9.0861e−1 (1.95e−1) − | 1.4675e−1 (1.21e−3) − | 2.2595e−1 (1.51e−2) − | 1.3864e−1 (8.98e−4) − | 1.4490e−1 (1.22e−3) − | 2.0163e−1 (4.13e−3) − | 1.4317e−1 (1.91e−3) − | 1.4302e−1 (2.45e−3) − | 1.3954e−1 (1.29e−3) − | **1.3730e−1 (3.72e−4)** |
| C2_DTLZ2 | 8 | 3.0943e−1 (1.66e−1) − | 1.3201e+0 (1.37e−1) − | 2.3839e−1 (1.75e−3) − | 2.9756e−1 (5.99e−2) − | 2.6084e−1 (6.99e−2) − | 2.3830e−1 (2.46e−1) − | 7.6692e−1 (2.10e−1) − | 2.7862e−1 (5.88e−3) − | 3.0119e−1 (1.12e−1) − | 2.7864e−1 (4.68e−3) − | **2.3475e−1 (1.03e−3)** |
| | 10 | 2.9867e−1 (8.17e−2) − | 1.2785e+0 (1.23e−1) − | 3.3770e−1 (4.50e−2) − | 3.9815e−1 (3.68e−2) − | 2.9453e−1 (7.55e−2) − | 2.8284e−1 (2.65e−2) − | 7.6441e−1 (2.07e−1) − | 3.7589e−1 (5.37e−2) − | 3.8182e−1 (5.30e−2) − | 3.0048e−1 (3.34e−2) − | **2.5342e−1 (1.75e−3)** |
| | 15 | 4.0315e−1 (1.95e−1) − | 1.4436e+0 (1.27e−1) − | 2.6972e−1 (5.49e−2) + | 3.5847e−1 (1.54e−2) − | 3.2636e−1 (7.16e−2) − | **2.6685e−1 (6.67e−3)** = | 1.0485e+0 (4.41e−2) − | 4.0502e−1 (1.17e−1) − | 3.7823e−1 (3.44e−2) − | 3.5942e−1 (4.34e−3) − | 2.7792e−1 (3.85e−2) |
| | 3 | 1.6714e−1 (2.29e−1) − | 4.9293e−1 (3.69e−1) − | 1.1251e−1 (2.84e−3) − | 1.9931e−1 (1.08e−2) − | 1.4715e−1 (1.89e−1) − | 1.0837e−1 (2.90e−3) − | 1.4903e−1 (4.12e−3) − | 1.7413e−1 (2.26e−1) − | 1.2544e−1 (1.35e−1) − | 1.0162e−1 (1.71e−3) − | **9.6727e−2 (6.76e−3)** |
| | 5 | **2.4577e−1 (9.07e−4)** + | 4.2896e−1 (6.00e−2) − | 2.9058e−1 (2.70e−2) − | 4.1223e−1 (1.71e−2) − | 2.9674e−1 (1.26e−1) − | 2.8014e−1 (2.23e−1) − | 4.3753e−1 (2.15e−2) − | 2.9621e−1 (5.60e−3) − | 2.9659e−1 (4.37e−3) − | 3.1528e−1 (6.81e−3) − | 2.4646e−1 (9.17e−4) |
| C3_DTLZ4 | 8 | 4.8463e−1 (8.85e−2) − | 7.8663e−1 (1.00e−1) − | 5.4737e−1 (3.61e−3) − | 6.3383e−1 (5.52e−3) − | 6.5925e−1 (7.66e−2) − | 5.3235e−1 (4.01e−2) − | 1.1467e+0 (2.29e−1) − | 2.1495e+0 (1.55e−2) − | 2.1532e+0 (2.35e−2) − | 1.0767e+0 (5.14e−2) − | **4.5240e−1 (1.37e−3)** |
| | 10 | 5.7647e−1 (2.95e−2) − | 7.4832e−1 (6.16e−2) − | 6.2254e−1 (5.49e−3) − | 7.4351e−1 (1.01e−2) − | 6.8257e−1 (2.99e−2) − | 6.1341e−1 (5.10e−2) − | 1.5677e+0 (2.24e−1) − | 2.2620e+0 (8.41e−3) − | 2.2605e+0 (9.40e−3) − | 1.2231e+0 (6.93e−2) − | **5.6575e−1 (1.92e−3)** |
| | 15 | 8.8392e−1 (7.18e−2) − | 1.2092e+0 (2.13e−1) − | **8.0478e−1 (8.14e−4)** = | 8.6019e−1 (1.05e−2) − | 8.9130e−1 (6.60e−2) − | 8.0885e−1 (1.06e−2) − | 2.0933e+0 (1.65e−1) − | 2.4880e+0 (9.28e−3) − | 2.4897e+0 (9.81e−3) − | 1.7600e+0 (1.66e−1) − | 8.0729e−1 (7.30e−3) |
**Table 4** (*continued*)

| Problem | M | NSGAIII | IDBEA | CTAEA | TiGE2 | DCNSGAIII | CMME | ToP | CCMO | DDCMOEA | BiCo | dCMaOEA-RAE |
|---|---|---|---|---|---|---|---|---|---|---|---|---|
| | 3 | 1.4171e−2 (2.08e−4) - | 8.5097e−2 (7.94e−2) - | 1.5193e−2 (2.86e−4) - | 6.9675e−1 (3.91e−1) - | 1.3659e−2 (1.71e−4) - | 1.5771e−2 (1.93e−4) - | 3.9118e−2 (5.26e−2) - | **1.2028e−2 (9.49e−5)** + | 1.2193e−2 (1.54e−4) + | 1.2155e−2 (1.76e−4) + | 1.3441e−2 (1.79e−4) |
| | 5 | 3.9838e−2 (1.66e−4) - | 1.0296e−1 (8.89e−2) - | 4.2027e−2 (4.10e−4) - | 4.1156e−1 (1.82e−1) - | 4.0467e−2 (1.37e−4) - | 4.2269e−2 (3.45e−4) - | 1.4487e+0 (2.95e+0) - | 4.2931e−2 (8.93e−4) - | 4.3592e−2 (1.10e−3) - | 4.3503e−2 (8.31e−4) - | **3.9222e−2 (2.01e−4)** |
| DC1_DTLZ1 | 8 | 7.9405e−2 (3.49e−4) - | 1.5747e−1 (1.01e−1) - | 8.3722e−2 (7.83e−4) - | 4.4538e−1 (1.39e−1) - | 7.9243e−2 (5.95e−3) - | 7.9963e−2 (7.57e−4) - | 1.9710e+0 (3.14e+0) - | 9.5292e−1 (7.71e−1) - | 2.4032e+0 (1.71e+0) - | 2.7226e+0 (6.16e+0) - | **7.8712e−2 (4.62e−4)** |
| | 10 | 8.6661e−2 (2.54e−3) - | 1.3065e−1 (6.75e−2) - | 9.4437e−2 (2.20e−3) - | 3.8517e−1 (1.32e−2) - | 9.1340e−2 (1.57e−2) - | **7.8762e−2 (2.11e−3)** + | 9.0116e−1 (2.35e+0) - | 1.5949e+1 (2.66e+1) - | 8.4254e+0 (1.31e+1) - | 6.6049e+1 (4.30e+1) - | 8.5535e−2 (3.50e−4) |
| | 15 | 1.2831e−1 (4.74e−1) = | 3.7718e−1 (1.00e−1) - | 1.4233e−1 (6.33e−3) - | 4.7192e−1 (1.85e−1) - | 1.3542e−1 (6.46e−3) - | 1.3437e−1 (6.44e−3) - | 1.6273e−1 (2.72e+0) - | 1.7559e+2 (3.41e+1) - | 1.9601e+2 (3.27e+1) - | 1.7509e+2 (4.41e+1) - | **1.2824e−1 (3.56e−5)** |
| | 3 | 4.6182e−2 (8.08e−4) - | 7.5739e−2 (6.20e−1) - | 4.3119e−2 (1.24e−3) - | 2.4204e+0 (6.29e−1) - | 4.5442e−2 (8.81e−4) - | 5.0164e−2 (2.60e−3) - | 1.8304e+0 (2.63e+0) - | **3.5491e−2 (2.87e−4)** + | 1.1967e−1 (1.22e−2) - | 3.6419e−2 (6.04e−4) + | 3.9421e−2 (9.47e−4) |
| | 5 | 1.4439e−1 (7.55e−4) - | 8.8198e−1 (4.30e−1) - | 2.1898e−1 (2.90e−2) - | 1.9932e+0 (5.10e−1) - | 1.4426e−1 (7.68e−4) - | 1.4814e−1 (1.26e−3) - | 2.7296e+0 (1.49e+0) - | 1.5380e−1 (3.49e−2) - | 2.3892e−1 (4.90e−2) - | 1.5655e−1 (5.05e−3) - | **1.4282e−1 (8.97e−4)** |
| DC1_DTLZ3 | 8 | 4.6456e−1 (1.06e−1) - | 1.4782e+0 (6.35e−1) - | 4.8976e−1 (3.74e−2) - | 1.8398e+0 (4.67e−1) - | 4.3037e−1 (1.13e−1) - | 4.8240e−1 (7.81e−2) - | 1.9521e+1 (6.68e+0) - | 9.7315e+1 (1.04e+1) - | 9.6550e+1 (9.92e+0) - | 9.2617e+1 (1.15e+1) - | **3.2833e−1 (3.88e−2)** |
| | 10 | 5.0848e−1 (3.36e−2) - | 1.1976e+0 (1.52e−1) - | 5.0451e−1 (2.88e−2) - | 1.5937e+0 (2.42e−1) - | 4.7635e−1 (5.52e−2) - | 5.3172e−1 (4.02e−2) - | 2.1042e+1 (5.77e+0) - | 1.2820e+2 (1.96e+1) - | 1.2783e+2 (2.22e+1) - | 1.2017e+2 (1.64e+1) - | **4.2703e−1 (3.11e−2)** |
| | 15 | 7.0664e−1 (5.26e−2) - | 1.4486e+0 (4.16e−1) - | 7.5391e−1 (6.29e−2) - | 1.3974e+0 (1.14e−1) - | 7.2232e−1 (6.14e−2) - | 6.7388e−1 (1.68e−2) - | 1.9561e+1 (6.34e+0) - | 1.8314e+2 (7.12e+0) - | 1.8222e+2 (6.10e+0) - | 1.7097e+2 (7.49e+0) - | **6.0801e−1 (1.53e−2)** |
| | 3 | 1.4121e−1 (5.47e−2) - | 4.4372e−1 (1.10e−1) - | 2.3142e−2 (1.64e−4) - | 3.4230e−1 (3.88e−2) - | 1.4038e−1 (2.59e−1) - | 3.9740e−2 (4.97e−2) = | NaN (NaN) | 2.1163e−2 (1.74e−4) - | 2.1303e−2 (2.77e−4) - | 8.4539e−2 (7.37e−2) - | **2.0585e−2 (3.16e−5)** |
| | 5 | 1.1904e−1 (5.13e−2) - | 4.4879e−1 (1.56e−1) - | 6.1253e−2 (3.40e−4) - | 3.6741e−1 (7.46e−2) - | 8.7978e−2 (6.54e−2) - | 6.6678e−2 (3.63e−2) - | NaN (NaN) | 5.6980e−2 (1.04e−3) - | 5.7216e−2 (1.01e−3) - | 8.3301e−2 (4.46e−2) - | **5.2703e−2 (1.24e−5)** |
| DC2_DTLZ1 | 8 | 1.5374e−1 (2.95e−2) - | 5.0946e−1 (1.44e−1) - | 1.2430e−1 (7.27e−4) - | 4.0216e−1 (7.02e−2) - | 1.1450e−1 (2.95e−2) - | 1.0856e−1 (2.59e−2) - | NaN (NaN) | NaN (NaN) | 2.2591e−1 (0.00e+0) = | NaN (NaN) | **9.7173e−2 (4.00e−5)** |
| | 10 | 1.6371e−1 (3.41e−2) - | 6.4506e−1 (1.47e−1) - | 1.4338e−1 (1.04e−1) - | 4.3323e−1 (6.80e−2) - | 1.1603e−1 (1.53e−2) - | 1.1535e−1 (1.98e−2) - | NaN (NaN) | 2.3380e−1 (7.98e−3) - | 2.4261e−1 (0.00e+0) - | 2.1240e−1 (0.00e+0) - | **1.0924e−1 (9.25e−5)** |
| | 15 | 2.6711e−1 (4.90e−2) - | 7.5599e−1 (7.79e−2) - | 2.1687e−1 (4.12e−3) - | 4.6611e−1 (1.28e−1) - | 2.0479e−1 (3.87e−2) - | 2.2509e−1 (5.07e−2) - | NaN (NaN) | NaN (NaN) | NaN (NaN) | NaN (NaN) | **1.8312e−1 (9.08e−4)** |
| | 3 | NaN (NaN) | NaN (NaN) | 5.1407e−1 (1.36e−1) - | 5.3954e−1 (2.34e−1) - | 5.6252e−1 (7.92e−4) - | 5.6290e−1 (3.93e−3) - | NaN (NaN) | **5.5577e−2 (5.55e−4)** - | 1.2880e−1 (1.66e−1) - | 5.6653e−1 (3.75e−3) - | 7.7453e−2 (8.03e−2) |
| | 5 | 5.9767e−1 (1.53e−3) - | 1.3285e+0 (1.12e−1) - | 3.0578e−1 (1.65e−1) - | 5.9047e−1 (1.69e−1) - | 6.3286e−1 (7.44e−2) - | 6.0501e−1 (9.68e−3) - | NaN (NaN) | 5.1390e−1 (2.02e−1) - | 6.2292e−1 (6.59e−2) - | 6.3820e−1 (1.88e−2) - | **1.6504e−1 (1.24e−4)** |
| DC2_DTLZ3 | 8 | 8.1512e−1 (4.09e−2) - | 1.5661e+0 (7.73e−2) - | 4.1644e−1 (1.10e−1) - | 1.1275e+0 (4.48e−1) - | 7.6298e−1 (6.29e−1) - | 6.9270e−1 (1.49e−2) - | NaN (NaN) | 8.2310e−1 (0.00e+0) = | NaN (NaN) | NaN (NaN) | **3.7867e−1 (6.42e−2)** |
| | 10 | 8.4419e−1 (3.12e−2) - | 1.6222e+0 (6.41e−2) - | **4.3850e−1 (4.61e−2)** - | 9.8833e−1 (1.93e−1) - | 7.8365e−1 (1.32e−1) - | 7.9176e−1 (1.05e−2) - | 9.5967e−1 (0.00e+0) - | 8.6445e−1 (0.00e+0) - | 8.4945e−1 (0.00e+0) - | 8.5563e−1 (6.60e−1) - | 4.4248e−1 (3.25e−2) |
| | 15 | 1.0026e+0 (1.98e−2) - | 1.7053e+0 (1.46e−3) - | 6.3739e−1 (1.87e−2) - | 1.4889e+0 (1.88e−1) - | 9.5856e−1 (1.09e−1) - | 9.7664e−1 (6.12e−3) - | 1.1355e+0 (1.38e−1) - | NaN (NaN) | NaN (NaN) | NaN (NaN) | **6.2688e−1 (1.30e−2)** |
| DC3_DTLZ1 | 3 | 1.8985e−1 (1.01e−1) - | 9.2166e+0 (1.21e+1) - | 9.2066e−3 (2.70e−4) - | 2.0076e+0 (1.25e+0) - | 1.5542e−2 (2.93e−2) - | 9.8425e−2 (8.07e−2) - | 3.6156e+0 (2.94e+0) - | **7.2053e−3 (8.90e−5)** + | 7.5990e−3 (3.66e−4) + | 5.7873e−2 (7.50e−2) = | 8.1517e−3 (1.02e−4) |
| | 5 | 1.0734e+0 (5.16e−2) - | 6.0361e+0 (9.03e+0) - | 3.2263e−2 (5.22e−3) - | 1.0480e+0 (5.36e−1) - | 2.8726e−2 (1.52e−2) - | 5.3204e−2 (4.84e−2) - | 8.7646e+0 (7.34e+0) - | **1.8746e−2 (6.67e−4)** + | 1.9470e−2 (6.96e−4) + | 2.6404e−2 (2.91e−2) - | 2.1367e−2 (7.46e−4) |
| DC3_DTLZ3 | 3 | 1.4704e+0 (4.10e−1) - | 1.0056e+0 (9.17e−0) - | 4.8771e−1 (3.30e−2) - | 4.0190e+0 (1.08e+0) - | 6.2978e−1 (3.10e−1) - | 9.3609e−1 (4.52e−1) - | 1.0109e+1 (3.88e+0) - | 3.3601e−2 (2.87e−2) - | 3.4534e−1 (1.98e−1) - | 9.0422e−1 (3.76e−1) - | **2.3069e−2 (4.38e−4)** |
| | 5 | 9.2709e−1 (4.17e−1) - | 3.1616e+0 (1.60e+0) - | 9.7792e−1 (1.36e−2) - | 3.4930e+0 (9.29e−1) - | 3.8017e−1 (2.44e−1) - | 7.3050e−1 (2.84e−1) - | 9.3152e+0 (6.55e+0) - | 2.8038e−1 (2.09e−1) - | 6.4652e−1 (1.49e−1) - | 6.8763e−1 (2.53e−1) - | **7.8344e−2 (1.85e−3)** |
| | 8 | 1.6474e+0 (5.88e−1) - | 2.7607e+0 (5.51e−1) - | 2.3947e−1 (6.44e−2) - | 3.5893e+0 (1.14e+0) - | 8.4779e−1 (6.01e−1) - | 1.2897e+0 (3.60e−1) - | 2.0797e+1 (5.84e+0) - | 5.3433e+1 (1.11e+1) - | 4.6698e+1 (1.08e+1) - | 4.3766e+1 (9.50e+0) - | **1.8735e−1 (2.87e−2)** |
| | 10 | 1.4446e+0 (6.44e−1) - | 2.4439e+0 (5.79e−1) - | 2.0325e−1 (6.70e−2) - | 3.4611e+0 (9.84e−1) - | 5.6973e−1 (3.24e−1) - | 9.5002e−1 (3.52e−1) - | 1.7307e+1 (4.91e+0) - | 6.8489e+1 (9.43e+0) - | 6.0150e+1 (9.55e+0) - | 6.5345e+1 (8.19e+0) - | **1.9546e−1 (8.80e−2)** |
| | 15 | 3.4727e+0 (1.44e+0) - | 7.0692e+0 (5.93e+0) - | 3.1701e+0 (6.91e−1) - | 3.6492e+0 (1.27e+0) - | 7.1588e−1 (2.45e−1) - | 2.3985e+0 (5.99e−1) - | 4.0817e+1 (5.82e+0) - | 1.2969e+2 (4.01e+1) - | 9.9678e+1 (1.42e+1) - | 1.3491e+2 (2.96e+1) - | **1.5429e−1 (1.73e−2)** |
| +/-/= | | 1/42/3 | 0/46/0 | 1/42/4 | 0/47/0 | 0/43/4 | 2/38/7 | 0/35/2 | 6/36/2 | 4/37/3 | 4/36/3 | |

**Notes.**

The best results are in bold.

Ji et al. (2024), *PeerJ Comput. Sci.*, DOI 10.7717/peerj-cs.2102

**Table 5** The IGDp performance values of dCMaOEA-RAE and other schemes on DTLZ benchmark problems.

| Problem | M | NSGAIII | IDBEA | CTAEA | TiGE2 | DCNSGAIII | CMME | ToP | CCMO | DDCMOEA | BiCo | dCMaOEA-RAE |
|---|---|---|---|---|---|---|---|---|---|---|---|---|
| | 3 | 1.5071e−2 (9.98e−4) − | 3.4300e−1 (5.89e−2) − | 1.6368e−2 (2.70e−4) − | 2.0904e−1 (5.56e−2) − | 1.4712e−2 (3.60e−4) = | 1.4676e−2 (1.07e−4) = | NaN (NaN) | 1.5076e−2 (3.51e−4) − | 1.5154e−2 (3.29e−4) − | 1.5405e−2 (4.84e−4) − | **1.4676e−2 (1.47e−4)** |
| | 5 | 3.7117e−2 (1.93e−4) − | 3.9090e−1 (7.37e−2) − | 4.1657e−2 (3.23e−4) − | 2.5660e−1 (6.84e−2) − | 3.7066e−2 (1.26e−4) − | 3.7268e−2 (1.88e−4) − | NaN (NaN) | 3.6398e−2 (2.58e−4) + | 3.6446e−2 (2.59e−4) + | **3.6230e−2 (2.70e−4)** + | 3.6910e−2 (1.82e−4) |
| C1_DTLZ1 | 8 | 6.3453e−2 (2.63e−4) − | 4.2557e−1 (7.79e−2) − | 8.5570e−2 (2.87e−3) − | 3.7210e−1 (7.69e−2) − | 6.4649e−2 (4.69e−3) − | 6.4857e−2 (3.95e−3) − | 3.8029e−1 (0.00e+0) − | 7.3596e−2 (3.21e−3) − | 7.2479e−2 (1.26e−3) − | 6.8557e−2 (8.36e−4) − | **6.2733e−2 (1.71e−4)** |
| | 10 | 6.8922e−2 (4.28e−4) − | 4.1215e−1 (1.01e−1) − | 1.0167e−1 (2.13e−3) − | 4.0702e−1 (3.91e−1) − | 7.0627e−2 (5.44e−3) − | 7.2273e−2 (8.50e−3) − | 2.6124e−1 (8.76e−2) − | 8.1057e−2 (3.47e−3) − | 8.0229e−2 (1.45e−3) − | 7.3970e−2 (8.14e−4) − | **6.8387e−2 (3.60e−4)** |
| | 15 | 1.2163e−1 (4.06e−3) − | 5.2070e−1 (5.15e−2) − | 1.6630e−1 (8.85e−3) − | 4.4015e−1 (3.11e−2) − | 1.1811e−1 (1.25e−2) = | 1.1411e−1 (1.60e−2) + | 2.8383e−1 (8.82e−2) − | 1.4231e−1 (1.06e−2) − | 1.4273e−1 (9.04e−3) − | **1.1198e−1 (5.50e−3)** + | 1.1982e−1 (2.67e−3) |
| | 3 | 4.5493e+0 (4.02e+0) − | 6.7784e+0 (2.85e+0) − | 3.7412e−2 (2.46e−2) − | 6.6561e+0 (2.83e+0) − | 4.3051e+0 (4.03e+0) − | 2.6882e+0 (3.83e+0) − | 2.5388e+0 (3.75e+0) − | 2.3800e−2 (4.84e−4) − | 2.8914e−2 (1.68e−2) − | 1.0915e+0 (2.77e+0) − | **2.2516e−2 (5.70e−5)** |
| | 5 | 5.9199e+0 (5.45e+0) − | 1.0469e+1 (3.49e+0) − | 8.9443e+0 (4.95e+0) − | 8.0714e+0 (5.16e+0) − | 5.2435e+0 (5.65e+0) − | 6.9822e+0 (5.73e+0) − | 4.6065e+0 (5.24e+0) − | 1.0212e−1 (7.25e−2) − | 1.8576e−1 (7.38e−2) − | 1.1577e−1 (2.58e−2) − | **6.2004e−2 (6.08e−5)** |
| C1_DTLZ3 | 8 | 6.3259e+0 (5.80e+0) − | 1.0061e+1 (4.30e+0) − | 8.7839e+0 (4.93e+0) − | 1.1529e+1 (2.79e+0) − | 5.1740e+0 (5.79e+0) − | 8.6753e+0 (5.08e+0) − | 2.1294e+1 (2.24e+0) − | 1.1603e+2 (1.93e+1) − | 1.2365e+2 (1.70e+1) − | 1.0942e+2 (9.85e+0) − | **1.2206e−1 (5.31e−4)** |
| | 10 | 3.3633e+0 (5.58e+0) − | 1.2420e+1 (5.29e+0) − | 1.1533e+1 (5.51e+0) − | 1.1579e+1 (5.90e+0) − | 4.4943e+0 (6.49e+0) − | 1.0965e+1 (6.01e+0) − | 2.2295e+1 (1.91e+0) − | 1.5002e+2 (1.55e+1) − | 1.4579e+2 (1.48e+1) − | 1.3473e+2 (1.57e+1) − | **1.7495e−1 (1.78e−4)** |
| | 15 | 1.3117e+1 (4.04e+0) − | 1.4850e+1 (9.03e−15) − | 9.2640e+0 (6.34e+0) − | 1.4056e+1 (3.53e+0) − | 1.2075e+1 (5.30e+0) − | 1.3848e+1 (2.53e+0) − | 2.2521e+1 (1.52e+0) − | 1.8118e+2 (1.05e+1) − | 1.7849e+2 (7.35e+0) − | 1.7511e+2 (7.15e+0) − | **2.4861e−1 (2.05e−2)** |
| | 3 | 2.1372e−2 (2.37e−4) − | 6.3197e−2 (1.17e−1) − | 2.3856e−2 (8.28e−4) − | 6.1040e−2 (9.35e−3) − | 2.1214e−2 (3.01e−4) − | 2.3871e−2 (4.98e−3) − | 9.8479e−2 (1.66e−1) − | 2.0700e−2 (7.61e−4) = | **2.0353e−2 (7.63e−4)** = | 2.0966e−2 (4.57e−4) − | 2.0447e−2 (3.32e−4) |
| | 5 | 5.5624e−2 (2.75e−4) − | 6.7427e−2 (1.39e−1) − | 5.6095e−2 (6.62e−4) − | 8.3355e−2 (4.45e−3) − | 5.7154e−2 (5.30e−4) − | 5.7630e−2 (7.30e−4) − | 1.4749e−2 (6.41e−3) − | 8.0053e−2 (4.70e−3) − | 7.7944e−2 (4.67e−3) − | 7.6340e−2 (2.56e−3) − | **5.5350e−2 (2.84e−4)** |
| C2_DTLZ2 | 8 | 1.3996e−1 (1.58e−1) − | 9.8003e−1 (1.56e−1) − | **6.0517e−2 (8.35e−4)** + | 6.8137e−2 (1.44e−1) + | 9.4139e−2 (5.59e−2) − | 6.5080e−2 (3.53e−2) + | 6.3935e−1 (1.58e−1) − | 1.8715e−1 (1.89e−2) − | 2.0550e−1 (9.89e−2) − | 1.8829e−1 (1.51e−2) − | 7.4946e−2 (1.53e−3) |
| | 10 | 1.1811e−1 (5.17e−2) − | 9.0742e−1 (1.35e−1) − | 1.4630e−1 (2.42e−2) − | 1.6031e−1 (1.31e−1) − | 1.1885e−1 (5.33e−2) − | 1.0104e−1 (7.09e−2) − | 6.3353e−1 (1.49e−1) − | 2.7440e−1 (2.83e−2) − | 2.7052e−1 (2.89e−2) − | 2.2964e−1 (1.21e−2) − | **9.6858e−2 (1.46e−3)** |
| | 15 | 1.8442e−1 (2.04e−1) = | 1.0801e+0 (1.69e−1) − | **6.1199e−2 (1.48e−3)** + | 7.6436e−2 (1.35e−1) − | 1.0765e−1 (5.17e−2) − | 6.4462e−2 (4.27e−2) + | 8.4881e−1 (4.40e−1) − | 3.1274e−1 (9.18e−2) − | 2.9106e−1 (1.72e−2) − | 2.8147e−1 (3.70e−2) − | 7.3645e−2 (2.69e−2) |
| | 3 | 8.1424e−2 (1.04e−1) − | 3.0240e−1 (2.36e−1) − | 6.1324e−2 (2.73e−3) − | 9.2891e−2 (3.91e−3) − | 7.7509e−2 (8.48e−2) − | 5.6539e−2 (2.04e−2) − | 1.1098e−2 (6.11e−3) − | 9.1339e−2 (1.00e−1) − | 7.0274e−2 (5.96e−2) − | 6.1555e−2 (2.72e−3) − | **5.1679e−2 (5.38e−3)** |
| | 5 | **1.1982e−1 (3.19e−3)** − | 2.6732e−1 (4.22e−2) − | 1.4618e−1 (2.01e−1) − | 1.7275e−1 (8.94e−3) − | 1.5202e−1 (5.97e−2) − | 1.3551e−1 (1.68e−2) − | 3.2848e−1 (1.85e−2) − | 1.9558e−1 (6.87e−3) − | 1.9549e−1 (6.97e−3) − | 2.1764e−1 (8.24e−3) − | 1.2111e−1 (2.55e−3) |
| C3_DTLZ4 | 8 | 2.6476e−1 (5.25e−2) = | 4.8561e−1 (4.53e−2) − | 2.4637e−1 (1.41e−3) − | 2.8048e−1 (2.06e−1) − | 3.6549e−1 (6.55e−2) − | 2.4301e−1 (1.78e−3) − | 9.8181e−1 (2.17e−1) − | 2.1174e+0 (1.66e−2) − | 2.1210e+0 (2.21e−2) − | 8.4204e−1 (7.49e−2) − | **2.4287e−1 (2.33e−3)** |
| | 10 | 2.8542e−1 (1.41e−2) = | 4.4309e−1 (4.32e−2) − | 2.9814e−1 (2.67e−3) − | 3.7084e−1 (1.09e−2) − | 3.5775e−1 (4.38e−2) − | 2.8302e−1 (3.49e−3) − | 1.3896e−1 (2.32e−1) − | 2.2284e+0 (9.38e−3) − | 2.2265e+0 (1.06e−2) − | 9.7925e−1 (9.13e−2) − | **2.8104e−1 (8.99e−4)** |
| | 15 | 5.0147e−1 (9.69e−2) − | 7.6127e−1 (2.14e−1) − | **4.0003e−1 (4.78e−4)** − | 4.5801e−1 (1.04e−2) − | 5.0953e−1 (8.36e−2) − | 4.0187e−1 (5.14e−3) + | 1.9129e−1 (2.14e−1) − | 2.4356e+0 (1.00e−2) − | 2.4366e+0 (1.10e−2) − | 1.4336e+0 (2.32e−1) − | 4.0669e−1 (1.06e−2) |
| | 3 | 1.0188e−2 (3.36e−4) − | 5.9860e−2 (5.73e−2) − | 1.0676e−2 (2.41e−4) − | 6.3877e−1 (4.09e−1) − | 9.7869e−3 (2.38e−4) − | 1.1292e−2 (2.06e−4) − | 3.4279e−2 (5.34e−2) − | **8.6929e−3 (2.44e−4)** + | 9.0721e−3 (4.35e−4) + | 8.8226e−3 (4.32e−4) + | 9.4730e−3 (1.52e−4) |
| | 5 | 2.8071e−2 (1.68e−4) − | 7.5701e−2 (7.52e−2) − | 2.8782e−2 (3.05e−4) − | 3.4897e−1 (1.90e−1) − | 2.8384e−2 (1.59e−4) − | 2.9380e−2 (2.75e−4) − | 1.4320e−2 (2.96e−2) − | 2.9986e−2 (6.57e−4) − | 3.1803e−2 (1.13e−3) − | 3.1071e−2 (7.60e−4) − | **2.7511e−2 (2.06e−4)** |
| DC1_DTLZ1 | 8 | 4.9284e−2 (3.08e−4) − | 1.1482e+0 (9.17e−2) − | 5.5199e−2 (6.80e−4) − | 3.8672e−1 (1.50e−1) − | 4.9115e−2 (3.16e−3) − | 5.1719e−2 (8.18e−3) − | 1.9325e+1 (3.17e+0) − | 9.3636e−1 (7.79e−1) − | 2.3949e+0 (1.72e+0) − | 2.7066e+0 (6.16e+0) − | **4.8935e−2 (4.24e−4)** |
| | 10 | 5.5262e−2 (1.05e−3) − | 9.1598e−1 (6.62e−2) − | 6.6001e−2 (2.20e−3) − | 3.3369e−1 (1.38e−1) − | 5.9687e−2 (9.38e−3) − | **5.0651e−2 (2.46e−3)** + | 8.5308e−1 (2.36e+0) − | 1.5943e+1 (2.66e+1) − | 8.4220e+0 (1.31e+1) − | 6.6049e+1 (4.30e+1) − | 5.4699e−2 (3.58e−4) |
| | 15 | 7.4143e−2 (9.39e−4) = | 3.0707e−1 (1.11e−1) − | 9.0981e−2 (4.49e−3) − | 4.1096e−1 (1.95e−1) − | 7.6779e−2 (4.96e−3) − | 7.6754e−2 (4.30e−2) − | 1.5758e+0 (2.75e−1) − | 1.7559e+2 (3.41e+1) − | 1.9601e+2 (3.27e+1) − | 1.7509e+2 (4.41e+1) − | **7.4060e−2 (4.41e−4)** |
| | 3 | 1.6203e−2 (7.38e−4) − | 7.2363e−2 (6.17e−1) − | 1.6831e−2 (8.74e−4) − | 2.3042e+0 (6.63e−1) − | 1.6513e−2 (7.26e−4) − | 1.7507e−2 (1.22e−1) − | 1.7876e−2 (2.64e+0) − | 1.4190e−2 (6.95e−4) = | 8.0907e−2 (1.60e−2) − | **1.4054e−2 (4.70e−4)** + | 1.4419e−2 (3.03e−4) |
| | 5 | 4.8471e−2 (3.79e−4) − | 6.4856e−1 (4.60e−1) − | 7.6827e−2 (9.02e−3) − | 1.8042e+0 (5.49e−1) − | 4.8692e−2 (6.71e−4) − | 4.9736e−2 (9.46e−4) − | 2.6777e+0 (1.53e+0) − | 8.3787e−2 (5.83e−3) − | 1.7425e−1 (6.03e−2) − | 8.8537e−2 (6.01e−3) − | **4.8206e−2 (6.44e−4)** |
| DC1_DTLZ3 | 8 | 2.4523e−1 (1.13e−1) − | 1.1572e+0 (6.92e−1) − | 2.1490e−1 (4.59e−2) − | 1.5979e+0 (5.21e−1) − | 2.1918e−1 (1.15e−1) − | 2.0597e−1 (4.43e−2) − | 1.9492e+1 (6.72e+0) − | 9.7313e+1 (1.04e+1) − | 9.6549e+1 (9.92e+0) − | 9.2615e+1 (1.15e+1) − | **1.2875e−1 (3.00e−2)** |
| | 10 | 2.3071e−1 (3.88e−2) − | 8.3413e−1 (1.14e−1) − | 1.9902e−1 (1.20e−2) − | 1.2943e+0 (2.85e−1) − | 2.1556e−1 (6.08e−2) − | 2.2052e−1 (1.72e−2) − | 2.1015e+1 (5.80e+0) − | 1.2820e+2 (1.96e+1) − | 1.2782e+2 (2.22e+1) − | 1.2017e+2 (1.64e+1) − | **1.8044e−1 (1.81e−2)** |
| | 15 | 3.6545e−1 (8.55e−2) − | 1.0587e+0 (4.81e−1) − | 4.0432e−1 (1.08e−1) − | 1.0242e+0 (1.45e−1) − | 3.9700e−1 (1.01e−1) − | 2.9844e−1 (1.97e−2) − | 1.9525e+1 (6.39e+0) − | 1.8314e+2 (7.12e+0) − | 1.8222e+2 (6.10e+0) − | 1.7097e+2 (7.49e+0) − | **2.4723e−1 (2.18e−2)** |
| | 3 | 1.4040e−1 (5.65e−2) − | 4.1064e−1 (9.82e−2) − | 1.6410e−2 (3.64e−4) − | 2.5655e−1 (3.47e−1) − | 1.2849e−1 (2.41e−1) − | 3.4643e−2 (5.17e−2) − | NaN (NaN) | 1.5119e−2 (4.08e−4) − | 1.5513e−2 (7.99e−4) − | 8.1403e−2 (7.66e−2) − | **1.4738e−2 (1.75e−4)** |
| | 5 | 1.1338e−1 (5.85e−2) − | 4.0660e−1 (1.53e−1) − | 4.3221e−2 (2.56e−4) − | 2.9019e−1 (7.33e−2) − | 7.5801e−2 (6.89e−2) − | 5.3527e−2 (4.14e−2) − | NaN (NaN) | 4.3852e−2 (1.57e−1) − | 4.4832e−2 (1.69e−3) − | 7.6669e−2 (4.77e−2) − | **3.7589e−2 (3.96e−5)** |
| DC2_DTLZ1 | 8 | 1.4102e−1 (3.95e−2) − | 4.5940e−1 (1.47e−1) − | 9.0082e−2 (9.09e−4) − | 3.2493e−1 (7.34e−2) − | 8.5652e−2 (4.06e−2) = | 7.9495e−2 (3.59e−2) − | NaN (NaN) | NaN (NaN) | 2.1694e−1 (0.00e+0) − | NaN (NaN) | **6.3804e−2 (1.49e−4)** |
| | 10 | 1.4286e−1 (4.52e−2) − | 6.0198e−1 (1.52e−1) − | 1.0586e−1 (1.10e−3) − | 3.6727e−1 (7.10e−2) − | 7.5415e−2 (2.04e−2) − | 7.8057e−2 (2.87e−2) − | NaN (NaN) | 2.0939e−1 (2.52e−2) − | 2.3065e−1 (0.00e+0) − | 2.0059e−1 (0.00e+0) − | **6.8932e−2 (1.00e−4)** |
| | 15 | 2.3120e−1 (6.31e−2) − | 7.1874e−1 (8.29e−2) − | 1.7049e−1 (5.05e−3) − | 4.0874e−1 (1.37e−1) − | 1.5173e−1 (5.11e−2) − | 1.7862e−1 (6.78e−2) = | NaN (NaN) | NaN (NaN) | NaN (NaN) | NaN (NaN) | **1.2195e−1 (1.12e−3)** |

Ji et al. (2024), *PeerJ Comput. Sci.*, DOI 10.7717/peerj-cs.2102

**Table 5** (*continued*)

| Problem | M | NSGAIII | IDBEA | CTAEA | TiGE2 | DCNSGAIII | CMME | ToP | CCMO | DDCMOEA | BiCo | dCMaOEA-RAE |
|---|---|---|---|---|---|---|---|---|---|---|---|---|
| | 3 | NaN (NaN) | NaN (NaN) | 5.0295e−1 (1.50e−1) - | 4.1451e−1 (3.39e−1) - | 5.6204e−1 (7.77e−4) - | 5.6240e−1 (3.76e−3) - | NaN (NaN) | **2.5172e−2 (1.69e−3)** + | 9.9415e−2 (1.68e−1) - | 5.6567e−1 (3.68e−3) - | 4.8172e−2 (8.28e−2) |
| | 5 | 5.8544e−1 (1.47e−3) - | 1.1968e+0 (6.67e−2) - | 1.9624e−1 (1.80e−1) - | 3.7730e−1 (2.49e−1) - | 6.0997e−1 (4.83e−2) - | 5.9176e−1 (8.30e−3) - | NaN (NaN) | 4.8620e−1 (2.30e−1) - | 6.0770e−1 (8.59e−2) - | 6.2759e−1 (1.86e−2) - | **6.2047e−2 (6.82e−5)** |
| DC2_DTLZ3 | 8 | 7.1946e−1 (2.62e−2) - | 1.3449e+0 (5.88e−2) - | 1.9755e−1 (1.37e−1) = | 8.3837e−1 (4.89e−1) - | 6.9204e−1 (3.97e−2) - | 6.4366e−1 (4.78e−3) - | NaN (NaN) | 7.8606e−1 (0.00e+0) = | NaN (NaN) | NaN (NaN) | **1.5498e−1 (3.21e−2)** |
| | 10 | 7.3825e−1 (2.74e−2) - | 1.3606e+0 (2.45e−2) - | **1.6340e−1 (4.04e−2)** + | 6.6764e−1 (1.92e−1) - | 6.6327e−1 (1.71e−1) - | 6.9057e−1 (4.48e−3) - | 7.5380e−1 (0.00e+0) = | 8.1869e−1 (0.00e+0) = | 8.0284e−1 (0.00e+0) = | 8.1081e−1 (9.59e−3) - | 1.8389e−1 (1.40e−2) |
| | 15 | 8.2436e−1 (2.18e−2) - | 1.4065e+0 (1.78e−3) - | **2.5376e−1 (1.83e−2)** + | 1.2206e+0 (2.03e−1) - | 7.5877e−1 (1.70e−1) - | 7.8952e−1 (3.78e−3) - | 9.6822e−1 (1.82e−1) - | NaN (NaN) | NaN (NaN) | NaN (NaN) | 2.5884e−1 (1.75e−2) |
| DC3_DTLZ1 | 3 | 1.8963e−1 (1.02e−0) - | 9.2109e−1 (1.21e+1) - | 6.4401e−3 (2.25e−4) - | 1.9866e+0 (1.26e+0) - | 1.2833e−2 (2.98e−2) - | 9.6676e−2 (8.24e−2) - | 3.6118e+0 (2.94e+0) - | **5.2763e−3 (2.52e−4)** + | 6.2000e−3 (7.51e−4) - | 5.6977e−2 (7.56e−2) - | 5.7532e−3 (1.16e−4) |
| | 5 | 1.0550e−1 (5.47e−2) - | 6.0264e+0 (9.03e+0) - | 2.2297e−2 (3.94e−2) - | 1.0259e+0 (5.49e−1) - | 1.8314e−2 (8.07e−3) - | 4.6251e−2 (5.19e−2) - | 8.7615e+0 (7.34e+0) - | **1.2866e−2 (5.80e−4)** + | 1.5066e−2 (1.16e−3) - | 2.1411e−2 (3.04e−2) - | 1.4656e−2 (7.67e−4) |
| | 3 | 1.4702e+0 (4.10e−1) - | 1.0041e+1 (9.17e+0) - | 2.5030e−2 (2.75e−2) - | 3.9622e+0 (1.10e+0) - | 6.2874e−1 (3.12e−1) - | 9.3571e−1 (4.52e−1) - | 1.0102e+1 (3.89e+0) - | 2.2802e−2 (3.19e−2) - | 3.1587e−1 (2.16e−1) - | 9.0262e−1 (3.76e−1) - | **8.3682e−3 (5.55e−4)** |
| | 5 | 9.2330e−1 (4.18e−1) - | 3.1029e+0 (1.62e+0) - | 3.6526e−2 (9.16e−3) - | 3.4067e+0 (9.54e−1) - | 3.5355e−1 (2.70e−1) - | 7.2354e−1 (2.86e−1) - | 9.2965e+0 (6.57e+0) - | 2.4476e−1 (2.24e−1) - | 6.3199e−1 (1.63e−1) - | 6.8480e−1 (2.53e−1) - | **2.9814e−2 (1.26e−3)** |
| DC3_DTLZ3 | 8 | 1.6388e+0 (5.91e−1) - | 2.6413e+0 (5.86e−1) - | 1.1580e−1 (4.07e−2) - | 3.5271e+0 (1.17e+0) - | 8.0235e−1 (6.24e−1) - | 1.2803e+0 (3.64e−1) - | 2.0791e+1 (5.84e+0) - | 5.3431e+1 (1.11e+1) - | 4.6694e+1 (1.08e+1) - | 4.3764e+1 (9.50e+0) - | **9.0387e−2 (2.96e−2)** |
| | 10 | 1.4288e+0 (6.42e−1) - | 2.3043e+0 (6.11e−1) - | **1.0824e−1 (3.12e−2)** + | 3.3303e+0 (1.02e+0) - | 4.8189e−1 (3.09e−1) - | 9.2788e−1 (3.60e−1) - | 1.7295e+1 (4.92e+0) - | 6.8487e+1 (9.43e+0) - | 6.0148e+1 (9.55e+0) - | 6.5343e+1 (8.19e+0) - | 1.2956e−1 (4.21e−2) |
| | 15 | 3.4292e+0 (1.45e+0) - | 6.9724e+0 (5.97e+0) - | 3.1329e+0 (6.98e−1) - | 3.5671e+0 (1.27e+0) - | 6.7397e−1 (2.49e−1) - | 2.3755e+0 (5.97e−1) - | 4.0806e+1 (5.82e+0) - | 1.2969e+2 (4.01e+1) - | 9.9677e+1 (1.42e+1) - | 1.3491e+2 (2.96e+1) - | **5.9004e−2 (2.50e−2)** |
| +/-/= | | 0/41/5 | 0/46/0 | 6/40/1 | 1/46/0 | 0/42/5 | 5/37/5 | 0/35/2 | 5/35/4 | 2/37/5 | 4/38/1 | |

**Notes.**

The best results are in bold.

**Table 6** The HV performance values of dCMaOEA-RAE and other schemes on DTLZ benchmark problems.

| Problem | M | NSGAIII | IDBEA | CTAEA | TiGE2 | DCNSGAIII | CMME | ToP | CCMO | DDCMOEA | BiCo | dCMaOEA-RAE |
|---|---|---|---|---|---|---|---|---|---|---|---|---|
| | 3 | 8.3079e−1 (9.63e−3) - | 8.1384e−2 (6.91e−2) - | 8.3313e−1 (4.44e−3) - | 3.0228e−1 (9.86e−2) - | 8.3546e−1 (5.14e−3) = | 8.3557e−1 (3.30e−3) = | NaN (NaN) | 8.3238e−1 (6.38e−3) - | 8.3264e−1 (6.65e−3) - | 8.3142e−1 (5.85e−3) - | **8.3668e−1 (3.15e−3)** |
| | 5 | 9.7357e−1 (4.68e−3) = | 9.7600e−2 (9.50e−2) - | 9.7664e−1 (1.68e−3) + | 4.1091e−1 (1.74e−1) - | **9.7730e−1 (2.65e−3)** + | 9.7655e−1 (4.20e−3) + | NaN (NaN) | 9.7420e−1 (8.26e−3) - | 9.7612e−1 (3.08e−3) + | 9.7372e−1 (3.90e−3) = | 9.7429e−1 (3.07e−3) |
| C1_DTLZ1 | 8 | 9.7893e−1 (1.34e−2) - | 1.0138e−1 (8.57e−2) - | 9.9099e−1 (7.62e−3) - | 2.0892e−1 (1.51e−1) - | 9.8851e−1 (9.15e−3) - | 9.9064e−1 (9.05e−3) + | 1.9269e−1 (0.00e+0) - | 9.7697e−1 (1.68e−2) - | 9.7169e−1 (2.73e−2) - | **9.9304e−1 (2.14e−3)** + | 9.8708e−1 (6.19e−3) |
| | 10 | 9.9386e−1 (8.00e−3) = | 1.3252e−1 (1.17e−1) - | **9.9857e−1 (1.57e−3)** + | 1.6268e−1 (5.70e−2) - | 9.9641e−1 (3.67e−2) - | 9.9648e−1 (3.24e−3) - | 4.7593e−2 (2.01e−2) - | 9.8585e−1 (1.66e−2) - | 9.8821e−1 (1.21e−2) - | 9.9726e−1 (1.44e−3) = | 9.9628e−1 (2.35e−3) |
| | 15 | 9.8987e−1 (1.51e−2) - | 3.7705e−2 (5.21e−2) - | 9.9672e−1 (4.40e−3) + | 1.3674e−1 (5.46e−2) - | **9.9904e−1 (1.68e−3)** + | 9.9473e−1 (6.73e−3) + | 4.5187e−1 (1.80e−1) - | 9.7278e−1 (2.57e−2) - | 9.7465e−1 (3.16e−3) - | 9.9397e−1 (6.78e−3) = | 9.9443e−1 (3.17e−3) |
| | 3 | 2.4047e−1 (2.80e−1) - | 7.0244e−2 (1.83e−1) - | 5.3015e−1 (4.06e−2) - | 0.0000e+0 (0.00e+0) - | 2.4159e−1 (2.81e−1) - | 3.7197e−1 (2.68e−1) - | 2.8998e−1 (2.34e−1) - | 5.5833e−1 (8.94e−4) - | 5.4974e−1 (2.74e−2) - | 4.8312e−1 (1.93e−1) - | **5.5951e−1 (9.75e−5)** |
| | 5 | 2.8929e−1 (3.88e−1) - | 6.0718e−2 (2.07e−1) - | 1.5756e−1 (2.94e−1) - | 0.0000e+0 (0.00e+0) - | 3.4392e−1 (4.00e−1) - | 3.1487e−1 (3.93e−1) - | 3.1292e−2 (5.77e−2) - | 7.6360e−1 (8.34e−3) - | 6.2188e−1 (1.29e−1) - | 7.4071e−1 (4.20e−2) - | **8.1268e−1 (3.91e−4)** |
| C1_DTLZ3 | 8 | 3.8594e−1 (4.52e−1) - | 6.7505e−2 (2.33e−1) - | 9.3339e−1 (2.14e−1) - | 0.0000e+0 (0.00e+0) - | 4.6273e−1 (4.44e−1) - | 1.5470e−1 (3.18e−1) - | 0.0000e+0 (0.00e+0) - | 0.0000e+0 (0.00e+0) - | 0.0000e+0 (0.00e+0) - | 0.0000e+0 (0.00e+0) - | **9.2377e−1 (4.09e−4)** |
| | 10 | 5.7396e−1 (4.76e−1) - | 1.5151e−2 (3.45e−2) - | 7.6375e−2 (2.12e−1) - | 0.0000e+0 (0.00e+0) - | 5.7572e−1 (4.72e−1) - | 1.9406e−1 (3.58e−1) - | 0.0000e+0 (0.00e+0) - | 0.0000e+0 (0.00e+0) - | 0.0000e+0 (0.00e+0) - | 0.0000e+0 (0.00e+0) - | **9.6960e−1 (2.38e−4)** |
| | 15 | 6.2260e−2 (2.37e−1) - | 0.0000e+0 (0.00e+0) - | 3.6518e−2 (1.39e−1) - | 0.0000e+0 (0.00e+0) - | 1.1863e−1 (3.05e−1) - | 1.6573e−2 (9.08e−2) - | 0.0000e+0 (0.00e+0) - | 0.0000e+0 (0.00e+0) - | 0.0000e+0 (0.00e+0) - | 0.0000e+0 (0.00e+0) - | **9.7806e−1 (2.12e−2)** |
| | 3 | 5.0596e−1 (1.55e−3) - | 4.4350e−2 (4.23e−1) - | 5.0400e−1 (2.50e−3) - | 4.0805e−1 (2.15e−1) - | 5.0219e−1 (2.25e−1) - | 5.0292e−1 (2.45e−2) - | 4.1234e−1 (1.30e−1) - | 5.1319e−1 (1.58e−1) - | **5.1392e−1 (1.50e−3)** + | 5.1238e−1 (1.60e−3) + | 5.0966e−1 (1.87e−3) |
| | 5 | 7.5435e−1 (1.16e−3) - | 6.8357e−2 (6.12e−2) - | 7.4050e−1 (1.97e−3) - | 6.5522e−1 (1.43e−2) - | 7.4789e−1 (2.07e−1) - | 7.4383e−1 (1.81e−3) - | 5.2333e−1 (1.83e−2) - | 7.3029e−1 (4.57e−2) - | 7.3504e−1 (4.48e−3) - | 7.3563e−1 (2.98e−3) - | **7.5518e−1 (7.24e−4)** |
| C2_DTLZ2 | 8 | 7.7736e−1 (1.61e−1) - | 2.9518e−2 (4.41e−2) - | 7.9456e−1 (6.73e−3) - | 7.8054e−1 (1.02e−2) - | 8.2090e−1 (4.81e−2) - | 7.9791e−1 (2.33e−2) - | 2.4464e−1 (9.70e−2) - | 7.7460e−1 (1.45e−2) - | 7.6116e−1 (1.08e−1) - | 7.7791e−1 (1.16e−1) - | **8.4337e−1 (1.23e−3)** |
| | 10 | 8.8403e−1 (2.61e−2) - | 5.3786e−2 (5.15e−2) - | 8.7656e−1 (4.53e−3) - | 8.7053e−1 (7.11e−3) - | 8.8440e−1 (2.66e−2) - | 8.8678e−1 (7.23e−4) - | 2.7555e−1 (1.18e−2) - | 8.3972e−1 (9.91e−3) - | 8.4168e−1 (7.36e−3) - | 8.3985e−1 (7.27e−3) - | **9.0091e−1 (1.21e−3)** |
| | 15 | 8.5497e−1 (2.23e−1) - | 2.6259e−2 (3.08e−2) - | 9.3111e−1 (4.11e−1) - | 8.6271e−1 (2.03e−2) - | 9.3501e−1 (2.49e−2) - | 9.4281e−1 (5.05e−2) - | 1.7040e−1 (5.61e−2) - | 8.6617e−1 (1.20e−1) - | 8.9485e−1 (9.76e−3) - | 9.0964e−1 (6.90e−3) - | **9.5419e−1 (1.36e−2)** |
| | 3 | 7.6826e−1 (8.07e−2) - | 5.8664e−1 (1.48e−1) - | 7.8483e−1 (1.75e−3) - | 7.6185e−1 (3.01e−3) - | 7.7262e−1 (6.57e−2) - | 7.8802e−1 (1.42e−1) - | 7.5108e−1 (4.54e−3) - | 7.6293e−1 (7.65e−2) - | 7.7913e−1 (4.57e−2) - | 7.8602e−1 (1.77e−3) - | **7.9176e−1 (3.85e−3)** |
| | 5 | **9.6444e−1 (6.33e−4)** + | 8.9923e−1 (2.56e−2) - | 9.5706e−1 (6.58e−4) - | 9.4841e−1 (2.77e−3) - | 9.5397e−1 (2.30e−2) - | 9.5980e−1 (5.37e−4) - | 8.9026e−1 (8.92e−3) - | 9.4627e−1 (2.02e−2) - | 9.4633e−1 (2.09e−3) - | 9.3852e−1 (3.04e−3) - | 9.6418e−1 (4.87e−4) |
| C3_DTLZ4 | 8 | 9.9365e−1 (5.77e−3) = | 9.4139e−1 (2.46e−2) - | 9.9435e−1 (1.28e−3) - | 9.9226e−1 (2.35e−2) - | 9.8274e−1 (8.96e−3) - | 9.9470e−1 (1.51e−2) - | 6.0447e−1 (1.91e−2) - | 4.6487e−2 (4.96e−2) - | 4.7454e−2 (5.64e−2) - | 6.8927e−1 (5.55e−2) - | **9.9592e−1 (1.21e−4)** |
| | 10 | 9.9930e−1 (3.41e−4) = | 8.8939e−1 (4.92e−3) - | 9.9903e−1 (3.95e−5) - | 9.9750e−1 (5.26e−4) - | 9.9740e−1 (1.61e−1) - | 9.9917e−1 (7.54e−5) - | 3.8396e−1 (1.71e−1) - | 8.5281e−2 (4.30e−1) - | 8.6897e−2 (3.72e−1) - | 6.5068e−1 (6.80e−2) - | **9.9941e−1 (2.50e−5)** |
| | 15 | 9.9907e−1 (1.30e−3) - | 8.9733e−1 (1.89e−1) - | **9.9998e−1 (3.35e−6)** + | 9.9965e−1 (1.01e−4) - | 9.9887e−1 (2.12e−1) - | 9.9997e−1 (1.70e−5) - | 1.2437e−1 (7.96e−2) - | 6.1290e−2 (5.04e−1) - | 5.9072e−2 (4.18e−3) - | 2.9820e−1 (8.53e−2) - | 9.9996e−1 (3.26e−5) |
| | 3 | 6.2853e−1 (1.80e−3) - | 4.7633e−1 (1.30e−1) - | 6.2711e−1 (8.56e−3) - | 4.4992e−2 (7.19e−2) - | 6.2523e−1 (1.56e−3) - | 6.2462e−1 (2.53e−3) - | 5.3679e−1 (1.04e−1) - | **6.3120e−1 (1.03e−3)** = | 6.3008e−1 (1.30e−3) - | 6.3007e−1 (1.70e−3) - | 6.3080e−1 (7.35e−4) |
| | 5 | 7.7271e−1 (1.02e−3) - | 6.5021e−1 (1.86e−1) - | 7.7004e−1 (6.49e−4) - | 1.0812e−1 (5.02e−2) - | 7.6993e−1 (8.79e−4) - | 7.7093e−1 (8.96e−4) - | 2.3086e−1 (2.55e−1) - | 7.7030e−1 (8.49e−4) - | 7.6717e−1 (2.62e−2) - | 7.6534e−1 (1.59e−3) - | **7.7335e−1 (2.79e−4)** |
| DC1_DTLZ1 | 8 | 7.9422e−1 (1.97e−4) - | 6.4863e−1 (2.37e−1) - | 7.9142e−1 (1.56e−3) - | 1.0051e−1 (5.52e−2) - | 7.9418e−1 (3.63e−3) - | 7.9443e−1 (3.26e−3) = | 1.2615e−1 (1.33e−1) - | 5.7474e−2 (1.35e−1) - | 1.9399e−2 (7.75e−2) - | 2.2937e−2 (7.98e−2) - | **7.9450e−1 (2.41e−4)** |
| | 10 | 7.9615e−1 (1.87e−4) - | 7.2697e−1 (1.82e−1) - | 7.9611e−1 (8.32e−5) - | 1.0508e−1 (2.29e−2) - | 7.9096e−1 (1.80e−2) - | 7.9620e−1 (1.07e−3) - | 2.0599e−1 (1.35e−1) - | 0.0000e+0 (0.00e+0) - | 0.0000e+0 (0.00e+0) - | 0.0000e+0 (0.00e+0) - | **7.9640e−1 (6.24e−5)** |
| | 15 | 7.9780e−1 (5.25e−5) - | 2.1553e−2 (1.25e−1) - | 7.9690e−1 (6.67e−4) - | 9.7776e−2 (3.14e−2) - | 7.9701e−1 (8.79e−4) - | 7.9692e−1 (1.77e−2) - | 1.3018e−1 (1.43e−1) - | 0.0000e+0 (0.00e+0) - | 0.0000e+0 (0.00e+0) - | 0.0000e+0 (0.00e+0) - | **7.9782e−1 (1.26e−5)** |
| | 3 | 4.6770e−1 (1.21e−3) - | 4.4299e−2 (9.85e−2) - | 4.5969e−1 (2.38e−3) - | 0.0000e+0 (0.00e+0) - | 4.6689e−1 (1.75e−3) - | 4.6456e−1 (1.75e−3) - | 1.2582e−1 (1.55e−1) - | 4.7178e−1 (1.19e−3) + | **4.8450e−1 (3.04e−2)** + | 4.7192e−1 (9.03e−4) + | 4.7020e−1 (6.42e−4) |
| | 5 | 7.6343e−1 (8.32e−4) - | 1.5999e−1 (1.91e−1) - | 6.9317e−1 (2.77e−2) - | 0.0000e+0 (0.00e+0) - | 7.6326e−1 (1.11e−3) - | 7.6066e−1 (1.70e−2) - | 6.7052e−3 (2.15e−2) - | 7.3878e−1 (6.61e−3) - | 6.3025e−1 (1.21e−1) - | 7.3260e−1 (7.72e−3) - | **7.6433e−1 (6.39e−4)** |
| DC1_DTLZ3 | 8 | 6.7891e−1 (2.12e−1) - | 3.8186e−2 (5.80e−2) - | 7.2227e−1 (9.72e−2) - | 0.0000e+0 (0.00e+0) - | 7.2117e−1 (2.11e−1) - | 7.2086e−1 (1.15e−1) - | 2.7222e−3 (1.49e−2) - | 0.0000e+0 (0.00e+0) - | 0.0000e+0 (0.00e+0) - | 0.0000e+0 (0.00e+0) - | **8.8759e−1 (2.46e−2)** |
| | 10 | 8.6427e−1 (6.05e−2) - | 1.1858e−2 (7.32e−2) - | 9.0575e−1 (1.34e−2) - | 4.3598e−5 (1.67e−4) - | 8.8727e−1 (8.32e−2) - | 8.5660e−1 (3.52e−2) - | 1.6414e−3 (8.99e−3) - | 0.0000e+0 (0.00e+0) - | 0.0000e+0 (0.00e+0) - | 0.0000e+0 (0.00e+0) - | **9.3396e−1 (2.37e−2)** |
| | 15 | 7.7122e−1 (1.99e−1) - | 4.0233e−2 (2.95e−2) - | 5.3964e−1 (2.87e−1) - | 1.3883e−2 (3.04e−2) - | 6.9292e−1 (2.23e−1) - | 8.7544e−1 (5.45e−2) - | 8.9910e−4 (4.92e−3) - | 0.0000e+0 (0.00e+0) - | 0.0000e+0 (0.00e+0) - | 0.0000e+0 (0.00e+0) - | **9.6829e−1 (2.64e−2)** |

**Table 6** (*continued*)

| Problem | M | NSGAIII | IDBEA | CTAEA | TiGE2 | DCNSGAIII | CMME | ToP | CCMO | DDCMOEA | BiCo | dCMaOEA-RAE |
|---|---|---|---|---|---|---|---|---|---|---|---|---|
| | 3 | 5.3565e−1 (1.38e−1) - | 4.1253e−2 (4.80e−2) - | 8.3793e−1 (8.03e−4) - | 2.2592e−1 (6.28e−2) - | 6.6866e−1 (3.24e−1) - | 7.9298e−1 (1.26e−1) = | NaN (NaN) | 8.4021e−1 (9.28e−4) - | 8.3918e−1 (1.91e−3) - | 6.7809e−1 (1.87e−1) - | **8.4132e−1 (3.90e−4)** |
| | 5 | 8.8733e−1 (7.15e−2) - | 9.4441e−2 (8.46e−2) - | 9.7703e−1 (2.35e−4) - | 3.1664e−1 (1.98e−1) - | 9.1189e−1 (1.52e−1) = | 9.6040e−1 (5.05e−2) - | NaN (NaN) | 9.7553e−1 (1.02e−3) - | 9.7512e−1 (9.53e−4) - | 9.3350e−1 (7.07e−2) - | **9.7981e−1 (1.25e−4)** |
| DC2_DTLZ1 | 8 | 9.4308e−1 (2.81e−2) - | 8.6927e−2 (8.75e−2) - | 9.9630e−1 (1.49e−4) - | 3.1972e−1 (1.85e−1) - | 9.8299e−1 (2.83e−2) - | 9.8646e−1 (2.53e−2) - | NaN (NaN) | NaN (NaN) | 8.9944e−1 (0.00e+0) = | NaN (NaN) | **9.9760e−1 (5.02e−5)** |
| | 10 | 9.8137e−1 (1.18e−2) - | 3.5186e−2 (5.71e−2) - | 9.9945e−1 (3.38e−5) - | 2.5602e−1 (1.71e−1) - | 9.9853e−1 (4.84e−3) - | 9.9742e−1 (6.90e−3) - | NaN (NaN) | 9.6718e−1 (2.83e−2) - | 9.6177e−1 (0.00e+0) - | 9.6774e−1 (0.00e+0) - | **9.9968e−1 (1.69e−5)** |
| | 15 | 9.7648e−1 (1.39e−2) - | 6.2107e−3 (2.28e−2) - | 9.9983e−1 (4.90e−5) - | 2.5592e−1 (2.64e−1) - | 9.9098e−1 (1.48e−2) - | 9.8683e−1 (1.42e−2) - | NaN (NaN) | NaN (NaN) | NaN (NaN) | NaN (NaN) | **9.9992e−1 (7.81e−6)** |
| | 3 | NaN (NaN) | NaN (NaN) | 5.5219e−2 (1.37e−1) - | 1.4459e−1 (1.67e−1) - | 7.7810e−3 (2.66e−4) - | 7.2387e−3 (1.58e−3) - | NaN (NaN) | **5.5615e−1 (2.58e−3)** + | 4.6617e−1 (1.84e−1) - | 1.3183e−2 (7.65e−4) - | 5.2320e−1 (1.03e−1) |
| | 5 | 7.1128e−2 (1.04e−3) - | 2.6517e−4 (5.24e−4) - | 5.9045e−1 (2.82e−1) - | 3.2567e−1 (2.40e−1) - | 6.1863e−2 (1.41e−2) - | 5.7198e−2 (1.80e−2) - | NaN (NaN) | 2.3767e−1 (3.15e−1) - | 7.1011e−2 (1.08e−1) - | 5.1734e−2 (1.04e−2) - | **8.1290e−1 (5.15e−4)** |
| DC2_DTLZ3 | 8 | 9.4008e−2 (2.98e−2) - | 7.8746e−4 (1.68e−3) - | 8.1248e−1 (2.46e−1) = | 2.5644e−1 (3.05e−1) - | 1.1812e−1 (3.80e−2) - | 1.5532e−1 (4.23e−2) - | NaN (NaN) | 6.0427e−2 (0.00e+0) = | NaN (NaN) | NaN (NaN) | **8.9720e−1 (2.79e−2)** |
| | 10 | 1.9009e−1 (3.83e−2) - | 5.1470e−4 (1.20e−3) - | 9.5124e−1 (9.40e−2) - | 2.7150e−1 (2.02e−1) - | 2.7644e−1 (2.20e−1) - | 1.3179e−1 (1.11e−1) - | 5.1449e−2 (0.00e+0) = | 1.2294e−1 (0.00e+0) - | 1.2314e−1 (0.00e+0) - | 9.4961e−2 (4.02e−3) - | **9.5968e−1 (1.77e−2)** |
| | 15 | 2.0471e−1 (1.55e−2) - | 0.0000e+0 (0.00e+0) - | 9.5590e−1 (2.43e−2) - | 1.4640e−2 (4.07e−2) - | 1.1581e−1 (2.93e−1) - | 0.0000e+0 (0.00e+0) - | 7.6188e−2 (8.71e−2) - | NaN (NaN) | NaN (NaN) | NaN (NaN) | **9.7109e−1 (1.80e−2)** |
| DC3_DTLZ1 | 3 | 1.0301e−1 (1.18e−1) - | 5.0216e−2 (2.75e−3) - | 5.2385e−1 (3.52e−1) - | 4.7724e−3 (1.82e−2) - | 5.1341e−1 (7.88e−2) - | 2.7861e−1 (2.20e−1) - | 1.8447e−3 (7.04e−3) - | **5.3458e−1 (1.44e−3)** + | 5.3150e−1 (2.74e−3) - | 3.9225e−1 (1.99e−1) - | 5.3403e−1 (1.29e−3) |
| | 5 | 3.3519e−2 (5.65e−2) - | 0.0000e+0 (0.00e+0) - | 8.8720e−2 (1.34e−2) - | 0.0000e+0 (0.00e+0) - | 1.2609e−2 (7.73e−3) - | 9.1221e−2 (5.15e−2) - | 1.7315e−3 (9.48e−3) - | **1.4590e−1 (1.19e−2)** + | 1.3700e−1 (4.01e−3) - | 1.3026e−1 (3.55e−2) - | 1.3680e−1 (1.54e−3) |
| | 3 | 0.0000e+0 (0.00e+0) - | 0.0000e+0 (0.00e+0) - | 3.2375e−1 (5.53e−2) - | 0.0000e+0 (0.00e+0) - | 1.2013e−2 (6.58e−2) - | 0.0000e+0 (0.00e+0) - | 4.9475e−3 (2.71e−2) - | 3.4162e−1 (5.41e−2) - | 2.0874e−1 (1.74e−1) - | 0.0000e+0 (0.00e+0) - | **3.6432e−1 (1.79e−2)** |
| | 5 | 0.0000e+0 (0.00e+0) - | 0.0000e+0 (0.00e+0) - | 5.2809e−1 (2.60e−2) - | 0.0000e+0 (0.00e+0) - | 2.2979e−1 (2.86e−1) - | 0.0000e+0 (0.00e+0) - | 3.6063e−3 (1.98e−2) - | 2.9958e−1 (2.52e−1) - | 3.3183e−2 (1.27e−1) - | 0.0000e+0 (0.00e+0) - | **5.8852e−1 (2.50e−3)** |
| DC3_DTLZ3 | 8 | 0.0000e+0 (0.00e+0) - | 0.0000e+0 (0.00e+0) - | 4.9611e−2 (6.96e−2) - | 0.0000e+0 (0.00e+0) - | 1.3614e−1 (2.78e−1) - | 0.0000e+0 (0.00e+0) - | 0.0000e+0 (0.00e+0) - | 0.0000e+0 (0.00e+0) - | 0.0000e+0 (0.00e+0) - | 0.0000e+0 (0.00e+0) - | **7.0329e−1 (3.00e−2)** |
| | 10 | 0.0000e+0 (0.00e+0) - | 0.0000e+0 (0.00e+0) - | 6.5133e−1 (4.80e−2) - | 0.0000e+0 (0.00e+0) - | 3.1313e−1 (3.60e−1) - | 0.0000e+0 (0.00e+0) - | 0.0000e+0 (0.00e+0) - | 0.0000e+0 (0.00e+0) - | 0.0000e+0 (0.00e+0) - | 0.0000e+0 (0.00e+0) - | **7.6613e−1 (2.81e−2)** |
| | 15 | 0.0000e+0 (0.00e+0) - | 0.0000e+0 (0.00e+0) - | 0.0000e+0 (0.00e+0) - | 0.0000e+0 (0.00e+0) - | 3.3276e−3 (1.33e−2) - | 0.0000e+0 (0.00e+0) - | 0.0000e+0 (0.00e+0) - | 0.0000e+0 (0.00e+0) - | 0.0000e+0 (0.00e+0) - | 0.0000e+0 (0.00e+0) - | **2.9572e−2 (3.65e−2)** |
| +/-/= | | 0/40/6 | 0/46/0 | 5/40/2 | 0/47/0 | 2/39/6 | 4/34/9 | 0/35/2 | 5/35/4 | 3/37/4 | 3/35/5 | |

**Notes.**

The best results are in bold.

**Table 7** The Friedman ranking of the 11 algorithms in the MW and CF test problems.

| | MW | | | CF | | |
|---|---|---|---|---|---|---|
| | Igd ranking | Igdp ranking | Hv ranking | Igd ranking | Igdp ranking | Hv ranking |
| NSGAIII | 3.58 | 3.41 | 3.66 | 4.25 | 4.40 | 4.78 |
| IDBEA | 10.16 | 10.19 | 9.96 | 8.33 | 8.02 | 8.71 |
| CTAEA | 5.73 | 5.04 | 4.22 | 4.19 | 4.28 | 4.25 |
| TiGE2 | 8.75 | 8.39 | 7.31 | 6.21 | 5.58 | 4.84 |
| DCNSGAIII | 3.78 | 3.52 | 3.61 | 3.50 | 3.81 | 3.82 |
| CMME | **3.30** | **2.95** | **3.38** | **3.45** | **2.90** | **3.02** |
| ToP | 7.20 | 7.53 | 7.90 | 8.04 | 8.16 | 8.07 |
| CCMO | 5.73 | 6.08 | 6.13 | 7.05 | 7.29 | 7.14 |
| DDCMOEA | 5.45 | 5.79 | 6.00 | 6.97 | 7.29 | 6.98 |
| BiCo | 5.04 | 5.93 | 6.22 | 7.17 | 7.45 | 7.13 |
| dCMaOEA-RAE | **1.96** | **1.86** | **2.31** | **1.52** | **1.49** | **1.94** |

**Notes.**
The best results are in bold.

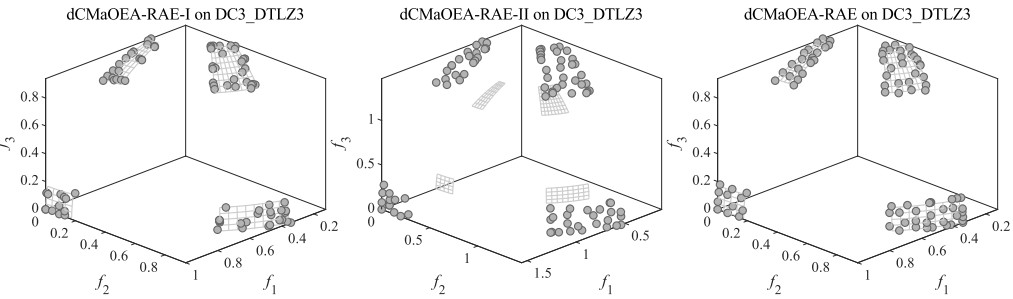

**Figure 8** Result of non-dominated solutions achieved on the three-objective DC3-DTLZ3 benchmark.

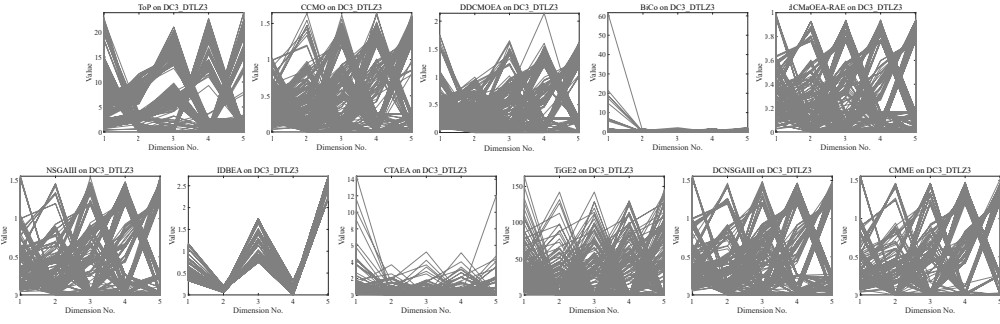

**Figure 9** Result of solutions on the five-objective DC3-DTLZ3 benchmark.

## *Intuitive experimental results*

To provide a visual representation of the experimental outcomes, Figs. 9, 10, 11 and 12 show the parallel coordinates of the final nondominated solutions that were achieved by
**Table 8  The IGD performance values of dCMaOEA-RAE and other schemes on MW and CF benchmark problems.**

| Problem | M | NSGAIII | IDBEA | CTAEA | TiGE2 | DCNSGAIII | CMME | ToP | CCMO | DDCMOEA | BiCo | dCMaOEA-RAE |
|---|---|---|---|---|---|---|---|---|---|---|---|---|
| MW4 | 3 | **4.1346e−2 (1.03e−4) +** | 5.3510e−1 (1.68e−1) − | 4.6574e−2 (3.86e−4) − | 9.2444e−2 (2.68e−2) − | 4.1775e−2 (3.05e−4) − | 4.1772e−2 (2.55e−3) − | NaN (NaN) | 4.2790e−2 (3.94e−4) − | 4.3134e−2 (4.43e−4) − | 4.3125e−2 (4.31e−4) − | 4.1556e−2 (8.50e−5) |
| | 5 | 1.0481e−1 (5.10e−5) − | 6.2830e−1 (1.63e−1) − | 1.2460e−1 (7.45e−4) − | 1.4335e−1 (4.64e−3) − | 1.0504e−1 (1.16e−4) − | 1.0471e−1 (4.73e−5) − | 3.4736e−1 (2.11e−1) − | 1.2323e−1 (1.90e−3) − | 1.2415e−1 (2.59e−3) − | 1.2582e−1 (2.54e−3) − | **1.0466e−1 (4.81e−5)** |
| | 8 | 1.9815e−1 (1.79e−2) = | 7.3962e−1 (1.92e−1) − | 2.4861e−1 (1.22e−3) − | 2.5424e−1 (1.29e−2) − | 1.9563e−1 (1.09e−2) − | 1.9424e−1 (5.16e−3) − | 3.3111e−1 (1.79e−1) − | 3.4269e−1 (1.21e−2) − | 3.1111e−1 (1.20e−2) − | 3.3144e−1 (1.04e−2) − | **1.9345e−1 (9.29e−5)** |
| | 10 | 2.1732e−1 (6.17e−4) − | 7.4992e−1 (1.78e−1) − | 2.8787e−1 (1.50e−2) − | 2.6195e−1 (4.54e−2) − | 2.2966e−1 (1.83e−2) − | 2.1779e−1 (7.00e−4) − | 2.7979e−1 (7.59e−3) − | 3.6658e−1 (7.92e−3) − | 3.7324e−1 (7.58e−2) − | 3.4167e−1 (7.16e−2) − | **2.1565e−1 (4.79e−4)** |
| | 15 | 3.7106e−1 (1.43e−3) − | 6.1677e−1 (8.02e−2) − | 4.3220e−1 (1.54e−2) − | 8.9734e−1 (1.45e−1) − | 3.7025e−1 (2.30e−3) = | 3.6383e−1 (3.40e−3) + | **3.3544e−1 (6.68e−3) +** | 3.8830e−1 (5.74e−3) − | 3.8721e−1 (5.39e−3) − | 3.6461e−1 (7.79e−3) + | 3.7002e−1 (2.57e−3) |
| MW8 | 3 | 5.5978e−2 (8.98e−3) − | 8.0068e−1 (1.77e−1) − | 5.5650e−2 (4.03e−3) − | 1.3725e−1 (1.03e−1) − | 4.9512e−2 (2.66e−3) − | 5.6059e−2 (2.87e−3) − | 7.5520e−1 (3.05e−1) − | 4.9376e−2 (5.24e−3) = | 5.0773e−2 (6.52e−3) − | 4.7367e−2 (1.50e−3) = | **4.6914e−2 (1.06e−3)** |
| | 5 | 1.7251e−1 (5.93e−2) − | 1.0041e+0 (1.35e−1) − | 1.7079e−1 (2.01e−1) − | 9.4621e−1 (1.03e−1) − | **1.5286e−1 (2.83e−4) +** | 1.5402e−1 (3.15e−3) − | 6.9024e−1 (4.27e−1) − | 1.6384e−1 (2.70e−3) − | 1.6491e−1 (2.46e−3) − | 1.6445e−1 (2.87e−3) − | 1.5297e−1 (1.57e−4) |
| | 8 | 3.3806e−1 (5.45e−2) − | 1.1610e+0 (1.23e−1) − | 3.5481e−1 (2.62e−3) − | 1.1195e+0 (5.39e−2) − | 3.1421e−1 (3.61e−2) − | 3.1036e−1 (1.08e−2) − | 5.1666e−1 (2.29e−1) − | 4.2377e−1 (5.63e−3) − | 3.8674e−1 (6.09e−3) − | 4.0797e−1 (6.46e−3) − | **3.0702e−1 (3.70e−4)** |
| | 10 | 4.0071e−1 (2.18e−2) − | 1.2049e+0 (1.01e−1) − | 4.1210e−1 (2.18e−3) − | 1.1830e+0 (5.14e−2) − | 4.0145e−1 (2.81e−2) − | 4.0576e−1 (1.00e−2) − | 4.6236e−1 (5.63e−3) − | 4.8420e−1 (4.96e−3) − | 4.8514e−1 (3.60e−3) − | 4.7549e−1 (4.31e−3) − | **3.9335e−1 (3.39e−4)** |
| | 15 | 6.3583e−1 (1.26e−2) − | 1.2613e+0 (1.10e−1) − | 6.4977e−1 (4.49e−3) − | 1.2661e+0 (3.78e−2) − | 6.3590e−1 (1.41e−2) − | 6.2361e−1 (1.99e−3) − | 6.4753e−1 (8.01e−3) − | 6.8639e−1 (5.22e−3) − | 6.8641e−1 (4.75e−3) − | 6.7954e−1 (4.56e−3) − | **6.2297e−1 (1.11e−3)** |
| MW14 | 3 | 1.4173e−1 (6.05e−2) − | 2.4198e+0 (4.30e−1) − | 1.1203e−1 (7.44e−3) − | 1.6621e−1 (8.43e−3) − | 1.3470e−1 (2.36e−2) − | 1.3348e−1 (5.32e−3) − | 3.9380e−1 (5.07e−1) − | 1.0776e−1 (1.86e−2) + | 1.0422e−1 (2.57e−3) + | **1.0359e−1 (3.47e−3) +** | 1.0718e−1 (2.33e−3) |
| | 5 | 3.5957e−1 (3.79e−2) − | 2.5722e+0 (6.35e−1) − | **3.0796e−1 (5.24e−3) +** | 5.1383e−1 (1.07e−1) − | 3.7462e−1 (3.84e−2) − | 5.9738e−1 (7.17e−2) − | 7.6569e−1 (1.06e−1) − | 3.3206e−1 (5.00e−2) = | 3.2991e−1 (7.00e−2) = | 3.4341e−1 (1.19e−2) − | 3.4779e−1 (6.39e−2) |
| | 8 | 8.4285e−1 (1.81e−2) − | 2.2333e+0 (2.56e−1) − | 1.0653e+0 (3.23e−2) − | 1.6886e+0 (1.66e−1) − | 8.5748e−1 (2.45e−2) − | 8.9219e−1 (4.80e−2) − | 1.1373e+0 (3.17e−2) − | 1.0958e+0 (8.18e−3) − | 9.6001e−1 (2.06e−2) − | 9.7758e−1 (1.20e−2) − | **8.2505e−1 (1.58e−2)** |
| | 10 | 1.1364e+0 (2.70e−2) − | 2.6975e+0 (1.64e−1) − | 1.4082e+0 (4.57e−2) − | 2.2269e+0 (8.02e−2) − | 1.1661e+0 (3.05e−2) − | 1.1100e+0 (5.22e−2) − | 1.3607e+0 (1.33e−2) − | 1.2092e+0 (2.07e−2) − | 1.2016e+0 (1.65e−2) − | 1.1445e+0 (8.87e−3) − | **1.0287e+0 (3.38e−2)** |
| | 15 | 3.3138e+0 (5.26e−2) − | 3.9256e+0 (1.29e−1) − | 3.0267e+0 (9.82e−2) − | 4.5330e+0 (3.33e−2) − | 3.2478e+0 (5.32e−2) − | 2.9776e+0 (1.27e−1) − | 2.3851e+0 (1.59e−2) = | **2.2046e+0 (4.51e−3) +** | 2.2060e+0 (3.90e−2) + | 2.2208e+0 (6.17e−3) + | 2.8376e+0 (1.07e−1) |
| Problem | M | NSGAIII | IDBEA | CTAEA | TiGE2 | DCNSGAIII | CMME | ToP | CCMO | DDCMOEA | BiCo | dCMaOEA-RAE |
| CF4 | 3 | 2.7874e−1 (2.36e−1) − | 1.9639e+0 (5.03e−1) − | 1.3014e−1 (1.13e−1) − | 3.1715e−1 (1.81e−1) − | 1.3509e−1 (7.97e−2) − | 3.0527e−1 (3.29e−1) − | 1.8169e+0 (0.00e+0) − | 1.4601e−1 (1.37e−1) = | 2.4416e−1 (2.33e−1) = | 1.4671e−1 (1.67e−1) − | **1.0281e−1 (6.06e−2)** |
| | 5 | 5.1186e−1 (3.53e−1) − | 1.7146e+0 (7.15e−1) − | 3.8089e−1 (1.52e−1) − | 7.0498e−1 (3.67e−1) − | 3.7751e−1 (2.53e−1) − | 4.2964e−1 (2.60e−1) − | 1.9103e+0 (2.33e−1) − | 4.6312e−1 (2.69e−1) − | 3.2679e−1 (1.80e−1) − | 3.5408e−1 (1.65e−1) − | **2.4837e−1 (7.57e−2)** |
| | 8 | 7.3985e−1 (3.45e−1) − | 1.4417e+0 (5.13e−1) − | 5.5977e−1 (1.74e−1) − | 8.5278e−1 (5.71e−1) − | 6.2013e−1 (2.12e−1) − | 5.7799e−1 (7.47e−2) − | 1.7747e+0 (8.05e−1) − | 1.0621e+0 (3.96e−1) − | 1.1457e+0 (5.08e−1) − | 1.1823e+0 (4.13e−1) − | **5.0722e−1 (2.23e−1)** |
| | 10 | 6.7419e−1 (1.24e−1) − | 1.1133e+0 (3.34e−1) − | 5.8305e−1 (1.33e−1) = | 7.7031e−1 (2.71e−1) − | 6.2555e−1 (2.94e−2) − | 6.5622e−1 (3.80e−2) − | 1.7666e+0 (9.80e−1) − | 1.3395e+0 (3.48e−1) − | 1.2291e+0 (3.44e−1) − | 1.5427e+0 (4.84e−1) − | **5.2266e−1 (2.64e−2)** |
| | 15 | 8.8459e−1 (4.42e−1) − | 1.0728e+0 (1.36e−1) − | 7.1502e−1 (1.90e−1) = | 7.6792e−1 (9.57e−2) − | 7.8034e−1 (1.39e−2) − | 7.8149e−1 (3.94e−2) − | 1.7647e+0 (7.64e−1) − | 1.8277e+0 (6.25e−1) − | 2.4202e+0 (1.34e+0) − | 1.8478e+0 (7.73e−1) − | **6.8444e−1 (7.63e−2)** |
| CF8 | 3 | 3.9755e−2 (1.64e−3) − | 1.1639e+0 (4.42e−1) − | 1.0639e−1 (2.95e−2) − | 8.6543e−2 (8.95e−2) − | 4.3037e−2 (1.56e−2) − | 4.0327e−2 (8.39e−4) − | 7.0123e−1 (3.23e−1) − | 3.7801e−2 (1.34e−3) − | 3.8347e−2 (1.09e−2) − | 3.8073e−2 (1.40e−3) = | **3.7546e−2 (9.19e−4)** |
| | 5 | 1.7553e−1 (2.60e−2) − | 1.0117e+0 (4.14e−1) − | 2.1613e−1 (5.33e−2) − | 2.0680e−1 (1.00e−1) − | 1.8213e−1 (6.87e−3) − | 1.6425e−1 (4.45e−3) − | 8.1078e−1 (1.48e−1) − | 1.9636e−1 (1.05e−2) − | 1.7891e−1 (7.62e−3) − | 2.0177e−1 (7.94e−3) − | **1.5310e−1 (4.78e−3)** |
| | 8 | 2.3475e−1 (1.34e−2) − | 6.3313e−1 (1.81e−1) − | 3.9845e−1 (1.62e−1) − | 3.2142e−1 (3.92e−2) − | 2.2564e−1 (1.95e−2) − | 2.1694e−1 (1.35e−2) − | 6.7238e−1 (1.37e−1) − | 7.0495e−1 (1.01e−1) − | 7.1514e−1 (1.16e−1) − | 8.2211e−1 (1.59e−1) − | **2.1246e−1 (7.63e−3)** |
| | 10 | 2.9697e−1 (2.49e−2) − | 7.9193e−1 (1.96e−1) − | 4.2456e−1 (1.20e−1) − | 3.8367e−1 (4.47e−1) − | 2.9648e−1 (3.08e−2) − | 3.0264e−1 (1.99e−2) − | 6.7894e−1 (7.12e−2) − | 1.0749e+0 (1.05e−1) − | 9.9753e−1 (9.53e−2) − | 1.1600e+0 (2.44e−1) − | **2.4757e−1 (1.86e−2)** |
| | 15 | 3.1541e−1 (2.25e−2) − | 7.3267e−1 (1.01e−1) − | 3.7981e−1 (1.81e−2) − | 4.8627e−1 (4.87e−2) − | 2.7887e−1 (1.89e−2) + | **2.7830e−1 (1.81e−2)** | 6.9293e−1 (8.91e−2) − | 1.4265e+0 (2.58e−1) − | 1.6848e+0 (9.48e−2) − | 1.6510e+0 (1.93e−1) − | 3.1348e−1 (1.19e−2) |
| CF12 | 3 | 4.8641e−1 (3.01e−1) − | 1.1384e+0 (4.25e−1) − | 1.5854e−1 (2.58e−1) − | 3.3710e−1 (2.96e−1) − | 1.1978e−1 (2.38e−1) − | 2.0562e−1 (2.41e−1) − | 8.0750e−1 (2.73e−1) − | 1.2534e−1 (2.68e−1) − | 1.2139e−1 (1.92e−1) = | 6.9080e−2 (1.41e−1) = | **4.5687e−2 (6.05e−2)** |
| | 5 | 3.5444e−1 (2.88e−1) − | 8.7013e−1 (3.99e−1) − | 1.8317e−1 (1.68e−1) = | 4.1561e−1 (1.12e−1) − | 2.3564e−1 (2.22e−1) − | 2.0452e−1 (1.22e−1) − | 4.4233e−1 (1.77e−1) − | 3.9534e−1 (2.39e−1) − | 2.6854e−1 (1.61e−1) − | 4.2466e−1 (2.28e−1) − | **1.2527e−1 (2.75e−2)** |
| | 8 | 3.2852e−1 (1.36e−1) − | 5.8694e−1 (9.36e−2) − | 3.3983e−1 (8.80e−2) − | 4.7955e−1 (9.13e−2) − | 2.8077e−1 (1.11e−1) − | 2.6319e−1 (3.47e−2) − | 4.0280e−1 (7.65e−2) − | 7.6433e−1 (7.39e−2) − | 7.5671e−1 (7.49e−2) − | 7.8730e−1 (2.80e−2) − | **2.0841e−1 (2.31e−2)** |
| | 10 | 3.4037e−1 (6.52e−2) − | 5.3685e−1 (1.04e−1) − | 3.0463e−1 (6.32e−2) − | 5.6965e−1 (9.14e−2) − | 3.0606e−1 (7.40e−2) − | 3.0401e−1 (2.01e−2) − | 3.5304e−1 (3.65e−2) − | 8.5742e−1 (3.75e−2) − | 8.1686e−1 (6.66e−2) − | 8.1813e−1 (3.55e−2) − | **2.2183e−1 (1.85e−2)** |
| | 15 | 3.0488e−1 (1.13e−1) − | 5.3377e−1 (1.04e−1) − | 4.8640e−1 (8.85e−2) − | 6.3978e−1 (3.16e−1) − | 3.6300e−1 (1.29e−1) − | **2.7465e−1 (2.70e−2) +** | 3.5823e−1 (2.49e−2) − | 1.0110e+0 (3.37e−2) − | 1.0934e+0 (3.53e−2) − | 8.7825e−1 (3.15e−2) − | 2.7757e−1 (6.48e−2) |
| +/−/= | | 0/13/2 | 0/15/0 | 0/12/3 | 0/15/0 | 1/14/0 | 2/12/1 | 0/14/1 | 0/12/3 | 0/12/3 | 0/13/2 | |

**Notes.**

The best results are in bold.
**Table 9  The IGDp performance values of dCMaOEA-RAE and other schemes on MW and CF benchmark problems.**

| Problem | M | NSGAIII | IDBEA | CTAEA | TiGE2 | DCNSGAIII | CMME | ToP | CCMO | DDCMOEA | BiCo | dCMaOEA-RAE |
|---|---|---|---|---|---|---|---|---|---|---|---|---|
| MW4 | 3 | **2.9289e−2 (1.03e−4)** + | 4.2468e−1 (1.58e−1) - | 3.2597e−2 (2.71e−4) - | 6.3474e−2 (1.63e−2) - | 2.9571e−2 (2.29e−4) = | 2.9623e−2 (2.04e−3) - | NaN (NaN) | 3.0663e−2 (4.53e−4) - | 3.0932e−2 (4.53e−4) - | 3.1015e−2 (4.29e−4) - | 2.9557e−2 (1.29e−4) |
|  | 5 | 7.4796e−2 (7.03e−5) - | 5.0429e−1 (1.61e−1) - | 8.8288e−2 (5.45e−4) - | 9.9578e−2 (3.04e−3) - | 7.5102e−2 (1.43e−2) - | **7.4678e−2 (6.44e−5)** - | 2.8461e−1 (1.74e−1) - | 9.4456e−2 (1.70e−3) - | 9.4854e−2 (2.38e−3) - | 9.8683e−2 (2.63e−3) - | 7.4730e−2 (7.12e−5) |
|  | 8 | 1.2979e−1 (9.87e−3) - | 5.8795e−1 (2.04e−1) - | 1.8021e−1 (1.46e−3) - | 1.7057e−1 (7.94e−3) - | 1.2823e−1 (4.97e−3) - | 1.2786e−1 (4.50e−3) - | 2.6204e−1 (1.54e−1) - | 3.0516e−1 (2.05e−2) - | 2.7824e−1 (1.93e−2) - | 2.9891e−1 (1.57e−2) - | **1.2754e−1 (1.83e−4)** |
|  | 10 | 1.3713e−1 (5.73e−4) - | 6.1016e−1 (1.99e−1) - | 2.1316e−1 (1.51e−3) - | 1.7944e−1 (4.32e−3) - | 1.4487e−1 (1.11e−2) - | 1.3768e−1 (5.31e−4) - | 2.1757e−1 (8.55e−3) - | 3.2563e−1 (1.47e−2) - | 3.3593e−1 (1.31e−2) - | 3.0570e−1 (1.26e−2) - | **1.3581e−1 (5.27e−4)** |
|  | 15 | 2.5309e−1 (2.03e−3) - | 4.5806e−1 (8.82e−2) - | 3.3969e−1 (2.11e−2) - | 7.7342e−1 (1.61e−1) - | 2.5193e−1 (3.41e−2) = | 2.4334e−1 (4.98e−2) + | **2.2496e−1 (8.13e−3)** + | 2.9788e−1 (6.89e−3) - | 2.9638e−1 (6.52e−3) - | 2.7074e−1 (9.05e−3) - | 2.5177e−1 (3.62e−3) |
| MW8 | 3 | 3.9993e−2 (1.42e−2) - | 5.7774e−1 (9.08e−2) - | 3.3213e−2 (9.00e−3) - | 7.0014e−2 (5.32e−2) - | 2.8480e−2 (6.68e−3) - | 3.2699e−2 (6.96e−3) - | 5.1278e−1 (2.02e−1) - | 3.2390e−2 (9.86e−3) - | 3.4976e−2 (1.07e−2) - | 2.5398e−2 (4.45e−3) - | **2.3117e−2 (4.69e−3)** |
|  | 5 | 7.0555e−2 (2.88e−2) - | 7.4154e−1 (7.69e−2) - | 6.9426e−2 (3.12e−3) - | 5.6557e−2 (9.53e−3) - | 5.9255e−2 (2.50e−3) - | 6.0619e−2 (3.94e−3) - | 5.1355e−1 (2.94e−2) - | 9.9077e−2 (5.55e−3) - | 1.0132e−1 (5.79e−3) - | 1.0259e−1 (5.12e−3) - | **5.8090e−2 (1.41e−3)** |
|  | 8 | 1.3577e−1 (3.06e−2) - | 8.5910e−1 (8.38e−2) - | 1.3838e−1 (4.46e−3) - | 7.3811e−1 (4.13e−1) - | 1.2060e−1 (2.03e−2) - | 1.1870e−1 (5.41e−2) - | 3.9139e−1 (1.74e−1) - | 3.2516e−1 (9.41e−3) - | 2.9701e−1 (8.61e−3) - | 3.1057e−1 (1.10e−2) - | **1.1446e−1 (1.31e−3)** |
|  | 10 | 1.6430e−1 (1.40e−2) - | 8.4317e−1 (5.78e−2) - | **1.4699e−1 (3.24e−3)** + | 7.7858e−1 (3.46e−2) - | 1.6406e−1 (1.83e−2) = | 1.5094e−1 (8.88e−3) + | 3.2069e−1 (1.32e−2) - | 3.7183e−1 (7.68e−3) - | 3.7267e−1 (5.58e−3) - | 3.6790e−1 (6.13e−3) - | 1.5847e−1 (2.07e−4) |
|  | 15 | 2.5413e−1 (1.25e−2) - | 8.2425e−1 (1.07e−1) - | 2.7904e−1 (5.16e−3) - | 8.3885e−1 (2.98e−2) - | 2.5562e−1 (1.42e−2) - | 2.3796e−1 (2.83e−3) = | 3.9988e−1 (1.48e−1) - | 5.3852e−1 (1.30e−2) - | 5.3698e−1 (1.03e−2) - | 5.3801e−1 (9.76e−3) - | **2.3786e−1 (3.33e−4)** |
| MW14 | 3 | 9.0654e−2 (4.74e−2) - | 1.3525e+0 (7.16e−2) - | **6.3732e−2 (6.30e−3)** + | 8.5369e−2 (5.71e−3) - | 8.3313e−2 (1.49e−2) - | 7.7527e−2 (4.80e−3) - | 3.1444e−1 (4.32e−1) - | 7.1622e−2 (1.21e−2) - | 6.9085e−2 (3.32e−3) - | 6.7223e−2 (3.63e−3) - | 6.5913e−2 (2.88e−3) |
|  | 5 | 2.5164e−1 (2.71e−2) - | 1.6923e+0 (2.39e−1) - | **1.8814e−1 (4.90e−3)** + | 2.9914e−1 (4.64e−2) - | 2.6388e−1 (2.77e−2) - | 4.4066e−1 (7.58e−2) - | 6.7475e−1 (1.23e−1) - | 2.2313e−1 (4.57e−1) = | 2.2090e−1 (4.88e−2) = | 2.6601e−1 (7.40e−3) - | 2.2524e−1 (3.29e−2) |
|  | 8 | 6.0095e−1 (1.82e−2) - | 1.6448e+0 (2.44e−1) - | 8.2161e−1 (4.28e−2) - | 9.7918e−1 (1.29e−1) - | 6.1068e−1 (1.57e−2) - | 6.0961e−1 (2.01e−2) - | 9.3345e−1 (5.24e−2) - | 8.5644e−1 (1.64e−2) - | 7.0419e−1 (2.68e−2) - | 7.5409e−1 (1.99e−2) - | **5.5625e−1 (1.02e−2)** |
|  | 10 | 6.8197e−1 (1.38e−2) - | 1.7954e+0 (1.32e−1) - | 1.0278e+0 (4.30e−2) - | 1.2109e+0 (4.29e−2) - | 7.0376e−1 (1.82e−2) - | 6.8679e−1 (2.60e−2) - | 1.0212e+0 (2.81e−2) - | 9.7390e−1 (2.92e−2) - | 9.6252e−1 (2.42e−2) - | 8.9199e−1 (1.40e−2) - | **6.2209e−1 (1.65e−2)** |
|  | 15 | 1.5370e+0 (3.28e−2) - | 1.9926e+0 (1.17e−1) - | 1.8635e+0 (4.55e−2) - | 2.4193e+0 (2.87e−2) - | 1.5042e+0 (3.45e−2) - | 1.6214e+0 (1.41e−1) - | 1.5120e+0 (1.85e−2) - | 1.3596e+0 (3.33e−2) = | 1.3643e+0 (2.48e−2) - | 1.4821e+0 (3.19e−2) - | **1.3495e+0 (3.76e−2)** |

| Problem | M | NSGAIII | IDBEA | CTAEA | TiGE2 | DCNSGAIII | CMME | ToP | CCMO | DDCMOEA | BiCo | dCMaOEA-RAE |
|---|---|---|---|---|---|---|---|---|---|---|---|---|
| CF4 | 3 | 2.5749e−1 (2.48e−1) - | 1.9349e+0 (4.94e−1) - | 1.0778e−1 (1.23e−1) - | 2.7235e−1 (2.08e−1) - | 1.1140e−1 (9.26e−2) - | 2.8786e−1 (3.41e−1) - | 1.7723e+0 (0.00e+0) - | 1.3125e−1 (1.46e−1) - | 2.3481e−1 (2.40e−1) = | 1.3039e−1 (1.75e−1) = | **7.9945e−2 (7.23e−2)** |
|  | 5 | 4.6652e−1 (3.79e−1) - | 1.6673e+0 (7.29e−1) - | 3.1371e−1 (1.77e−1) - | 6.1577e−1 (4.18e−1) - | 3.0539e−1 (2.79e−1) - | 3.6565e−1 (2.90e−1) - | 1.9036e+0 (2.44e−1) - | 4.2723e−1 (2.90e−1) - | 2.8740e−1 (2.02e−1) - | 3.0548e−1 (1.90e−1) - | **1.6527e−1 (1.04e−1)** |
|  | 8 | 6.0122e−1 (4.24e−1) - | 1.3679e+0 (5.52e−1) - | 4.2633e−1 (2.35e−1) - | 7.4308e−1 (6.19e−1) - | 4.4543e−1 (2.89e−1) - | 3.9912e−1 (1.60e−1) - | 1.7444e+0 (8.22e−1) - | 9.7088e−1 (4.37e−1) - | 1.0620e+0 (5.51e−1) - | 1.1019e+0 (4.60e−1) - | **3.4447e−1 (2.81e−1)** |
|  | 10 | 4.7550e−1 (2.01e−1) - | 9.6959e−1 (3.97e−1) - | 4.1533e−1 (1.87e−1) - | 5.9737e−1 (3.44e−1) - | 3.8232e−1 (9.90e−2) - | 4.4791e−1 (1.32e−1) - | 1.7040e+0 (1.03e+0) - | 1.2273e+0 (3.76e−1) - | 1.1269e+0 (3.87e−1) - | 1.4596e+0 (5.21e−1) - | **2.9085e−1 (4.97e−2)** |
|  | 15 | 6.8697e−1 (5.09e−1) - | 9.0722e−1 (2.11e−1) - | 5.0084e−1 (2.38e−1) - | 5.2603e−1 (1.83e−1) - | 5.1275e−1 (1.25e−1) - | 5.3619e−1 (1.58e−1) - | 1.7417e+0 (7.70e−1) - | 1.6453e+0 (6.89e−1) - | 2.2494e+0 (1.43e+0) - | 1.7250e+0 (8.22e−1) - | **4.4977e−1 (1.70e−1)** |
| CF8 | 3 | 3.1652e−2 (1.94e−3) - | 1.1386e+0 (4.73e−1) - | 8.5985e−2 (2.06e−2) - | 6.0930e−2 (6.76e−3) - | 3.7001e−2 (1.72e−3) - | 3.0683e−2 (7.98e−4) = | 6.9780e−1 (3.25e−1) - | 3.2867e−2 (1.70e−2) - | 3.3349e−2 (1.27e−3) - | 3.3070e−2 (1.80e−3) - | **2.9468e−2 (1.01e−3)** |
|  | 5 | 1.1606e−1 (1.13e−2) - | 9.0091e−1 (4.76e−1) - | 1.3901e−1 (4.64e−2) - | 1.0173e−1 (6.84e−3) - | 1.2223e−1 (6.03e−2) - | 9.7557e−2 (3.71e−3) - | 7.7930e−1 (1.54e−1) - | 1.6096e−1 (1.34e−2) - | 1.5010e−1 (8.61e−3) - | 1.6823e−1 (9.00e−3) - | **9.0641e−2 (3.57e−3)** |
|  | 8 | 1.6626e−1 (1.16e−2) - | 4.5019e−1 (2.51e−1) - | 2.9224e−1 (1.59e−1) - | 1.5502e−1 (1.64e−2) - | 1.6170e−1 (1.47e−2) - | **1.2843e−1 (5.95e−3)** + | 6.3877e−1 (1.47e−1) - | 6.5518e−1 (1.16e−1) - | 6.7043e−1 (1.24e−1) - | 7.6669e−1 (1.91e−1) - | 1.4202e−1 (9.04e−3) |
|  | 10 | 1.8864e−1 (1.84e−2) - | 5.7974e−1 (2.60e−1) - | 2.7780e−1 (1.11e−1) - | 1.9705e−1 (2.25e−2) - | 1.9541e−1 (2.13e−2) - | **1.5847e−1 (8.78e−3)** + | 6.2191e−1 (8.30e−2) - | 1.0119e+0 (1.21e−1) - | 9.3561e−1 (1.19e−1) - | 1.0793e+0 (2.77e−1) - | 1.5876e−1 (1.47e−2) |
|  | 15 | 2.5077e−1 (2.40e−2) - | 5.3343e−1 (1.58e−1) - | 2.4071e−1 (1.27e−2) - | 2.6381e−1 (2.25e−2) - | 2.3607e−1 (1.65e−2) - | **1.7904e−1 (1.19e−2)** + | 6.3907e−1 (9.60e−2) - | 1.3891e+0 (2.91e−1) - | 1.6652e+0 (1.13e−1) - | 1.6332e+0 (2.15e−1) - | 2.2342e−1 (1.19e−2) |
| CF12 | 3 | 3.8140e−1 (3.05e−1) - | 1.0455e+0 (4.42e−1) - | 1.5463e−1 (2.60e−1) - | 3.0743e−1 (3.11e−1) - | 1.1515e−1 (2.39e−1) - | 1.9649e−1 (2.40e−1) - | 7.3841e−1 (2.89e−1) - | 1.2271e−1 (2.69e−1) - | 1.1983e−1 (1.92e−1) - | 6.6585e−2 (1.42e−1) = | **4.1701e−2 (6.16e−2)** |
|  | 5 | 3.0979e−1 (2.77e−1) - | 7.7571e−1 (4.18e−1) - | 1.6041e−1 (1.75e−1) = | 3.4556e−1 (1.24e−1) - | 2.1588e−1 (2.30e−1) - | 1.7765e−1 (1.33e−1) - | 3.9326e−1 (1.63e−1) - | 3.7825e−1 (2.44e−1) - | 2.5649e−1 (1.64e−1) - | 4.0799e−1 (2.34e−1) - | **1.0132e−1 (3.18e−2)** |
|  | 8 | 2.6319e−1 (1.32e−1) - | 4.1857e−1 (1.21e−1) - | 2.8634e−1 (1.03e−1) - | 3.9054e−1 (1.07e−1) - | 2.1781e−1 (1.28e−1) - | 2.0438e−1 (4.80e−2) - | 3.4100e−1 (8.98e−2) - | 7.3697e−1 (9.46e−2) - | 7.2558e−1 (9.66e−2) - | 7.8037e−1 (2.94e−2) - | **1.5545e−1 (3.19e−2)** |
|  | 10 | 2.7927e−1 (8.13e−2) - | 3.8577e−1 (1.28e−1) - | 2.4053e−1 (7.44e−2) - | 4.8997e−1 (1.10e−1) - | 2.3407e−1 (8.52e−2) - | 2.2826e−1 (2.57e−2) - | 2.8170e−1 (4.36e−2) - | 8.3995e−1 (4.44e−2) - | 7.9354e−1 (8.68e−2) - | 8.0399e−1 (4.35e−2) - | **1.5412e−1 (2.55e−2)** |
|  | 15 | 2.1264e−1 (1.40e−1) = | 3.7450e−1 (1.29e−1) - | 4.1921e−1 (1.04e−1) - | 5.4726e−1 (3.67e−2) - | 2.7843e−1 (1.55e−1) - | **1.7495e−1 (3.61e−2)** + | 2.4987e−1 (2.01e−2) - | 9.8260e−1 (3.76e−2) - | 1.0658e+0 (3.91e−2) - | 8.5561e−1 (3.35e−2) - | 1.8532e−1 (8.48e−2) |
| +/-/= | | 0/14/1 | 0/15/0 | 0/13/2 | 0/15/0 | 0/15/0 | 2/11/2 | 0/14/1 | 0/13/2 | 0/14/1 | 0/13/2 | |

**Notes.**

The best results are in bold.

**Table 10  The HV performance values of dCMaOEA-RAE and other schemes on MW and CF benchmark problems.**

| Problem | M | NSGAIII | IDBEA | CTAEA | TiGE2 | DCNSGAIII | CMME | ToP | CCMO | DDCMOEA | BiCo | dCMaOEA-RAE |
|---|---|---|---|---|---|---|---|---|---|---|---|---|
| MW4 | 3 | **8.4145e−1 (1.08e−4)** + | 2.3817e−1 (9.33e−2) − | 8.3812e−1 (2.84e−4) − | 7.7874e−1 (2.80e−2) − | 8.4107e−1 (3.45e−4) − | 8.4071e−1 (3.79e−3) − | NaN (NaN) | 8.3983e−1 (5.83e−4) − | 8.3959e−1 (5.83e−4) − | 8.3914e−1 (5.40e−4) − | 8.4112e−1 (1.44e−4) |
|  | 5 | 9.7987e−1 (1.42e−4) = | 2.5165e−1 (8.31e−2) − | 9.7694e−1 (2.15e−4) − | 9.6827e−1 (2.11e−3) − | 9.7980e−1 (1.33e−4) = | **9.7988e−1 (1.50e−4)** + | 7.0289e−1 (2.37e−1) − | 9.7335e−1 (7.11e−4) − | 9.7311e−1 (9.90e−4) − | 9.7028e−1 (1.14e−3) − | 9.7980e−1 (1.36e−4) |
|  | 8 | 9.9713e−1 (2.01e−3) − | 2.5338e−1 (9.73e−2) − | 9.9625e−1 (1.35e−4) − | 9.8708e−1 (6.65e−3) − | 9.9750e−1 (5.78e−4) − | 9.9754e−1 (2.22e−4) − | 8.9256e−1 (2.10e−1) − | 9.7439e−1 (3.49e−2) − | 9.8316e−1 (2.28e−3) − | 9.7518e−1 (2.04e−3) − | **9.9758e−1 (5.23e−5)** |
|  | 10 | 9.9968e−1 (2.27e−5) = | 2.5479e−1 (1.14e−1) − | 9.9944e−1 (2.77e−5) − | 9.9858e−1 (3.97e−4) − | 9.9902e−1 (1.80e−3) − | **9.9968e−1 (1.37e−5)** = | 9.9412e−1 (9.99e−4) − | 9.9332e−1 (4.29e−4) − | 9.9302e−1 (3.61e−4) − | 9.9418e−1 (3.02e−4) − | 9.9967e−1 (1.96e−5) |
|  | 15 | 9.9990e−1 (1.02e−5) − | 6.3492e−1 (1.25e−1) − | 9.9986e−1 (1.50e−5) − | 2.4233e−1 (1.50e−1) − | 9.9990e−1 (7.66e−6) − | 9.9990e−1 (1.01e−5) − | 9.9976e−1 (3.31e−5) − | 9.9992e−1 (8.87e−6) − | 9.9992e−1 (1.02e−5) − | **9.9993e−1 (8.97e−6)** + | 9.9990e−1 (1.09e−5) |
| MW8 | 3 | 5.1180e−1 (2.47e−2) − | 6.0227e−2 (4.48e−2) − | 5.1631e−1 (1.59e−2) − | 4.3885e−1 (9.40e−2) − | 5.3113e−1 (1.17e−2) − | 5.1983e−1 (1.28e−2) − | 1.1356e−1 (1.29e−1) − | 5.2841e−1 (1.79e−2) − | 5.2341e−1 (1.88e−2) − | **5.4133e−1 (7.78e−3)** + | 5.4121e−1 (8.41e−3) |
|  | 5 | 7.9491e−1 (3.93e−2) − | 4.9124e−1 (3.14e−2) − | 7.8761e−1 (4.73e−3) − | 1.4422e−1 (5.68e−2) − | 8.1026e−1 (2.86e−2) − | 8.0784e−1 (5.90e−3) − | 2.8673e−1 (2.49e−1) − | 7.6652e−1 (7.43e−3) − | 7.6249e−1 (7.65e−3) − | 7.6013e−1 (7.46e−3) − | **8.1157e−1 (1.77e−3)** |
|  | 8 | 9.0627e−1 (2.86e−2) − | 5.5880e−2 (3.60e−2) − | 9.0678e−1 (4.21e−3) − | 1.2706e−1 (2.40e−2) − | 9.1896e−1 (1.60e−2) − | 9.2039e−1 (3.85e−3) − | 5.2887e−1 (1.65e−2) − | 6.9051e−1 (1.78e−2) − | 7.5224e−1 (1.66e−2) − | 7.2151e−1 (1.95e−2) − | **9.2357e−1 (8.28e−4)** |
|  | 10 | 9.6606e−1 (9.22e−3) − | 7.6649e−2 (2.52e−2) − | 9.6408e−1 (1.36e−2) − | 1.2046e−1 (2.66e−2) − | 9.6550e−1 (1.43e−2) − | 9.6760e−1 (1.91e−2) − | 7.3265e−1 (2.11e−2) − | 7.8154e−1 (9.06e−3) − | 7.7941e−1 (8.40e−3) − | 7.8296e−1 (9.39e−3) − | **9.6948e−1 (1.91e−4)** |
|  | 15 | 9.8076e−1 (8.48e−3) − | 1.4583e−1 (1.61e−1) − | 9.3112e−1 (7.04e−3) − | 1.1517e−1 (3.28e−2) − | 9.7927e−1 (9.93e−3) − | 9.8822e−1 (4.46e−3) − | 7.3366e−1 (2.97e−2) − | 7.0558e−1 (9.71e−3) − | 7.0550e−1 (8.88e−3) − | 6.9588e−1 (1.64e−2) − | **9.9046e−1 (1.24e−4)** |
| MW14 | 3 | 4.5879e−1 (2.83e−2) − | 1.3010e−1 (6.73e−3) − | 4.6637e−1 (3.77e−3) − | 4.4582e−1 (4.54e−2) − | 4.6379e−1 (7.79e−3) − | 4.6464e−1 (3.70e−2) − | 3.5887e−1 (1.61e−1) − | 4.6872e−1 (6.28e−3) − | 4.7030e−1 (3.63e−3) − | 4.6572e−1 (4.10e−3) − | **4.7200e−1 (2.36e−3)** |
|  | 5 | 3.5753e−1 (1.61e−2) − | 3.3504e−3 (7.86e−3) − | **3.9543e−1 (3.88e−3)** + | 3.6694e−1 (1.62e−2) − | 3.5286e−1 (1.91e−2) − | 2.7439e−1 (2.08e−2) − | 1.4241e−1 (4.90e−2) − | 3.8912e−1 (3.12e−3) + | 3.9010e−1 (2.31e−3) − | 3.7165e−1 (2.79e−3) − | 3.7269e−1 (1.59e−2) |
|  | 8 | 2.2297e−1 (5.40e−2) − | 7.7490e−3 (1.13e−2) − | 2.4924e−1 (6.82e−3) − | **2.5048e−1 (1.98e−2)** + | 2.2710e−1 (7.45e−3) − | 2.1931e−1 (1.82e−2) − | 7.1659e−2 (1.66e−2) − | 1.3412e−1 (1.21e−2) − | 1.6040e−1 (9.08e−3) − | 1.8116e−1 (9.99e−3) − | 2.3804e−1 (2.19e−2) |
|  | 10 | 2.0448e−1 (3.94e−3) − | 5.9731e−3 (1.03e−2) − | 2.3281e−1 (5.15e−3) − | **2.4060e−1 (1.02e−2)** + | 2.0460e−1 (4.41e−3) − | 2.1629e−1 (1.22e−2) = | 8.3505e−2 (2.35e−2) − | 9.8631e−2 (2.09e−2) − | 8.7729e−2 (1.93e−2) − | 1.0571e−1 (2.06e−2) − | 2.1926e−1 (6.31e−3) |
|  | 15 | 1.5314e−1 (2.81e−3) − | 1.5010e−1 (6.97e−3) − | 1.5495e−1 (2.72e−3) − | 1.0444e−1 (4.94e−3) − | 1.4540e−1 (2.01e−3) − | **1.5952e−1 (3.96e−3)** = | 1.5091e−1 (1.43e−2) − | 5.0262e−3 (4.34e−3) − | 3.8760e−3 (3.44e−3) − | 4.5549e−3 (9.60e−3) − | 1.5950e−1 (3.10e−3) |
| Problem | M | NSGAIII | IDBEA | CTAEA | TiGE2 | DCNSGAIII | CMME | ToP | CCMO | DDCMOEA | BiCo | dCMaOEA-RAE |
| CF4 | 3 | 2.1851e−1 (1.39e−1) − | 0.0000e+0 (0.00e+0) − | 3.2993e−1 (1.08e−1) − | 1.6366e−1 (1.21e−1) − | 3.1812e−1 (1.03e−1) − | 2.2409e−1 (1.47e−1) − | 0.0000e+0 (0.00e+0) − | 3.0642e−1 (1.31e−1) = | 2.3407e−1 (1.47e−1) − | 3.1949e−1 (1.32e−1) − | **3.5245e−1 (8.95e−2)** |
|  | 5 | 1.4248e−2 (1.94e−2) − | 0.0000e+0 (0.00e+0) − | 2.2867e−2 (1.81e−2) − | 1.0441e−2 (1.34e−2) − | 2.8453e−2 (2.06e−2) − | 2.4517e−2 (2.64e−2) − | 0.0000e+0 (0.00e+0) − | 1.6121e−2 (1.97e−2) − | 3.2761e−2 (2.77e−2) − | 2.6162e−2 (2.06e−2) − | **5.7694e−2 (2.60e−2)** |
|  | 8 | 6.8585e−4 (7.51e−4) − | 2.7990e−7 (1.29e−6) − | 5.5645e−4 (4.88e−4) − | 2.1065e−4 (3.95e−4) − | 1.0133e−3 (6.83e−4) − | 1.3023e−3 (1.07e−3) − | 0.0000e+0 (0.00e+0) − | 9.9240e−8 (2.70e−7) − | 2.6130e−7 (9.19e−7) − | 2.3805e−6 (6.21e−6) − | **1.3651e−3 (8.93e−4)** |
|  | 10 | 5.7391e−5 (5.13e−5) − | 3.1228e−7 (8.93e−7) − | 2.5140e−5 (1.96e−5) − | 3.7022e−5 (3.77e−5) − | 1.0095e−4 (5.25e−5) − | 1.1084e−4 (1.05e−4) − | 0.0000e+0 (0.00e+0) − | 2.2733e−11 (8.99e−11) − | 1.9845e−9 (1.06e−8) − | 6.1339e−9 (3.21e−8) − | **1.2048e−4 (2.76e−5)** |
|  | 15 | 8.3250e−8 (8.70e−8) = | 6.3372e−11 (1.61e−10) − | 1.3009e−8 (1.23e−8) − | 6.2761e−8 (4.98e−8) − | 1.2302e−7 (7.58e−8) − | **1.4354e−7 (1.12e−7)** + | 0.0000e+0 (0.00e+0) − | 7.7199e−16 (3.62e−15) − | 3.1682e−15 (9.49e−15) − | 2.3660e−15 (9.48e−15) − | 9.6979e−8 (5.33e−8) |
| CF8 | 3 | 2.2498e−1 (2.72e−2) − | 3.7582e−2 (1.10e−2) − | 1.6136e−1 (2.08e−2) − | 1.9275e−1 (7.88e−3) − | 2.1746e−1 (2.22e−2) − | 2.2573e−1 (1.57e−2) − | 7.0995e−2 (1.47e−2) − | 2.2579e−1 (2.10e−2) − | 2.2531e−1 (1.55e−2) − | 2.2433e−1 (2.19e−2) − | **2.2680e−1 (1.60e−2)** |
|  | 5 | 2.6927e−1 (1.29e−2) − | 1.0666e−2 (1.62e−2) − | 2.3169e−1 (6.21e−2) − | 2.8069e−1 (9.69e−2) − | 2.6228e−1 (8.43e−2) − | 2.9133e−1 (4.99e−2) − | 1.6904e−1 (3.06e−2) − | 2.1877e−1 (1.47e−2) − | 2.3418e−1 (1.03e−2) − | 2.0919e−1 (1.01e−2) − | **3.0242e−1 (4.74e−3)** |
|  | 8 | 7.0072e−2 (1.03e−2) − | 1.0188e−2 (8.94e−3) − | 3.6058e−2 (3.85e−2) − | 9.3233e−2 (7.72e−3) − | 7.0002e−2 (1.36e−2) − | **1.0138e−1 (6.13e−3)** + | 1.7981e−4 (4.74e−4) − | 9.3405e−5 (2.92e−4) − | 1.0920e−4 (5.65e−4) − | 3.8865e−5 (2.13e−4) − | 8.2937e−2 (6.07e−2) |
|  | 10 | 1.2409e−1 (1.72e−2) − | 4.8998e−2 (5.36e−3) − | 8.4751e−2 (4.19e−2) − | 1.2798e−1 (1.85e−2) − | 1.1367e−1 (1.59e−2) − | **1.5730e−1 (5.07e−3)** + | 1.7567e−1 (1.81e−2) − | 5.1527e−8 (2.82e−7) − | 9.2729e−6 (5.06e−6) − | 1.4456e−5 (7.92e−5) − | 1.5109e−1 (7.48e−3) |
|  | 15 | 1.0529e−2 (4.47e−3) − | 5.9518e−4 (1.27e−3) − | **3.8934e−2 (5.49e−3)** + | 3.5485e−2 (9.58e−3) − | 5.7219e−3 (2.66e−3) − | 2.4147e−2 (4.10e−3) − | 1.0581e−2 (1.25e−2) − | 0.0000e+0 (0.00e+0) − | 0.0000e+0 (0.00e+0) − | 0.0000e+0 (0.00e+0) − | 3.4936e−2 (2.89e−3) |
| CF12 | 3 | 3.8683e−1 (2.11e−1) − | 4.2158e−2 (7.32e−2) − | 5.6890e−1 (2.59e−1) − | 3.4110e−1 (2.22e−1) − | 6.3439e−1 (2.06e−1) − | 4.9787e−1 (2.77e−1) − | 6.5318e−2 (8.63e−2) − | 6.4245e−1 (2.30e−1) = | 6.1007e−1 (2.18e−1) − | 6.8629e−1 (1.54e−1) − | **7.0904e−1 (1.17e−1)** |
|  | 5 | 7.0639e−1 (2.84e−1) − | 1.2679e−1 (1.16e−1) − | 8.7269e−1 (2.01e−1) − | 6.0749e−1 (2.02e−1) − | 8.2216e−1 (2.52e−1) − | 8.6234e−1 (1.58e−1) − | 4.5740e−1 (1.57e−1) − | 5.7809e−1 (3.20e−1) − | 7.4101e−1 (2.30e−1) − | 5.2243e−1 (2.70e−1) − | **9.4250e−1 (3.27e−2)** |
|  | 8 | 8.3101e−1 (1.66e−1) − | 2.9044e−1 (8.27e−2) − | 8.6778e−1 (1.02e−1) − | 8.1219e−1 (1.69e−1) − | 8.9698e−1 (1.58e−1) = | 9.1898e−1 (7.88e−2) − | 6.9062e−1 (9.47e−2) − | 2.6858e−1 (4.48e−2) − | 2.7911e−1 (5.48e−2) − | 2.6118e−1 (2.30e−2) − | **9.7087e−1 (2.17e−2)** |
|  | 10 | 9.7140e−1 (3.72e−2) − | 3.0165e−1 (8.13e−2) − | 9.8856e−1 (2.73e−2) − | 8.3423e−1 (1.83e−2) − | 9.7873e−1 (4.03e−2) − | 9.8856e−1 (5.68e−3) − | 8.2950e−1 (3.84e−2) − | 3.3259e−1 (2.30e−2) − | 4.1362e−1 (3.30e−2) − | 3.8638e−1 (1.76e−2) − | **9.9038e−1 (5.79e−3)** |
|  | 15 | 8.2411e−1 (1.20e−1) − | 3.1671e−1 (7.34e−2) − | 9.4826e−1 (7.77e−2) − | 9.2284e−1 (2.93e−2) − | 9.4371e−1 (1.07e−1) − | 9.1963e−1 (8.64e−2) − | 8.7218e−1 (9.69e−2) − | 3.4662e−1 (2.29e−2) − | 2.2330e−1 (2.19e−2) − | 5.1265e−1 (2.42e−2) − | **9.7434e−1 (6.94e−2)** |
| +/−/= | | 0/14/1 | 0/15/0 | 1/13/1 | 1/13/1 | 1/10/4 | 3/10/2 | 0/14/1 | 0/13/2 | 0/14/1 | 0/14/1 | | |

**Notes.**

The best results are in bold.

**Table 11** The IGD performance of dCMaOEA-RAE and the variants on DTLZ benchmark problems.

| Problem | M | dCMaOEA-RAE-I | dCMaOEA-RAE-II | dCMaOEA-RAE |
|---|---|---|---|---|
| | 3 | 2.0307e−2 (1.63e−4) = | 2.0479e−2 (5.54e−4) = | **2.0304e−2 (1.06e−4)** |
| | 5 | 5.1975e−2 (1.97e−4) = | 6.1374e−2 (5.13e−2) - | **5.1809e−2 (3.13e−4)** |
| C1_DTLZ1 | 8 | 9.5552e−2 (4.64e−4) - | 1.0287e−1 (1.21e−2) - | **9.5362e−2 (3.70e−4)** |
| | 10 | 1.0781e−1 (3.40e−4) = | 1.1133e−1 (5.99e−3) - | **1.0777e−1 (4.01e−4)** |
| | 15 | **1.8056e−1 (3.54e−3) =** | 1.8646e−1 (6.00e−3) - | 1.8093e−1 (2.25e−3) |
| | 3 | **5.4470e−2 (4.84e−6) =** | 2.2349e+0 (3.55e+0) - | 5.4471e−2 (5.79e−6) |
| | 5 | **1.6510e−1 (4.01e−5) =** | 2.4353e+0 (4.25e+0) - | 1.6511e−1 (5.45e−5) |
| C1_DTLZ3 | 8 | **3.1541e−1 (5.37e−4) =** | 4.8279e+0 (5.41e+0) - | 3.1542e−1 (4.01e−4) |
| | 10 | 4.1997e−1 (4.15e−4) = | 2.5805e+0 (4.71e+0) - | **4.1996e−1 (4.59e−4)** |
| | 15 | 6.2727e−1 (1.07e−2) = | 1.0509e+1 (6.08e+0) - | **6.2513e−1 (1.07e−2)** |
| | 3 | 4.7274e−2 (4.76e−4) - | 4.7139e−2 (5.42e−4) - | **4.6736e−2 (4.47e−4)** |
| | 5 | 1.3763e−1 (3.99e−4) - | 1.3775e−1 (3.98e−4) - | **1.3730e−1 (3.72e−4)** |
| C2_DTLZ2 | 8 | 2.4102e−1 (2.64e−2) - | 2.4176e−1 (2.07e−2) - | **2.3475e−1 (1.03e−3)** |
| | 10 | 2.5928e−1 (2.13e−3) - | 2.6511e−1 (5.23e−2) - | **2.5342e−1 (1.75e−3)** |
| | 15 | 2.8690e−1 (4.83e−2) = | **2.7273e−1 (2.16e−2) +** | 2.7792e−1 (3.85e−2) |
| | 3 | 1.2057e−1 (1.36e−1) = | **9.3242e−2 (5.43e−4) +** | 9.6727e−2 (6.76e−3) |
| | 5 | 2.4665e−1 (9.25e−4) = | **2.4623e−1 (1.17e−3) =** | 2.4646e−1 (9.17e−4) |
| C3_DTLZ4 | 8 | 4.5868e−1 (3.11e−2) = | 4.5755e−1 (3.35e−2) = | **4.5240e−1 (1.37e−3)** |
| | 10 | 5.6612e−1 (2.08e−3) = | 5.6809e−1 (1.90e−3) - | **5.6575e−1 (1.92e−3)** |
| | 15 | 8.0926e−1 (1.33e−2) = | 8.0865e−1 (1.04e−2) - | **8.0729e−1 (7.30e−3)** |
| | 3 | 1.3872e−2 (2.32e−4) - | 1.3633e−2 (2.21e−4) - | **1.3441e−2 (1.79e−4)** |
| | 5 | 3.9356e−2 (1.80e−4) - | 3.9801e−2 (1.79e−4) - | **3.9222e−2 (2.01e−4)** |
| DC1_DTLZ1 | 8 | 7.8816e−2 (3.73e−4) = | 7.9226e−2 (3.57e−4) - | **7.8712e−2 (4.62e−4)** |
| | 10 | 8.5651e−2 (2.45e−4) = | 8.5942e−2 (2.73e−4) - | **8.5535e−2 (3.50e−4)** |
| | 15 | 1.2824e−1 (3.14e−5) = | **1.2822e−1 (2.07e−4) =** | 1.2824e−1 (3.56e−5) |
| | 3 | 4.3553e−2 (9.34e−4) - | 4.1805e−2 (1.07e−2) - | **3.9421e−2 (9.47e−4)** |
| | 5 | 1.4300e−1 (7.63e−4) = | 1.4290e−1 (9.92e−4) = | **1.4282e−1 (8.97e−4)** |
| DC1_DTLZ3 | 8 | 3.3113e−1 (4.32e−2) = | 4.2799e−1 (9.94e−2) - | **3.2833e−1 (3.88e−2)** |
| | 10 | **4.2458e−1 (3.37e−2) =** | 4.5276e−1 (3.55e−2) - | 4.2703e−1 (3.11e−2) |
| | 15 | 6.1091e−1 (1.98e−2) = | 7.0825e−1 (8.42e−2) - | **6.0801e−1 (1.53e−2)** |
| | 3 | **2.0576e−2 (2.74e−5) =** | 2.5392e−2 (2.63e−2) = | 2.0585e−2 (3.16e−5) |
| | 5 | 5.2706e−2 (1.63e−5) = | **5.2703e−2 (1.08e−5) =** | 5.2703e−2 (1.24e−5) |
| DC2_DTLZ1 | 8 | 9.7176e−2 (5.01e−5) = | 1.0520e−1 (1.64e−2) = | **9.7173e−2 (4.00e−5)** |
| | 10 | 1.0926e−1 (8.35e−5) = | 1.2099e−1 (2.56e−2) = | **1.0924e−1 (9.25e−5)** |
| | 15 | 1.8335e−1 (8.73e−4) = | 1.9088e−1 (2.87e−2) = | **1.8312e−1 (9.08e−4)** |
| | 3 | 8.8731e−2 (1.13e−1) = | 5.6710e−1 (8.12e−3) - | **7.7453e−2 (8.03e−2)** |
| | 5 | **1.6503e−1 (7.61e−5) =** | 5.8769e−1 (7.98e−2) - | 1.6504e−1 (1.24e−4) |
| DC2_DTLZ3 | 8 | **3.6504e−1 (6.94e−2) =** | 7.2973e−1 (6.67e−2) - | 3.7867e−1 (6.42e−2) |
| | 10 | **4.4066e−1 (4.17e−2) =** | 7.9912e−1 (3.39e−2) - | 4.4248e−1 (3.25e−2) |
| | 15 | 6.3066e−1 (1.64e−2) = | 9.8775e−1 (1.77e−2) - | **6.2688e−1 (1.30e−2)** |

**Table 11** (*continued*)

| Problem | M | dCMaOEA-RAE-I | dCMaOEA-RAE-II | dCMaOEA-RAE |
|---|---|---|---|---|
| DC3_DTLZ1 | 3 | 9.8363e−3 (3.60e−4) - | 1.1807e−2 (1.80e−2) - | **8.1517e−3 (1.02e−4)** |
| | 5 | 2.2668e−2 (8.43e−4) - | **2.1306e−2 (4.72e−4) =** | 2.1367e−2 (7.46e−4) |
| | 3 | 3.2289e−2 (1.81e−2) - | 6.6723e−1 (2.40e−1) - | **2.3069e−2 (4.38e−4)** |
| | 5 | 8.2585e−2 (3.19e−3) - | 5.4976e−1 (1.42e−1) - | **7.8344e−2 (1.85e−3)** |
| DC3_DTLZ3 | 8 | **1.7583e−1 (4.57e−2) +** | 8.7158e−1 (3.96e−1) - | 1.8735e−1 (2.87e−2) |
| | 10 | 2.2435e−1 (8.87e−2) = | 7.1864e−1 (2.46e−1) - | **1.9546e−1 (8.80e−2)** |
| | 15 | 1.7843e−1 (6.40e−2) = | 1.0644e+0 (5.13e−1) - | **1.5429e−1 (1.73e−2)** |
| +/-/= | | 1/12/34 | 2/34/11 | |

**Notes.**
The best results are in bold.

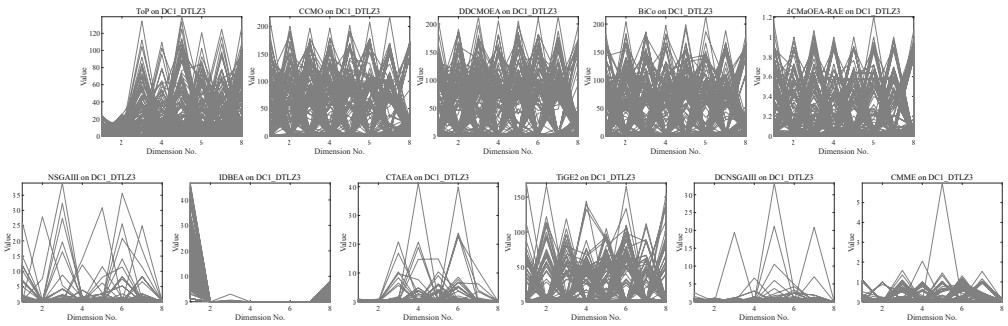

**Figure 10** **Result of solutions on the eight-objective DC1-DTLZ3 benchmark.**

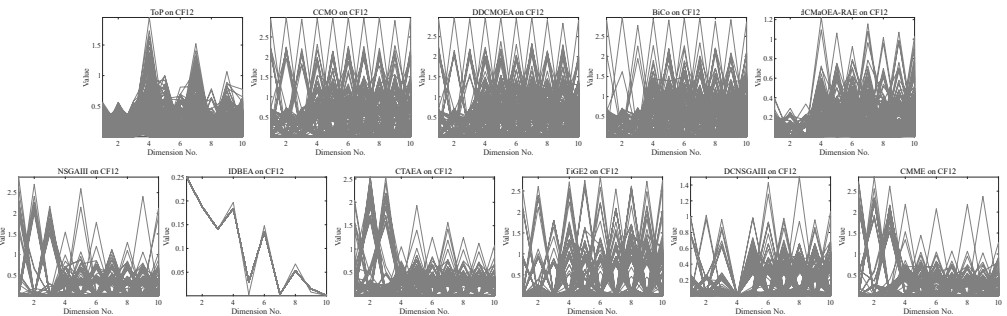

**Figure 11** **Result of solutions on the ten-objective CF12 benchmark.**

the 11 algorithms in problems like DC3-DTLZ3, DC1-DTLZ3, and C1-DTLZ3, with five, eight, and 15 objectives, respectively. To demonstrate the efficacy of the designed algorithm, we magnified part of the images. It is evident that dCMaOEA-RAE attains a more diverse and superior set of feasible solutions than the other algorithms. This further supports the high performance of the designed environmental selection strategy in enhancing the population's search for better feasible regions. It directed the main population to be evenly distributed within the feasible region. Additionally, Fig. 13 illustrates the evolution process

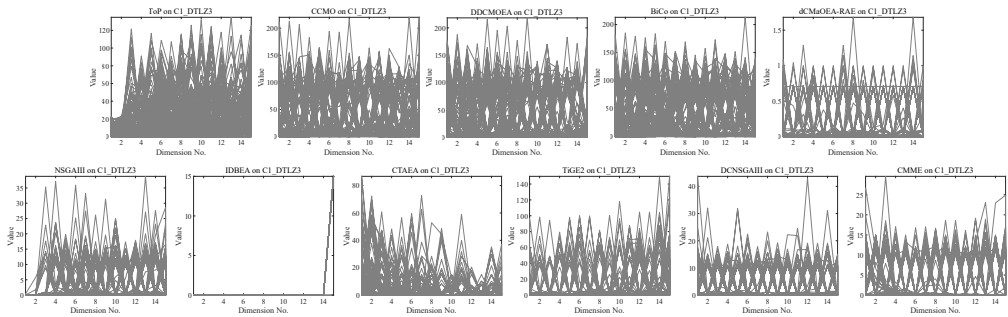

**Figure 12** Result of solutions on the fifteen-objective C1-DTLZ3 benchmark.

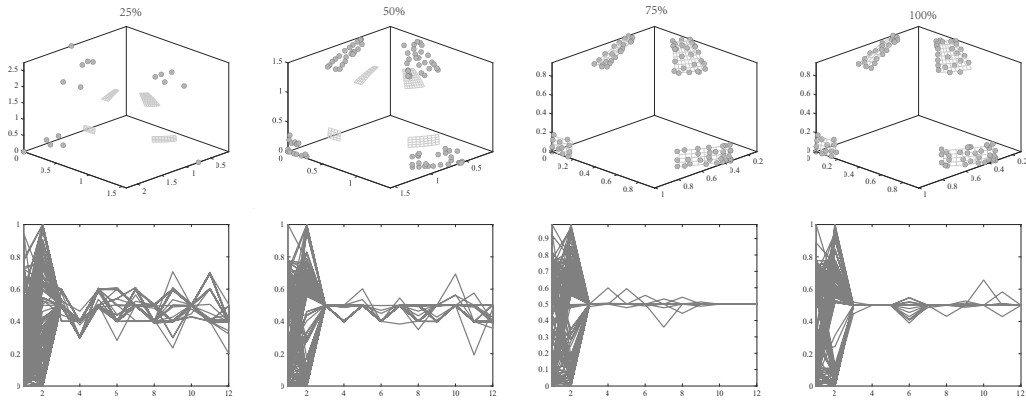

**Figure 13** Changes in decision variables and objective values of the conventional algorithm in C1-DTLZ3.

concerning changes in the objective space and decision variables for both the main and exploration populations. The proposed algorithm managed to maintain certain suboptimal solutions, thereby preventing the decision variables from becoming overly uniform and avoiding local optima.

## Summaries and discussion

Based on the aforementioned results, the performance of dCMaOEA-RAE in various types of CMaOPs can be summarized as follows:

- dCMaOEA-RAE demonstrated suitability for problems with complex constraints, such as C1-DTLZ3 and DC2-DTLZ1, due to its ability to retain solutions that might not perform well in the current context but are critical for the overall optimization process. Moreover, it effectively utilized information from these solutions to guide the population towards CPFs in a timely manner.
- Additionally, dCMaOEA-RAE was well-suited for problems featuring irregular or discrete CPFs, like CF4 and CF12. The selection strategy based on reference points and

angles adapted well to such CPFs by maintaining population distribution and obtaining a set of uniformly distributed solutions.

- Furthermore, dCMaOEA-RAE proved suitable for CMAOPs exhibiting multi-modal attributes, such as DC3-DTLZ3. This method achieved an optimal balance between convergence, diversity and feasibility.
- However, when it came to problems with simple constraints, other simpler and low-complexity methods achieved similar or better results than dCMaOEA-RAE, and the advantages of dCMaOEA-RAE were not demonstrated.
- dCMaOEA-RAE was also not suitable for problems with small feasible regions. The retention of poor solutions reduced pressure on the solution feasibility. Therefore, it often led to dCMAOEA-RAE being unable to identify all feasible solution regions adequately. Simultaneously, smaller feasible solution regions decreased in efficiency when diversity strategies were employed by dCMaOEA-RAE. This results in poorer population distribution.

## CONCLUSIONS

Within this report, we introduced a novel dual-population constrained many-objective optimization algorithm named dCMaOEA-RAE. This approach addresses the issue of reduced performance in traditional algorithms that happens. Specifically, we designed two distinct search strategies for the two populations to ensure that the population could identify the optimal feasible regions in a timely manner. We found that coordination between different populations when using dCMaOEA-RAE reached a satisfactory balance between feasibility, convergence, and diversity. However, there are still some limitations to be addressed. For instance, further enhancements are needed in the performance of dCMaOEA-RAE when dealing with problems featuring small feasible regions. The number of poor solutions retained in this method is fixed, which may affect the efficiency of population search. In addition, using the most efficient method possible to ensure the uniform distribution of the population on the irregular CPFs has also become a challenge. The above problems are likely to be solved by the theory of swarm intelligence algorithms. We suggest focusing on exploring this research field. In the future, we will make an attempt to add the theory of swarm intelligence algorithm and machine learning to create simple and efficient optimization algorithms that would enhance their effectiveness in handling CMaOPs.

### Funding

This work is supported by the National Natural Science Foundation of China under (Grant No. 61806138), the Central Government Guides Local Science and Technology Development Funds (Grant NO. YDZJSX2021A038), the China University Industry-University-Research Collaborative Innovation Fund (Future Network Innovation Research and Application Project) (Grant 2021FNA04014), the Key R&D program of Shanxi

Province, under Grant No. 202202020101012 and the Open Fund of State Key Laboratory for Novel Software Technology (Nanjing University) (Grant KFKT2022B18). The funders had no role in study design, data collection and analysis, decision to publish, or preparation of the manuscript.

## Grant Disclosures

The following grant information was disclosed by the authors:

National Natural Science Foundation of China: No. 61806138.

Central Government Guides Local Science and Technology Development Funds: NO. YDZJSX2021A038.

China University Industry-University-Research Collaborative Innovation Fund (Future Network Innovation Research and Application Project): 2021FNA04014.

Key R&D program of Shanxi Province: No.202202020101012.

Open Fund of State Key Laboratory for Novel Software Technology (Nanjing University): KFKT2022B18.

## Competing Interests

The authors declare there are no competing interests.

## Author Contributions

- Chen Ji conceived and designed the experiments, performed the experiments, analyzed the data, performed the computation work, prepared figures and/or tables, authored or reviewed drafts of the article, and approved the final draft.
- Linjie Wu conceived and designed the experiments, performed the experiments, analyzed the data, performed the computation work, prepared figures and/or tables, authored or reviewed drafts of the article, and approved the final draft.
- Tianhao Zhao conceived and designed the experiments, performed the experiments, analyzed the data, performed the computation work, prepared figures and/or tables, authored or reviewed drafts of the article, and approved the final draft.
- Xingjuan Cai conceived and designed the experiments, performed the experiments, analyzed the data, performed the computation work, prepared figures and/or tables, authored or reviewed drafts of the article, and approved the final draft.

## Data Availability

The source code is available in the Supplementary File.

The raw data is available at figshare: Ji, Chen (2024). dCMaOEA_RAE.7z. figshare. Dataset. https://doi.org/10.6084/m9.figshare.25893292.v1.

## Supplemental Information

Supplemental information for this article can be found online at http://dx.doi.org/10.7717/peerj-cs.2102#supplemental-information.

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
