# Peer review of "A dual-population Constrained Many-Objective Evolutionary Algorithm based on reference point and angle easing strategy"

_PeerJ Computer Science, doi:10.7717/peerj-cs.2102_

## Round 0.1 · original submission · Major Revisions

Dear authors,

Thank you for submitting your revised article. Reviewers have now commented on your article. Your article has not been recommended for publication in its current form. We encourage you to clearly address the concerns and criticisms raised by reviewers and resubmit your revised article once you have updated it accordingly.

Best wishes,

**Language Note:** The review process has identified that the English language must be improved. PeerJ can provide language editing services - please contact us at [email protected] for pricing (be sure to provide your manuscript number and title). Alternatively, you should make your own arrangements to improve the language quality and provide details in your response letter. – PeerJ Staff

Reviewer 1 ·

Basic reporting

The authors propose A dual-population constrained many-objective evolutionary algorithm based on reference point and angle easing strategy. I think paper is well-structured and explained; however, the manuscript needs to be revised. Here are some specific points:

1-In the introduction, it is imperative to refer to the complexity theory for constraint multi-objective optimization problem (CMOP) and elucidate the necessity for evolutionary algorithms, along with the rationale for proposing a new evolutionary algorithm. The discussion should incorporate the concept of the 'no free lunch' theory.

2-The motivation for discussing the limitations of existing algorithms. It should be clearly outlined in the introduction and not within the literature review.

3- The literature survey must be extended using recent works.

4-The conclusion must be revised to encompass discussions on the findings, limitations, recommendations, and future research works.

Experimental design

no comment

Validity of the findings

no comment

Additional comments

no comment

Reviewer 2 ·

Basic reporting

The paper presents a novel approach to addressing Constrained Many-Objective Optimization Problems (CMaOPs), which are prevalent in various real-life scenarios and characterized by their complex constraints and intricate Pareto frontiers. The primary challenge in CMaOPs is the traditional algorithms' tendency for premature convergence, which restricts the exploration of constrained Pareto frontiers and leads to suboptimal performance. To overcome this, the authors introduce a dual-population constrained many-objective evolutionary algorithm based on reference point and angle easing strategy (dCMaOEA-RAE). This algorithm employs a relaxed selection strategy to foster cooperation between dual populations, allowing for the retention of currently underperforming solutions that contribute to the overall optimization process. The paper demonstrates the algorithm's effectiveness through extensive experiments on 77 test problems, showing its superiority over 10 other state-of-the-art algorithms across three evaluation indicators. Overall, the paper is well-organized and carries out an extensive evaluation.

Experimental design

Comprehensive Evaluation: The algorithm is rigorously tested on a wide array of benchmark problems, and the comparison with 10 cutting-edge algorithms provides a strong validation of its effectiveness.

Validity of the findings

The evaluation results are very thorough and explained very clearly.

Additional comments

I have two additional improvement suggestions for this paper:

First, the authors mention a lot of related work in introduction section to illustrate the challenges of CMaOP. I may suggest put some discussion into the related work section.

Second, in line 415, the authors mention that dCMaOEA-RAE exhibits relatively inferior performance in D1-DTLZ1 and DC1-DTLZ1 and explains that it might be due to the simplicity of these test problems in terms of constraint complexity. I would recommend the authors to add a discussion section about what problems or situations are suitable for using dCMaOEA-RAE, and what are not.

---

## Round 0.2 · Minor Revisions

Dear authors,

Thank you for clearly addressing the reviewers' comments and suggestions. Although the previous review process has shown that the English language needs to be improved, the paper still needs to be proofread. Please pay particular attention to this.

Best wishes,

Reviewer 2 ·

Basic reporting

The paper is well-composed and I recommend accepting it.

Experimental design

no comment

Validity of the findings

no comment

---

## Round 0.3 · accepted · Accept

Dear authors,

Thank you for clearly addressing all the reviewers' comments. I confirm that the quality of your paper is improved. The paper now seems to be ready for publication in light of this revision.

Best wishes,

Reviewer 1 ·

Basic reporting

I have no other comments.

Experimental design

no comment

Validity of the findings

no comment

Additional comments

no comment